# A distributed and efficient population code of mixed selectivity neurons for flexible navigation decisions

Shinichiro Kira [1], Houman Safaai [1,2], Ari S. Morcos [1], Stefano Panzeri [2,3] & Christopher D. Harvey [1] ✉

Decision-making requires flexibility to rapidly switch one's actions in response to sensory stimuli depending on information stored in memory. We identified cortical areas and neural activity patterns underlying this flexibility during virtual navigation, where mice switched navigation toward or away from a visual cue depending on its match to a remembered cue. Optogenetics screening identified V1, posterior parietal cortex (PPC), and retrosplenial cortex (RSC) as necessary for accurate decisions. Calcium imaging revealed neurons that can mediate rapid navigation switches by encoding a mixture of a current and remembered visual cue. These mixed selectivity neurons emerged through task learning and predicted the mouse's choices by forming efficient population codes before correct, but not incorrect, choices. They were distributed across posterior cortex, even V1, and were densest in RSC and sparsest in PPC. We propose flexibility in navigation decisions arises from neurons that mix visual and memory information within a visual-parietal-retrosplenial network.

As animals navigate for survival, they combine signals from their sensory environment with internal information stored in memory to select a desirable route. Such navigation arises from a rich repertoire of sensorimotor associations that has expanded through evolution[1]. In reflexive behaviors, a given sensory input always leads to a stereotyped action. Animals have acquired the ability to rapidly switch the actions they take in response to a sensory stimulus depending on internally stored information in the form of memory. We refer to this ability as the flexibility of decision-making. In many laboratory decision-making paradigms, however, animals are trained to make one action in response to a given sensory cue and to make the opposite action in response to an alternate cue, which involves fixed sensorimotor associations but not the flexibility. In contrast, in flexible decision-making, animals switch their action in response to a given sensory cue from moment to moment, such as responding to the same sensory cue with one action at one moment and with the opposite action at the

next moment as the context changes. A critical feature of this flexibility is its rapidity to switch actions when information stored in memory or sensory cues changes from one moment to the next. This rapidity sets flexible decision-making apart from the learning or re-learning of different sensorimotor associations over longer timescales. Together, these characteristics imply that specific neural mechanisms exist for rapid flexibility over times as short as seconds. Here, we aimed to reveal the cortical areas and neural activity patterns that are central to flexible decisions during spatial navigation by understanding how information stored in short-term memory influences navigational action selection in response to sensory cues.

The flexibility of decision-making has often been investigated in experimental paradigms that do not involve spatial navigation. Across studies using different tasks, diverse areas have been found to mix memory and sensory information for flexible decisions, including higher sensory cortices, association cortices, and premotor

[1]Department of Neurobiology, Harvard Medical School, Boston, MA, USA. [2]Neural Computation Laboratory, Istituto Italiano di Tecnologia, Rovereto, Italy. [3]Department of Excellence for Neural Information Processing, Center for Molecular Neurobiology (ZMNH), University Medical Center Hamburg-Eppendorf (UKE), Hamburg, Germany. ✉e-mail: harvey@hms.harvard.edu

cortices[2–12]. Other studies have assessed slow changes in sensorimotor associations over many behavioral trials or sessions, which might rely on mechanisms distinct from those underlying rapid, moment-to-moment flexibility[13,14]. Recently, several studies have identified cortical and subcortical areas that have a causal role in flexible decisions[3,4,10,11,15]. However, in some studies, a limitation has been that many areas have not been systematically screened to compare their causal involvement and neural coding properties (but see Condylis et al.[4], Wu et al.[3]). Thus, it remains unclear whether flexible decisions involve different areas depending on the specific features of the task and/or are mediated by a widely distributed network. Furthermore, it is unknown whether these areas are involved in flexible decisions during navigation.

In contrast, many studies of navigation have focused on the encoding of current spatial variables, such as location and heading in place cells, grid cells, and head direction cells[16]. Beyond well-established spatial coding in hippocampus and entorhinal cortex, retrosplenial cortex (RSC) and posterior parietal cortex (PPC) represent heading direction, running velocity, and navigational routes with world-centered (allocentric) and self-centered (egocentric) reference frames[17–30]. In addition, spatial signals have been found even in primary and secondary visual cortices[31,32]. Often, however, these studies of navigation have not investigated the mechanisms of decision-making in which animals must choose a navigational path among alternatives.

Approaches have been developed to bridge navigation and decision-making. Earlier work has studied the interplay of navigation coding and working memory during decision-making in T-mazes[33–36]. More recently, studies have revealed that sequences of neural activity in PPC correlate with upcoming choices[37] and short-term memories of previous cues, including during evidence accumulation[38–41]. Similar choice-related sequential activity is also observed in RSC[41,42]. These approaches have used behavioral tasks with a fixed sensorimotor association needed for reward. For example, they employed a task in which cue A instructs turn left and cue B instructs turn right[37–39,42–44]. Thus, these studies have not investigated the flexibility of decision-making during navigation in which animals switch their action in response to a sensory stimulus depending on information stored in short-term memory.

Therefore, it remains unclear which areas may be most critical for the flexibility of decision-making during navigation. A leading candidate is PPC because of its established role in navigation decision tasks[37–39,45]. Another candidate is RSC due to its function in spatial memory and coding of navigation and decision-related variables[17,24,25,42,46–48]. Alternatively, flexible navigation decisions may arise from frontal regions of cortex that have been shown to be necessary for flexibility in tasks not involving navigation[3,15]. More generally, it is unclear if the flexibility of navigation decisions arises mostly from activity in one of these areas or if it occurs through distributed processing across cortex.

In addition, at the level of neural computation, it is an open question how memory signals are incorporated into the circuits important for navigation to mediate flexible decisions. One possibility is that the signals for memory and sensory cues are encoded in largely separate sets of neurons and converge onto neurons that relate to the behavioral choices of the mouse. In this case, there may exist separate groups of sensory, memory, and choice neurons. Alternatively, memory and sensory signals may be extensively combined in individual neurons in the form of mixed selectivity neurons. Such a code based on mixed selectivity could allow for an easy readout of arbitrary task variable combinations[49–51]. There could also be a hybrid coding scheme between these alternatives[52].

Here, we studied the flexibility of decision-making during navigation by designing a delayed match-to-sample task in virtual reality. We systematically screened the contributions of a wide range of cortical areas using optogenetics and cellular-resolution calcium imaging.

We demonstrate that neural activity in posterior cortex is necessary for flexible navigation decisions. We discovered neurons that mix short-term memory and visual information, and these neurons were present in most parts of posterior cortex, even in V1. Surprisingly, RSC had the highest density of these neurons, whereas these cells were sparsest in PPC, with a near absence in anterior PPC. These neurons formed an efficient population code, which appeared to support accurate decisions because their activity was more informative when the mouse made correct decisions compared to errors. This code emerged through the course of task learning. Our results suggest a mechanism contributing to flexible navigation decisions based on mixed visual and memory representations in individual neurons within a distributed visual-parietal-retrosplenial network.

## Results

### A task that requires combining short-term memory and sensory information to make flexible navigation decisions

We developed a delayed match-to-sample task for mice, based on navigation in a virtual reality T-maze (Fig. 1a). A black (B) or white (W) sample cue was presented on the walls at the start of the T-stem, followed by a delay segment in which the identity of the sample cue had to be stored in memory for a short period (1.21 ± 0.65 s, mean ± s.d., $n = 17$ mice). The delay segment duration was similar to or longer than delays used in other delayed match-to-sample tasks, including for human and non-human primates[5,7,53–55]. Next, when the mouse reached a defined spatial position, a test cue appeared instantaneously in one of two configurations: black walls on the left and white walls on the right (BW) or vice versa (WB). To receive a reward, the mouse turned toward the T-arm with the wall color that matched the sample cue (Fig. 1b). The two sample cues and two test cues defined four trial types (Fig. 1a, b, the example maze shows a B/WB trial). Importantly, in the test segment, the mouse combined its memory of the sample cue with the sensory information of the test cue to choose an appropriate action (left or right turn). This process involves flexible navigation decisions in the sense that the mouse made different choices (left or right turns) for the identical test cue depending on the short-term memory of the sample cue. This task is thus different from ones with fixed sensorimotor associations in which mice select the same behavioral action for a given sensory cue. Furthermore, the flexibility in this task was rapid and required mice to combine their short-term memory of the sample cue with the visual signals from the test cue differently from trial to trial. The mice performed the task with high accuracy after 2–4 months of training (90.5 ± 5.5% correct, mean ± s.d., $n = 17$ mice, Supplementary Fig. 1).

This task required the mouse to navigate through a virtual maze by using its movements to translocate in virtual space toward reward locations. The timing and progress of a trial were therefore controlled solely by the mouse, not the experimenter, as in real-world navigation. Previous work has shown that virtual mazes of this type trigger activity in navigation circuits, including place cells and grid cells, supporting the idea that this task involves spatial navigation[56–63]. To control the visual scene observed by the mouse, we fixed the heading angle and lateral position of the mouse in the virtual T-stem. Specifically, whereas the mouse could run in any direction on the spherical treadmill (Fig. 1c), our virtual reality software did not translate the mouse's lateral running velocity on the treadmill into movement in the virtual maze throughout the T-stem of the maze, and only did so at the T-intersection and T-arms (Fig. 1d). Consequently, along the T-stem during trials with the same cues, the mouse observed the identical visual scene on every trial (Fig. 1d).

The running velocities of mice on the treadmill for right and left choices diverged soon after the onset of the test cue (Supplementary Fig. 1i). Based on decoding of choice direction from these running patterns, we estimated that the relevant decision-making process that

combined visual and memory signals to inform choices happened within the first one second of the test segment (Supplementary Fig. 1i, j). Also, it appeared that the mouse used short-term memory to remember the sample cue, instead of using only a behavioral mnemonic, for several reasons. First, the mouse was unable to use heading angle and lateral position to remember the sample cue because these parameters were fixed and identical across all trials during the delay segment, as mentioned above (Fig. 1d). Second, although in some sessions mice had distinct running patterns on the treadmill for

different sample cues during the delay segment (arrow in Fig. 1c, bottom), these running patterns were variable across trials, and the mouse's choice on a trial was better explained by the presented sample cue than the running patterns during the delay (Fig. 1e). Third, the running patterns in the delay segment did not strongly correlate with task performance in individual mice, and sample-cue-related running in the delay segment was absent in some sessions with high performance (Supplementary Fig. 1k). Together, the running patterns suggest that the mice made decisions at the beginning of the test segment

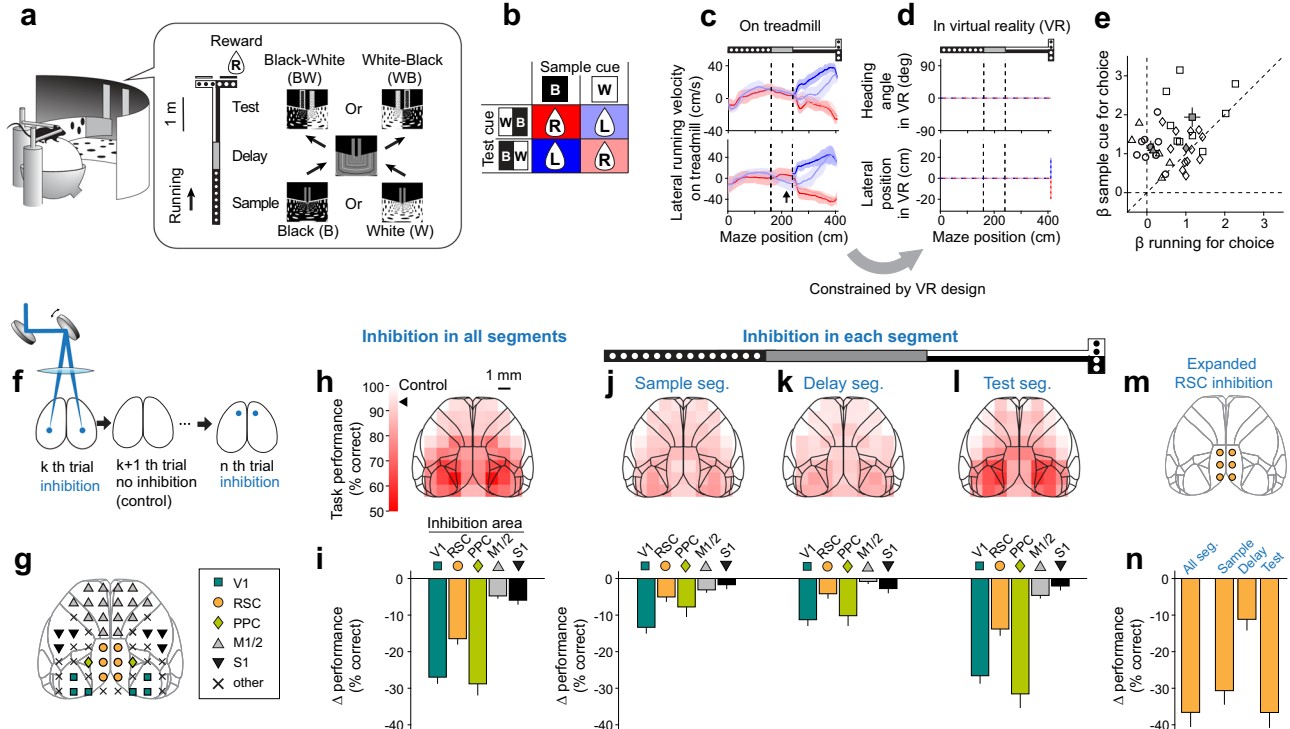

**Fig. 1 | Optogenetics screen for cortical areas involved in a flexible navigation task. a** Schematic of experimental setup and delayed match-to-sample task. **b** Reward direction on each of the four trial types defined by a combination of the sample cue and test cue. **c** Lateral running velocity of a mouse on the treadmill in two example sessions for four trial types colored as in panel (**b**). Shading indicates mean ± s.d. for correct trials. Prior to the test cue onset, trials with the black or white sample cue are colored by dark red and light blue, respectively. Arrow in the bottom panel indicates a position in the delay segment where the velocity differed between trials with the black and white-sample cue. **d** Heading angle (top) and lateral position (bottom) of a mouse in VR for four trial types shown as dashed lines and colored as in panel (**b**). **e** Logistic regression to explain the mouse's choice based on the sample cue identity vs. running patterns in the delay segment. Beta coefficients from a single session are shown as an open symbol, with different shapes indicating sessions from different mice. Filled symbols with error bars indicate mean ± s.e.m. for each mouse. $n = 35$ sessions from 4 mice. The data only include sessions in which calcium imaging data were acquired from posterior brain areas (V1, RSC, MM, A). **f** Schematic of optogenetic inhibition experiments. Bilateral light was delivered randomly to one of 28 pairs of target sites and interleaved with control trials (no laser). **g** Colored symbols indicate grouping of inhibition sites based on cortical areas defined in the Allen Mouse Brain Common Coordinate Framework[71]. **h** Task performance with inhibition throughout the trial for each cortical location (bilateral inhibition). The location of inhibition sites is overlaid on the cortical areal map based on the Allen Mouse Brain Common Coordinate Framework[71]. The average performance in control trials (93.5 ± 2.8%, mean ± s.d.) is indicated by an arrowhead on the color bar. $n = 212 ± 10$ trials/site (mean ± s.d.), 265 sessions, and 7 mice. **i** Change in task performance with inhibition in all segments (throughout the trial) relative to control trials. The change was computed from the data in panel (**h**) and averaged across sites in each brain area, as marked in panel (**g**). Error bars indicate mean ± s.e.m. The performance significantly

decreased with the inhibition in all areas ($p < 10^{-4}$; bootstrap, compared to zero). The effect of inhibition in V1, RSC, and PPC was larger than those for M1/M2 or S1 ($p < 10^{-4}$, bootstrap). The significance threshold was adjusted by Bonferroni correction with α = 0.05 to account for 5 area-wise comparisons for panels (**i**–**l**) and 10 between-area comparisons. V1: $n = 636$ trials, RSC: n = 618 trials, PPC: 216 trials, M1/2: $n = 1689$ trials, S1: 632 trials, from 7 mice. **j** Similar to panels (**h, i**), except with inhibition during the sample segment. $n = 141 ± 8$ trials/site (mean ± s.d.), 253 sessions, and 4 mice in panels (**j**–**l**). The performance significantly decreased with the inhibition in V1 ($p < 10^{-4}$, $n = 434$ trials), RSC ($p < 10^{-4}$, $n = 425$ trials), PPC ($p = 0.002$, $n = 138$ trials), M1/M2 ($p < 10^{-4}$, $n = 1130$ trials), but not in S1 ($p = 0.16$, $n = 397$ trials) (bootstrap, compared to zero). **k** Similar to panels (**h, i**), except with inhibition during the delay segment. The performance significantly decreased with the inhibition in V1 ($p < 10^{-4}$, $n = 409$ trials), RSC ($p = 0.0009$, $n = 438$ trials), PPC ($p < 10^{-4}$, $n = 151$ trials), but not in M1/M2 ($p = 0.25$, $n = 1141$ trials), and S1 ($p = 0.026$, $n = 424$ trials) (bootstrap, compared to zero). **l** Similar to panels (**h, i**), except with inhibition during the test segment. The performance significantly decreased with the inhibition in V1 ($p < 10^{-4}$, $n = 427$ trials), RSC ($p < 10^{-4}$, $n = 404$ trials), PPC ($p < 10^{-4}$, $n = 141$ trials), and M1/M2 ($p < 10^{-4}$, $n = 1110$ trials), but not in S1 ($p = 0.083$, $n = 420$ trials) (bootstrap, compared to zero). **m** Simultaneously inhibited six cortical sites for expanded RSC inhibition. **n** Similar to panels (**i**–**l**), except with expanded inhibition of RSC with six sites shown in panel (**m**) in all or specific segments. The performance significantly decreased with the inhibition in all segments ($p < 10^{-4}$, $n = 154$ trials, 38 sessions, 4 mice), sample segment ($p < 10^{-4}$, $n = 148$ trials, 37 sessions, 3 mice), delay segment ($p < 10^{-4}$, $n = 153$ trials, 37 sessions, 3 mice), and test segment: ($p < 10^{-4}$, $n = 139$ trials, 35 sessions, 3 mice) (bootstrap, compared to zero). Cortical areal map in panels (**g, h, j**–**m**) was adapted from Cell 181, Wang, Q., et al., The Allen Mouse Brain Common Coordinate Framework: A 3D Reference Atlas. 936-953, Copyright Elsevier (2020). Source data are provided as a Source Data file.

using both short-term memory of the sample cue and visual information of the test cue.

This task involves multiple neural processes, including flexible decision-making, short-term memory, and sensory perception, and each part of the task is critical for accurate behavioral performance. We thus analyzed all parts of the task, but because our goal is to study flexible decision-making during navigation, we focused mostly on the test segment in which decision-making occurred as mice combined a memory of the sample cue with the sensory information of the test cue.

### An optogenetics screen for cortical areas involved in a flexible navigation decision task

We screened for cortical locations with activity necessary for successful performance of the task. In VGAT-ChR2 mice, blue laser light was delivered transcranially to activate inhibitory interneurons and silence neighboring excitatory neurons (up to ~1 mm radius with 2–5 mW per site, Supplementary Fig. 2a–i)[64,65]. On a given trial, bilateral inhibition sites were chosen randomly from a grid of locations with 1 mm spacing (Fig. 1f, g). Inhibition trials were interleaved with control trials (Fig. 1f). With inhibition in all segments throughout a trial, the strongest impairment of task performance occurred when the inhibition site was chosen in primary and secondary visual areas, PPC, and RSC (Fig. 1h, i). In contrast, the effects on task performance were markedly smaller when the inhibition site was in the dorsal-anterior cortex, including motor and premotor areas, and somatosensory areas (Fig. 1h, i). Statistics and p-values for these and subsequent results are reported in the figure legends, based on two-tailed bootstrap tests adjusted for multiple comparisons unless noted otherwise (see Methods). Our inhibition results indicate that performing the task prominently involved the posterior parts of cortex, although it may also involve uninhibited parts of the brain, including subcortical areas, ventral cortex, or cortical neurons more than 1 mm from the surface (Supplementary Fig. 2g–i).

The inhibition of cortical sites did not induce apparent motor deficits in mice. Compared to control trials, the average running speed of mice varied only modestly across inhibition conditions ($3.1 \pm 11.0\%$ increase; mean ± s.d.; Supplementary Fig. 2j–o). Therefore, the decrease in performance was unlikely the result of difficulty running or decreased running speed that would result in a prolonged delay and potential memory decay.

To test if these posterior cortical areas were necessary only during specific trial epochs, we restricted inhibition to either the sample segment, delay segment, or test segment on each trial. Inhibition of V1, RSC, and PPC had significant effects on behavioral performance during each of the three trial segments. Thus, these areas appeared to have a role in each phase of the trial. However, the inhibition effects were largest in the test segment (Fig. 1j–l, $p < 0.01$, bootstrap test). Interestingly, only modest effects on performance occurred with inhibition of these areas during the delay segment, suggesting that the short-term memory might be maintained in a distributed cortical network, in uninhibited parts of the brain, or in a format other than neural spiking[66–68]. Also, although it is expected that V1, and perhaps other parts of posterior cortex, have a critical role in visual processing, the modest effects in the sample segment suggest that uninhibited parts of V1 can compensate for the inhibited locations.

Together, the optogenetics experiments identified cortical locations that contribute to the task. Furthermore, they revealed that each of the areas involved appeared to contribute in each task segment, which could indicate that these areas participate in multiple ways. In the context of our goal to identify areas that mix sensory and memory signals during decision-making, the strong inhibition effects during the test segment reveal V1, PPC, and RSC as candidates for this function. However, these inhibition effects could relate to other functions, such as visual processing. Because the optogenetics experiments were not designed to pinpoint the computations performed in each area, we proceeded to calcium imaging experiments to evaluate neural activity in V1, PPC, and RSC during the task.

### Neural activity represents essential task variables in a visual-parietal-retrosplenial network

We used two-photon calcium imaging to monitor layer 2/3 neurons in V1 (monocular region), RSC (dysgranular region), and PPC (Fig. 2a). We divided PPC into a medial region (area MM: mediomedial) and an anterior region (area A: anterior) because our previous work and anatomical studies suggest divisions exist in this part of posterior cortex[69–73]. Each imaged area had activity throughout the full trial with transient peaks associated with the onset of the sample cue and test cue, except for area A, which had a peak as the mouse turned into a T-arm (Fig. 2c). Individual neurons in each area had transient activity, and different cells were active at different time points, forming a sequence of activity that spanned the full trial (Fig. 2d). Many cells therefore contained activity selective for particular maze positions, and thus at any time point, only a small fraction of the population was active.

To gain initial insights into how cells could contribute to flexible decision-making, we looked for neural activity that combines the memory of the sample cue and the visual information of the test cue. This combination indicates the reward direction on a given trial, which can be interpreted as the logical exclusive OR (XOR) operation on the sample cue and test cue identities (Fig. 2b, top). The XOR takes on two values: one for the trial types B/BW and W/WB for which the reward is on the left, and a second value for the trial types B/WB and W/BW for which the reward is on the right. The reward direction and mouse's choice were identical on correct trials and opposite on error trials (Fig. 2b).

We performed an initial inspection of the activity in individual cells and found cells that appeared to encode key task variables. We deconvolved the calcium fluorescence time series to estimate the time points at which neural activity triggered calcium transients. Some cells were active on the two trial types with the same sample cue (Fig. 2e), and others were active on the two trial types with the same test cue (Fig. 2f), suggesting selectivity for the sample cue and test cue, respectively. However, these cells did not directly represent the XOR of the two cues to inform the reward direction. Other cells appeared to be choice selective with activity only when the mouse turned to a specific direction (left or right) regardless of the trial type (Fig. 2g). Notably, the choice selective cells tended to appear in the later part of the test segment (see the next section for analyses), whereas the mouse's choice-dependent running appeared at the initial part of the test segment (Fig. 1c, Supplementary Fig. 1i), suggesting that these cells were not the initial trigger for the mouse's choice.

Surprisingly, we also found cells that were active mostly on only one of the four trial types defined by a specific combination of a sample cue and a test cue. This type of activity is illustrated by an example neuron in Fig. 2h that was active mostly on B/WB trials. Neurons of this type have selectivity for the sample cue (B vs. W trials), test cue (BW vs. WB trials), and reward direction (XOR; B/BW and W/WB trials vs. B/WB and W/BW trials) (Supplementary Fig. 3). Such activity is an effective way to encode many task variables in single neurons and leads to separate representations of all four trial types across the population of cells. Importantly, these cells became active for a single trial type in the initial part of the test segment, which was early enough to influence upcoming choices. Interestingly, some single-trial-type selective cells responded differently on correct and error trials. Some cells were less active on error trials (Fig. 2i), and others were active for a different trial type on error trials (Fig. 2j), suggesting that the activity of these cells may be crucial for making accurate choices. Cells with each type of selectivity were found in multiple cortical areas, as will be shown in the next section.

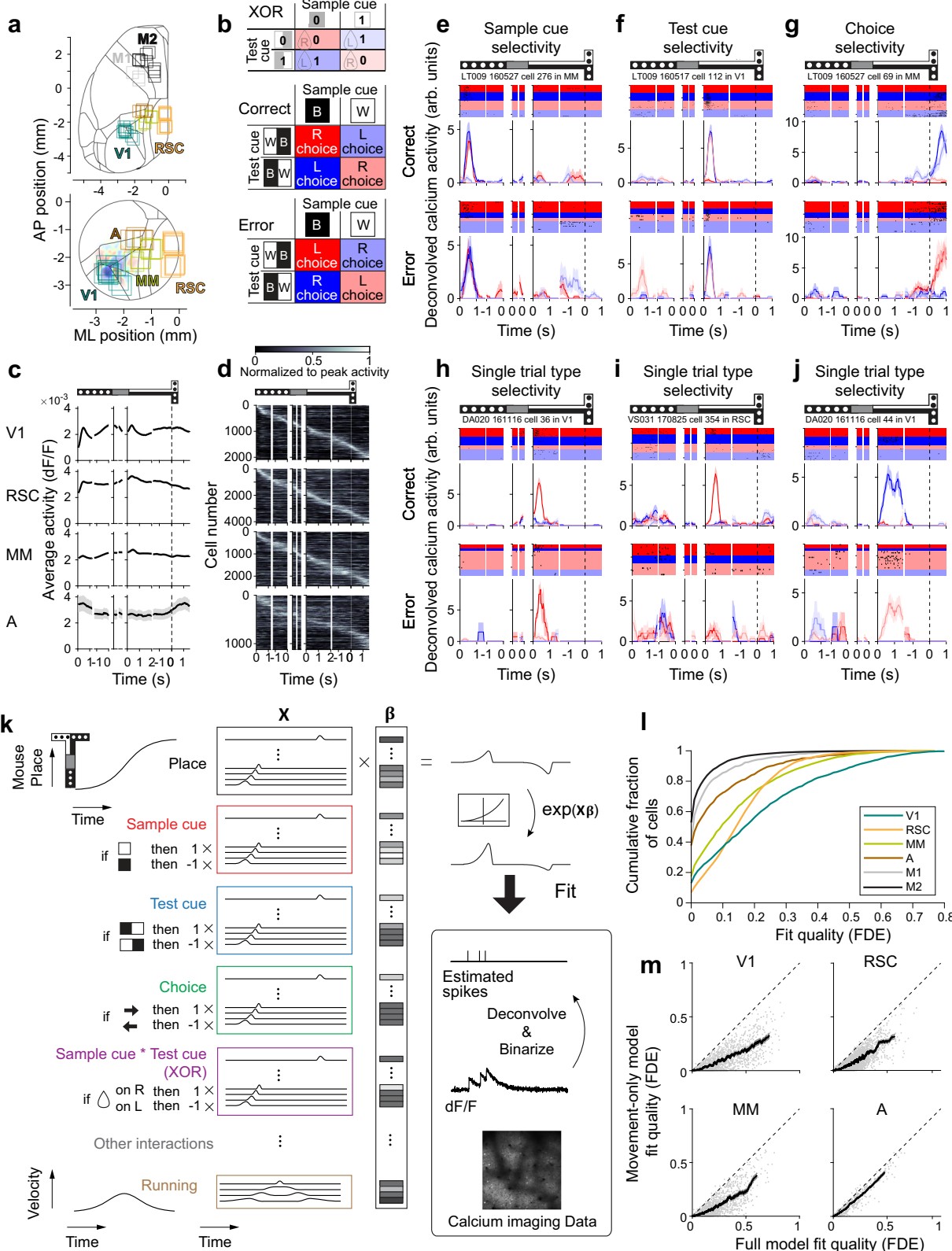

We evaluated the selectivity of individual cells in a more systematic manner using a generalized linear model (GLM)[39,74]. The GLM included each task variable (sample cue, test cue, choice) and their interactions (including interactions between sample cue and test cue to allow for XOR selectivity) as predictors of a neuron's activity (Fig. 2k, Methods). Because cells tended to be transiently active, we modeled each predictor as having selectivity conjunctive with maze position. Such position-specific selectivity is consistent with cortical activity useful for navigation[19,22,24,37]. In addition, because movements can substantially correlate with neural activity in posterior cortex[18,69,73,75,76], the GLM also included predictors for the mouse's running velocity, measured as rotations of the treadmill around three axes. We quantified the model's explanatory power as the fraction of deviance explained (FDE),

**Fig. 2 | Calcium imaging and related analysis of neural activity. a** Top: Individual imaging fields-of-view shown on the cortical areal map based on the Allen Mouse Brain Common Coordinate Framework[71]. Bottom: same except superimposed on the field sign map for retinotopy. The circular outline shows the typical location of the cranial window. **b** Reward direction on the four trial types determined by XOR combination of the sample cue and test cue (top). Choices follow XOR on correct trials (middle) or its opposite on error trials (bottom). **c** Average activity across neurons, aligned to the start and end of the sample segment, start and end of the delay segment, start of the test segment, and T-intersection (vertical dashed line). V1: $n = 2084$ cells, RSC: $n = 4103$ cells, MM: $n = 2439$ cells, A: $n = 1120$ cells. Shading indicates mean ± s.e.m. **d** Average activity normalized to its peak for each neuron (rows) sorted by time of peak average activity. The sequence of activity was cross-validated by plotting activity on even-numbered trials sorted by peak time on odd-numbered trials. **e** Example cell with sample cue selectivity in MM. Top row, correct trials; bottom row, error trials. Rasters of deconvolved and binarized calcium activity are shown for individual trials along with average activity (smoothed by running mean of 350 ms). Shading indicates mean ± s.e.m. Colors are the same as in panel (**b**). For error trials, the average activity is not shown for some trial types if they had less than three error trials per trial type. **f** Similar to panel (**e**), an example cell in V1 with test cue selectivity. **g** Similar to panel (**e**), an example cell in MM with choice selectivity. **h** Similar to panel (**e**), an example cell in V1 with single-trial-type

selectivity. **i** Similar to panel (**e**), an example cell in RSC with single-trial-type selectivity. **j** Similar to panel (**e**), an example cell in V1 with single-trial-type selectivity. **k** Schematic of the GLM fitted to the deconvolved and binarized calcium activity of each neuron. Predictors were divided into groups for task variables and movement. Task variables were basis expanded with position along the maze to reflect the sequential activity observed in panel (**d**). See Methods for full details. **l** GLM fit quality, measured as the fraction of deviance explained (FDE) on test data. Cells with converging fits for the full GLM were included. FDE per cell (mean ± s.e.m.); V1: 0.212 ± 0.005 ($n = 1744$ cells; 84% of detected cells), RSC: 0.155 ± 0.002 ($n = 3865$ cells; 94%), MM: 0.139 ± 0.003 ($n = 2310$ cells; 95%), A: 0.081 ± 0.004 ($n = 1106$ cells; 99%), M1: 0.045 ± 0.003 ($n = 883$ cells; 100%), M2: 0.027 ± 0.001 ($n = 3243$ cells; 100%). The mean FDE was significantly different across areas ($p < 10^{-4}$). **m** Comparison of model fits for the full model (task and movement variables) and the movement-only model (no task variables). Dots indicate individual cells that had converging fits for both models. Black traces show the running mean across cells (window size, 50 cells). Shading indicates mean ± s.e.m. V1: $n = 1744$ cells (84% of detected cells), RSC: $n = 3865$ cells (94%), MM: $n = 2310$ cells (95%), A: $n = 1081$ cells (97%). Cortical areal map in panel (**a**) was adapted from Cell 181, Wang et al., The Allen Mouse Brain Common Coordinate Framework: A 3D Reference Atlas. 936-953, Copyright Elsevier (2020). Source data are provided as a Source Data file.

computed on test data left out from fitting (Fig. 2l, m, Supplementary Fig. 4).

Cells in V1 were best explained by the GLM across the imaged areas, followed by cells in RSC and MM (Fig. 2l). The large majority of cells in these areas were better explained by the full GLM that included task variables and running variables than by a reduced GLM that only included running variables (Fig. 2m). The task variables thus contributed significantly to neural activity in V1, RSC, and MM. In contrast, most cells in area A were well explained by the GLM with running variables alone, indicating that A had activity predominantly related to the movements of the mouse. We also imaged cells in M1 and M2, even though optogenetic manipulation of these areas did not strongly affect task performance (Fig. 1h–l). The GLM poorly predicted neural activity in M1 and M2 (Fig. 2l) although the mice showed similar performance during imaging sessions for each area (Supplementary Fig. 5a, b), suggesting these areas may be less informative for solving the task compared to posterior cortical areas.

We evaluated the information in each neuron about each task variable by using the GLM framework to compute the likelihoods of neural activity conditioned on task and movement variables[39] (Methods). At each time point in each trial, we estimated how likely one identity of a given variable was relative to the other (e.g., black vs. white sample cue) by computing a log-likelihood ratio (logLR), a well-established measure of single-trial neural information[77–80]. Although logLR is a signed value that indicates the estimated identity of the variable, we adjusted the sign so that logLR was positive (or negative) for neural activity representing the correct (or incorrect) identity of the task variable in a trial. The magnitude of logLR was larger for more informative neural representations of the task variable. We used logLR to quantify information about task variables in the subsequent analyses. Potential confounds concerning locomotion-related neural activity were mitigated by using the GLM that included all task and movement variables in a single model and by conditioning on the movement variables when computing information metrics, which together helped isolate the neural coding for a task variable of interest.

**Widespread but distinct encoding distributions across cortical areas**

We first tried to identify an area critical for mixing sample cue and test cue information by looking at major encoding patterns in individual cells. We therefore focused on the early part of the test segment and analyzed the encoding of the sample cue, test cue, and the mixture of the two that indicates the reward direction (XOR) for solving the task.

We will initially show the magnitude of information per cell by averaging single-cell information across cells in each area (Fig. 3, Fig. 4, Fig. 5) and then evaluate population information in a later section (Fig. 6). In this section, we used only correct trials to avoid analyzing complex modulations between correct and error trials (e.g., Fig. 2i, j). We note that XOR information and choice information are identical when considering only correct trials because the reward direction and the mouse's choice perfectly match on correct trials.

In V1, the dominant information at any time point was related to the cue that was currently visible. V1 cells had high sample cue information on average in the sample segment that decayed during the delay segment and high test cue information in the test segment (Fig. 3a, d, e). Despite the predominance of information about the current visual cue, V1 also contained substantial XOR information about the reward direction (Fig. 3b, c, f). To understand the mixing of sample cue information and test cue information in single cells in the decision-making period (i.e., the beginning of the test segment; gray shading in Fig. 3a–c), for each cell, we plotted its sample cue information in the test segment versus its test cue information in the test segment (Fig. 3g). We used the polar angle as a measure for how much a cell encoded one cue relative to the other. Cells residing close to 0° (horizontal-axis) or 90° (vertical-axis) corresponded to those encoding mostly sample cue or test cue information, respectively. In contrast, cells located around the diagonal (45°) had mixed representations of both cues. In the test segment, many V1 neurons had high information about the test cue and much less information about the sample cue (Fig. 3g–i). V1 thus prominently represented the current visual stimulus and had a skewed distribution of information that strongly favored the test cue in the test segment.

The distribution of information in RSC was strikingly different. RSC had approximately equal levels of sample cue and test cue information per cell in the test segment, and thus its activity was less dominated by the current visual cue (Fig. 3a, d, e). Importantly, RSC had on average significantly larger XOR information per cell than V1 (Fig. 3b, c, f). This XOR information rose from the onset of the test segment and was present even before mice began to report their choice in the form of a turn direction (Fig. 3b, c, Supplementary Fig. 1i, j, Supplementary Fig. 5c). XOR information thus appeared early enough to influence the decision-making process. The most striking feature of RSC activity was the extent to which sample cue information and test cue information were mixed at the level of single cells (Fig. 3g, h). Many cells had approximately equal sample cue information and test cue information in the test segment (Fig. 3g, h). The distribution of information in RSC

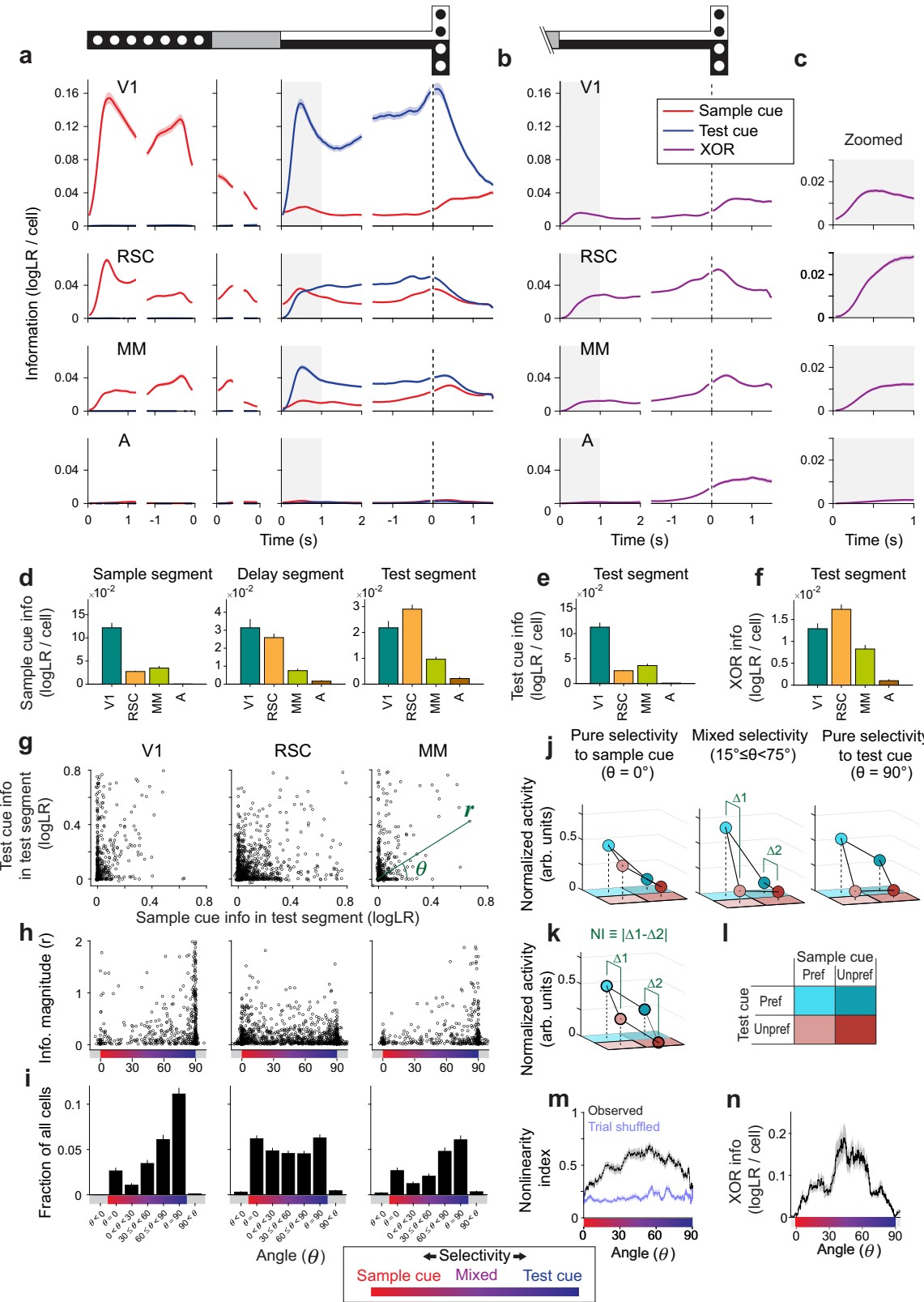

cells in the test segment was approximately uniform between sample cue selective, mixed selective, and test cue selective cells, which was markedly different from the distribution in V1 (Fig. 3i). As a result, RSC contained a larger fraction of cells that equally mixed the memory information of the sample cue and visual information of the test cue than V1. Interestingly, RSC's sample cue information was maintained by the sequential activation of cells during the delay segment until it was

mixed with the test cue information (Fig. 3a, Supplementary Fig. 5d–f). These results show that RSC has approximately equal mixing of information, the largest fraction of cells with mixed information, and the highest XOR information per cell.

The profile of information per cell in MM was intermediate between V1 and RSC. MM also contained information about the sample cue, test cue, and XOR (Fig. 3a–f), with a profile of information in the

**Fig. 3 | Task-related information and its mixing in posterior cortical areas.**
**a** Sample cue and test cue information per cell quantified as logLR. The information in individual cells was averaged across correct trials and then across cells with a converging fit for the full GLM. Shading indicates mean ± s.e.m. The time series of information is aligned to and shown for the beginning and end of each segment (sample segment: first 1.2 s and last 1.2 s, delay segment: first 0.3 s and last 0.3 s, test segment: first 2 s and last 1.5 s prior to T-intersection, T-arm: 1.5 s after T-intersection). Gray regions indicate the period (first one second) analyzed for the test segment in panels (**d–n**). V1: $n = 1962$ cells (94% of detected cells), RSC: $n = 4052$ cells (99%), MM: $n = 2409$ cells (99%), A: $n = 1096$ cells (98%) for panels (**a–i**).
**b** Similar to panel (**a**), except for XOR information. **c** Zoomed view of XOR information from panel (**b**) for the first one second of the test segment. **d** Average sample cue information per cell in the sample, delay, and test segments. The information was averaged over the last 1 s for the sample segment, last 0.35 s for the delay segment, and first 1 s for the test segment. Error bars indicate mean ± s.e.m. The information was significantly different between areas ($p < 10^{-4}$), except for between RSC and MM in the sample segment ($p = 0.11$) and between V1 and RSC in the delay ($p = 0.31$) and test segments ($p = 0.03$). All p values were calculated by bootstrap. The significance threshold was adjusted by Bonferroni correction with $\alpha = 0.05$ to account for 6 between-area comparisons for panels (**d–f**). **e** Similar to panel (**d**) except for test cue information in the test segment. The information was significantly different between areas ($p \leq 0.0080$). **f** Similar to panel (**d**) except for XOR information in the test segment. The information was significantly different between areas ($p \leq 0.0074$). **g** For each cell (circle), the sample cue information in the test segment and the test cue information in the test segment on correct trials. **h** Data from panel (**g**) replotted in polar coordinates as the magnitude ($r$) and angle ($\theta$). Cells closer to 0 degrees have more sample cue information, and cells closer to 90 degrees have more test cue information. Skewness of the distribution (mean ± s.e.m.); V1: $-1.20 \pm 0.10$, RSC: $-0.06 \pm 0.05$, MM: $-0.67 \pm 0.11$. Skewness was computed without cells with extreme angles (the highest and lowest 1% of cells) or noise-level information (magnitude $r < 0.01$). The distribution was significantly skewed for V1 and MM ($p < 10^{-4}$), but not for RSC ($p = 0.88$). The skewness was significantly different between V1, RSC, and MM ($p < 0.01$). All p values were calculated by bootstrap. **i** Distribution of cells from panel (**h**) in discrete angle bins.

Bins for $\theta = 0°$ and $\theta = 90°$ included cells on the axes and those with chance-level deviation from the axes (Methods). Error bars indicate s.e.m. The fraction of cells with equal mixing ($30° \leq \theta < 60°$) was significantly greater in RSC than in V1 or MM ($p < 10^{-4}$). Cells with noise-level information (magnitude $r < 0.01$) were not assigned angles but included in the total number of cells to calculate the fractions (see Methods). **j** Normalized mean activity for the four trial types. The sum of mean activity across the four trial types in each cell was normalized to one. For each cell, the trial type with the highest mean activity was defined to have a preferred sample cue and preferred test cue. With respect to this trial type, the other three trial types had preferred or unpreferred cues, as shown in panel (**l**). The cue preference of each cell was cross-validated (Methods). Cells from V1, RSC, and MM were combined because of their similarities (Supplementary Fig. 5g–i). Error bars inside the colored circles indicate mean ± s.e.m. Pure sample cue selectivity: $n = 350$ cells, mixed selectivity: $n = 640$ cells, pure test cue selectivity: $n = 568$ cells.
**k** Schematized activity distributions across the four trial types. The schematic illustrates mixed selectivity where changes associated with different test cue identities ($\Delta1$ and $\Delta2$) do not depend on the sample cue identity ($\Delta1 = \Delta2$). This dependency was quantified as nonlinearity Index (NI $\equiv |\Delta1\text{-}\Delta2|$. NI = 0 indicates linear mixing (the lowest level of nonlinear mixing) as in this schematic, and NI = 1 indicates the highest level of nonlinear mixing. **l** Color scheme used for preferred and unpreferred cues in panels (**j**, **k**). **m** Nonlinearity index for cells across angles (running average, window of 50 cells). Cells from V1, RSC, and MM were combined because of their similarities (Supplementary Fig. 5j). Cells with noise-level information (magnitude $r < 0.01$) were excluded. Shading indicates mean ± s.e.m. Blue trace shows chance nonlinearity index value computed with shuffled trial identities. Nonlinearity Index (mean ± s.e.m.); mixed selectivity cells ($15° \leq \theta < 75°$): $0.56 \pm 0.01$, pure sample cue selectivity cells ($\theta = 0°$): $0.25 \pm 0.01$, pure test cue selectivity cells ($\theta = 90°$): $0.32 \pm 0.01$. Note that the chance level NI is $0.198 \pm 0.004$ after shuffling trial identities. $n = 2046$ cells. **n** XOR information for cells across angles (running average, window of 50 cells). Cells from V1, RSC, and MM were combined because of their similarities (Supplementary Fig. 5k). Cells with noise-level information (magnitude $r < 0.01$) were excluded. Shading indicates mean ± s.e.m. $n = 2046$ cells. Source data are provided as a Source Data file.

test segment biased toward the test cue (Fig. 3i). This intermediate profile is consistent with MM residing at the interface of two spatial encoding gradients centered at V1 and RSC[69,73]. In contrast to the other areas, single cells in area A lacked information about the sample cue and test cue throughout the trial and had XOR information only when the mouse started turning at the T-intersection (Fig. 3a–f). The lack of prominent encoding of task variables in area A was consistent with its activity being mostly explained by the locomotion of the mouse (Fig. 2m). When considering MM and A together, PPC surprisingly had the smallest fraction of cells with mixed information, and thus, at least in its anterior portion, PPC may not be a key area for mixing memory and visual information. M1 and M2 had little information about the task variables in the initial part of the test segment, but they showed choice-related XOR information as mice made a turn at the T-intersection (Supplementary Fig. 6). Similar trends were observed using population decoding methods with and without deconvolution of the calcium fluorescence time series (Supplementary Fig. 7), and when we restricted analysis to only the earlier part of the test segment (first 0.5 s; Supplementary Fig. 8).

Notably, the cells with mixed sample cue information and test cue information tended to be the cells active preferentially on single trial types (single-trial-type selective cells; Fig. 2h–j). We quantified this observation by comparing the mean activity on each of the four trial types, normalized by the sum of the mean activities across the four trial types (Fig. 3j, Supplementary Fig. 5g–i). Therefore, if a cell is active on only one trial type, this trial type would have a normalized activity value of one. Instead, if a cell is active equally on all four trial types, then each trial type would have a normalized activity value of 0.25. On average, mixed selectivity cells had activity for one trial type that was about 6 times greater than for the other trial types (Fig. 3j, center). Therefore, these cells tended to respond to a specific combination of a sample cue and a test cue, showing striking differences from cells with

pure selectivity to either the sample cue or test cue (Fig. 3j). From a computational perspective, these cells mixed the information of the sample cue and test cue in a nonlinear manner. That is, the activity change associated with different test cues depended on the memory of the sample cue in the same trial ($\Delta1 \neq \Delta2$ in Fig. 3j). This contrasts with linear mixing, where changes associated with one variable are independent of the other variable ($\Delta1 = \Delta2$ in Fig. 3k). Further, a nonlinearity index (NI $\equiv |\Delta1\text{-}\Delta2|$) ranging from 0 (linear mixing) to 1 (most nonlinear mixing) confirmed the nonlinear mixing in the mixed selectivity cells (NI = $0.59 \pm 0.02$, mean ± s.e.m., Fig. 3m, Supplementary Fig. 5j). This nonlinear mixing allows single neurons to encode XOR information about the reward direction (Fig. 3n, Supplementary Fig. 5k) and can be advantageous for linear decoding in downstream areas[49,50].

Surprisingly, we observed very few cells that encoded only the reward direction (or choice direction). On correct trials, these cells would be active on two of the four trial types and have XOR information but not sample cue or test cue information. When considering all the cells with appreciable XOR information in the early part of the test segment (logLR > 0.01, noise estimated by the test cue information in the sample segment; Methods), approximately 85% of these cells also had sample cue and/or test cue information (Supplementary Fig. 5l, m). Furthermore, of all the cells with XOR information, roughly two-thirds contained sample cue information prior to the test cue onset (Supplementary Fig. 5n, o). Therefore, the dominant carriers of XOR information for the reward direction were the cells that encoded multiple task variables in the form of single-trial-type selectivity.

While our goal was to identify neural activity related to the mixing of information about the sample cue and the test cue, which led us to focus on the early part of the test segment, we noticed prominent signals related to the sample cue in V1, RSC, and MM in the sample and delay segments. In addition, in the test segment, there were signals

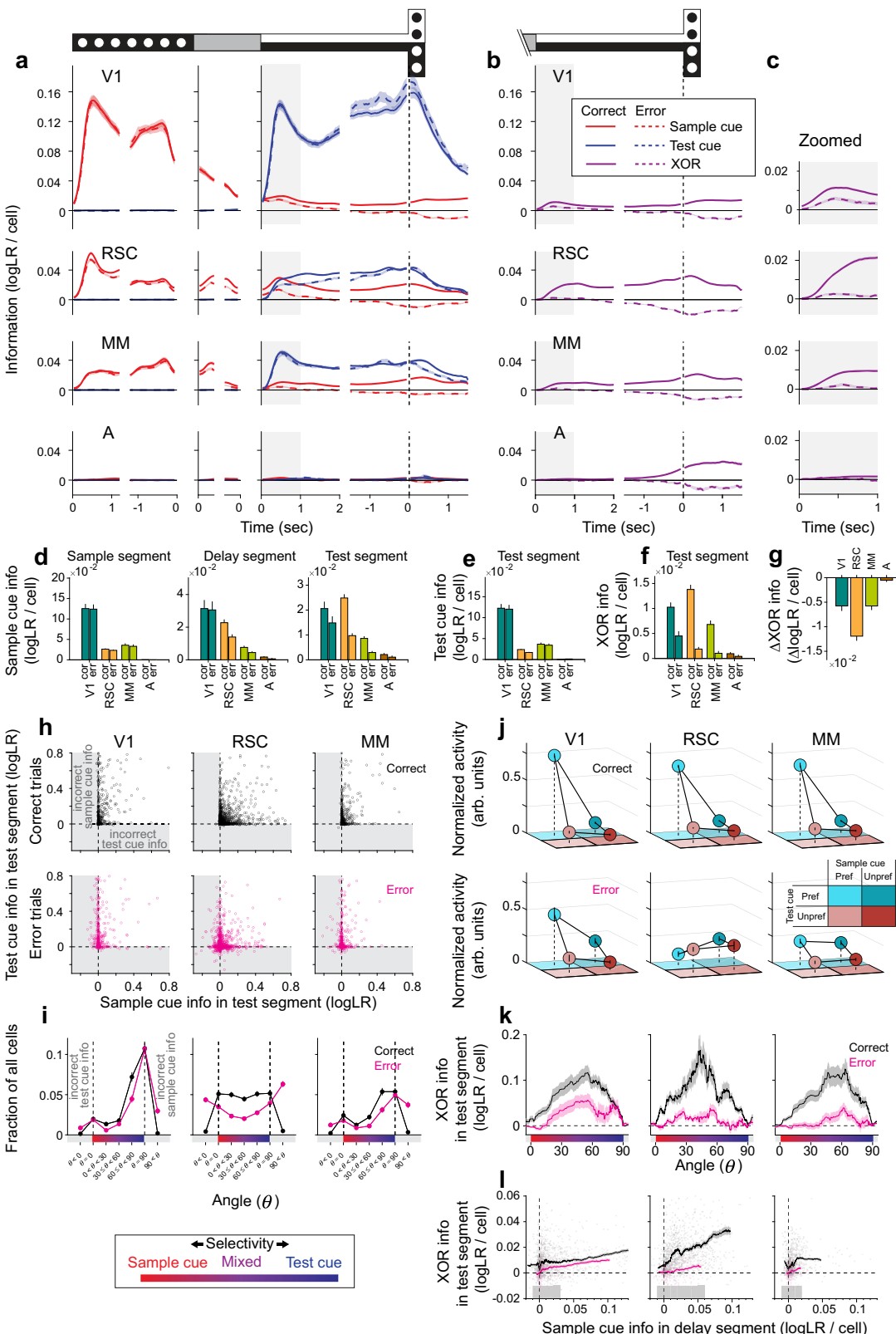

that mostly encoded information about the test cue. Together, these signals suggest that each area likely performs additional functions, such as visual processing, besides the mixing of memory and visual signals.

Together, these results reveal widespread representations in posterior cortex. Visual information was robustly represented in V1, RSC, and MM as demonstrated by the large amount of sample cue

information per cell in the sample segment and test cue information per cell in the test segment. The task required mixing of the current visual information and memory information, which can generate XOR information to signal the reward direction. This mixing manifested as single-trial-type selectivity, which appears to be an effective way for single neurons to encode many relevant task variables, including the reward direction. These single-trial-type selective neurons and XOR

**Fig. 4 | Comparing correct and error trials to identify activity patterns important for accurate decision-making. a** Sample cue and test cue information per cell quantified as logLR for correct (solid) and error (dashed) trials. The information in individual cells was averaged across trials and then across cells with converging fits for the full GLM. Shading indicates mean ± s.e.m. Gray regions indicate the period (first one second) analyzed for the test segment in panels (**d**–**l**). V1: $n = 1744$ cells (84% of detected cells), RSC: $n = 3865$ cells (90%), MM: $n = 2310$ cells (95%), A: $n = 1105$ cells (99%) for panels (**a**–**i**). **b** Similar to panel (**a**), except for XOR information. **c** Zoomed view of XOR information from panel (**b**) for the first one second of the test segment. **d** Average sample cue information per cell in the sample, delay, and test segments for correct and error trials. The information was averaged over the last 1 s for the sample segment, last 0.35 s for the delay segment, and first 1 s for the test segment. Error bars indicate mean ± s.e.m. The difference between correct and error trials in the sample segment was significant in RSC ($p = 0.004$), but not for V1 ($p = 0.55$), MM ($p = 0.03$), and A ($p = 0.03$). The difference in the delay segment was significant for RSC and MM ($p < 10^{-4}$), but not for V1 ($p = 0.47$) and A ($p = 0.03$). The difference in the test segment was significant for V1, RSC, and MM ($p < 10^{-4}$), but not for A ($p = 0.04$). All $p$ values were calculated by bootstrap for panels (**d**–**g**). The significance threshold was adjusted by Bonferroni correction with $\alpha = 0.05$ to account for 4 area-wise comparisons for panels (**d**–**g**), and 6 between-area comparisons for panel (**g**). **e** Similar to panel (**d**) except for test cue information in the test segment. The difference was significant in RSC ($p < 10^{-4}$), but not for V1 ($p = 0.50$), MM ($p = 0.10$), and A ($p = 0.08$). **f** Similar to panel (**d**) except for XOR information in the test segment. The difference was significant in V1, RSC, and MM ($p < 10^{-4}$), but not for A ($p = 0.08$). **g** Difference in XOR information between correct and error trials in the test segment, calculated per cell and averaged across cells. Error bars indicate mean ± s.e.m. The decrease was significantly different from zero for V1, RSC, and MM ($p < 10^{-4}$), but not for A ($p = 0.08$). The amount of decrease was significantly different between areas ($p < 10^{-4}$), except for between V1 and MM ($p = 0.98$). **h** For each cell (circle), the sample cue information in the test segment and the test cue information in the test segment on correct trials (top/black) and error trials (bottom/magenta). Cells plotted in the top right quadrant correctly encoded the identity of the cues, and those plotted in other quadrants incorrectly encoded the identity of the sample cue, test cue, or both (gray shading). **i** Distribution of cells from panel (**h**) in discrete polar angle bins for

correct (black) and error (magenta) trials. Bins for $\theta < 0°$ and $90° < \theta$ show the fraction of cells that incorrectly encoded the cue identity (gray shading). Cells with noise-level information (magnitude $r < 0.01$) were not assigned angles but included in the total number of cells to calculate the fractions. The fraction of cells was significantly different between correct and error trials ($p < 0.002$) in the following bins; V1: $\theta < 0°$, $60° \leq \theta < 90°$, $90° < \theta$; RSC: all bins except for $\theta = 90°$; MM: $\theta < 0°$, $30° \leq \theta < 60°$, $60° \leq \theta < 90°$, $90° < \theta$. The significance threshold was adjusted by Bonferroni correction with $\alpha = 0.05$ to account for 7 bin-wise comparisons. Error bars indicate s.e.m and are smaller than the data marker for some bins. **j** Top: Normalized mean activity of mixed selectivity cells for the four trial types on correct trials. Bottom: mean activity on error trials scaled by the normalization factors computed on correct trials. Nonlinearity Index of mixed selectivity cells ($15° \leq \theta < 75°$ in panel (**h**)) on correct trials was $0.64 \pm 0.03$ (mean ± s.e.m.) for V1 ($n = 115$ cells), $0.52 \pm 0.02$ for RSC ($n = 350$ cells), $0.53 \pm 0.03$ for MM ($n = 106$ cells). The mean activity for the preferred trial type was significantly lower on error trials than on correct trials in V1 ($p = 0.0006$), RSC ($p < 10^{-4}$), and MM ($p < 10^{-4}$). **k** XOR information for cells across angles (running average, window of 50 cells) for correct (black) and error (magenta) trials. The angle was defined on correct trials in panel (**h**). Shading indicates mean ± s.e.m. Cells with noise-level information (magnitude $r < 0.01$) were excluded. V1: $n = 487/475$ cells, RSC: $n = 1055/1035$ cells, MM: $n = 418/409$ cells were included for the analysis of correct/error trials. **l** Comparison of XOR information on correct (black) and error (magenta) trials, controlling for the sample cue information immediately before making decisions (last 0.35 s in the delay segment). Data points indicate individual trials, showing the sample cue information and XOR information averaged across simultaneously imaged cells in each trial. The running mean (window of 100 trials) is shown with shading indicating mean ± s.e.m. Gray bar at the bottom indicates bins of sample cue information in which XOR information was higher on correct trials than on error trials ($p < 0.05$, bootstrap). The correct-error trial difference was significantly larger in RSC compared to V1 for 0.01–0.05 logLR (4 bins) of the sample cue information, and compared to MM for 0–0.02 logLR (2 bins) of the sample cue information ($p < 0.05$, bootstrap). V1: $n = 1476$ correct/307 error trials, RSC: $n = 1420$ correct/265 error trials, MM: $n = 1009$ correct/221 error trials. Source data are provided as a Source Data file.

information were most prominent in RSC, less prominent in V1, and surprisingly weak in some parts of PPC.

## Mixed representations predict choices in RSC, MM, and V1

If the mixed representations of sample cue information and test cue information are important for task performance, then we expect the XOR information to influence the mouse's choice. We therefore compared trials of the same trial type, except with the opposite choice (e.g., turning left vs. right on B/BW trials). That is, we compared correct and error trials. For trials with identical sensory cues, if cells have higher information about the reward direction (XOR) on correct trials than on error trials, then such an observation would support the notion that the information in those cells may guide accurate choices[81,82].

Strikingly, XOR information in RSC was markedly different between correct and error trials (Fig. 4b, c, f, g). The amount of XOR information per cell in RSC was 86% lower on error trials than on correct trials and was thus nearly absent when mice made errors (Fig. 4f). Importantly, mice appeared engaged in error trials because error trials were interspersed with correct trials and were completed by mice with similar timing compared to correct trials (Supplementary Fig. 1a–f). A similar difference was also present in MM and V1 neurons, suggesting that their XOR information could also be behaviorally relevant, but at a lesser magnitude than in RSC (Fig. 4g). These results therefore suggest XOR information in RSC, MM, and V1 may be used to guide the mouse's choice.

Remarkably, a substantially lower fraction of cells had mixed selectivity on error trials in RSC (Fig. 4h, i). Whereas correct trials had approximately equal numbers of cells with sample cue, mixed, and test cue selectivity, error trials had notably fewer cells with mixed

selectivity. These differences in the fraction of neurons with mixed selectivity were most prominent in RSC and also present in MM and V1 (Fig. 4i). Whereas mixed selectivity cells tended to be active on single trial types, they lost the single-trial-type selectivity on error trials especially in RSC and MM (Fig. 4j). Consequently, XOR information in mixed selectivity cells was greatly reduced on error trials (Fig. 4k). These trends can be seen in the example cells shown earlier. Some single-trial-type selective cells were less active on error trials (Fig. 2i) and thus had less XOR information when mice made incorrect choices (Fig. 4h, magenta points near the origin). Other cells were active on different trial types on correct versus error trials (Fig. 2j) and encoded the incorrect identity of the sample cue or test cue on error trials (Fig. 4h, magenta points in gray shading) and thus incorrect (negative) XOR information (Fig. 4i, gray shading ranges). Together, these observations support the idea that aberrant activity of mixed selectivity cells across a distributed set of areas, and most remarkably in RSC, may contribute to incorrect choices.

The lower XOR information on error trials appeared to arise in multiple ways. One possibility is a failure to receive sufficient sample cue information, such as due to fading memory during the delay segment. Indeed, the average sample cue information per cell was lower in RSC and MM in the delay segment on error trials (Fig. 4a, d). Moreover, in the test segment, sample cue information was more reduced on error trials compared to test cue information (Fig. 4d, e). Another possibility is a failure to mix sample cue and test cue information in the test segment to generate XOR information despite the presence of sufficient sample cue information. To test this possibility, on each trial, we computed the information per cell for the sample cue immediately prior to information mixing (i.e., the end of the delay) and for XOR in the test segment on the same

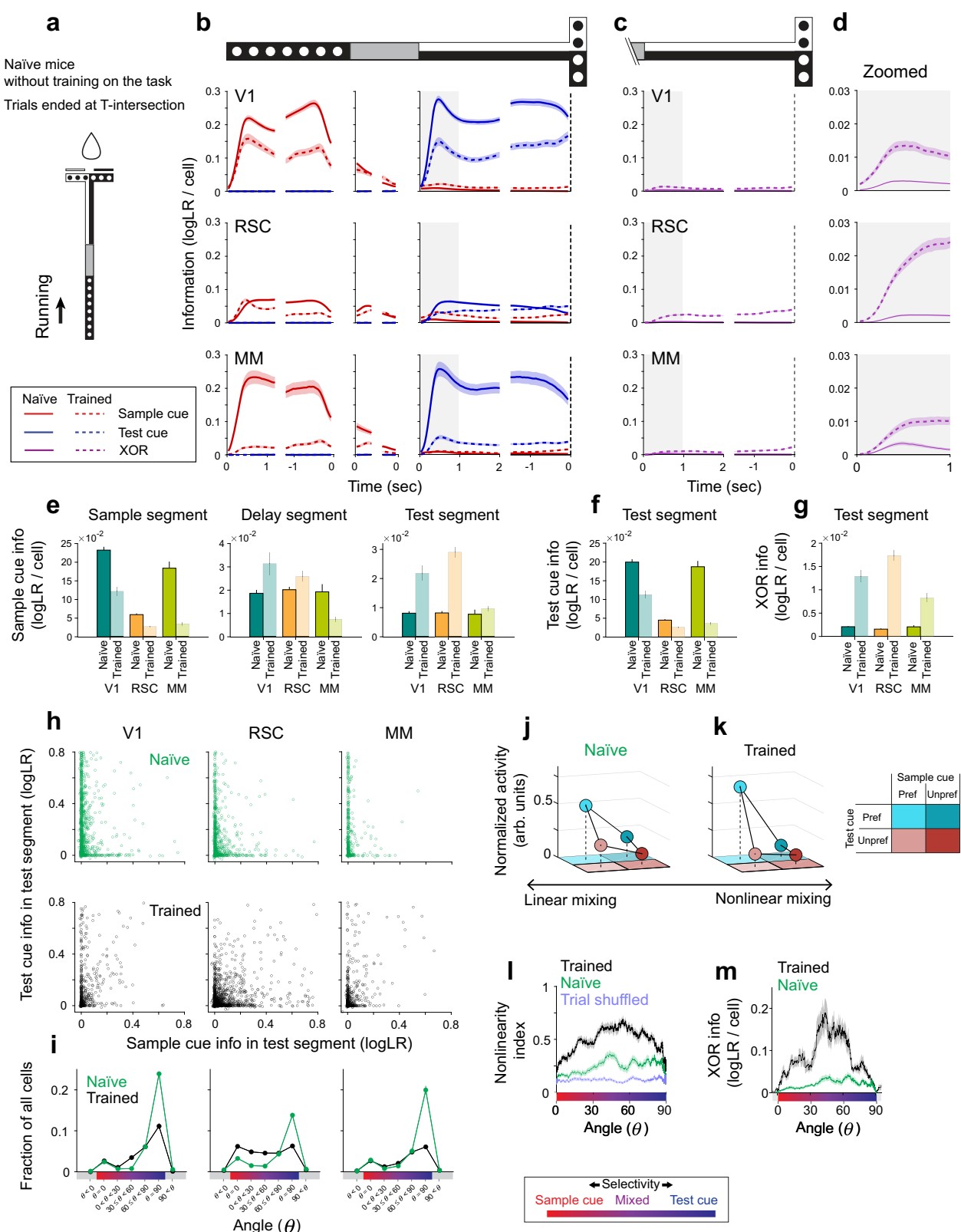

individual trials (Fig. 4l). For a given level of sample cue information, XOR information in RSC cells in the test segment was much reduced on error trials (compare black and magenta along a vertical slice of Fig. 4l). This result implies that mixing to produce XOR information was less effective on error trials. This difference was more prominent in RSC than in V1 and MM (Fig. 4l). These results suggest that incorrect choices in this task may arise from the fading of memory

signals as well as a failure to mix memory signals with current sensory signals, particularly in RSC.

It was surprising that the highest density of mixed selectivity cells was in RSC and that the largest difference between correct and error trials in mixed selectivity cells was in RSC, given that our optogenetics experiments showed smaller effects on behavioral performance when inhibiting RSC relative to V1 and PPC. However, our optogenetics

**Fig. 5 | Information profile in mice traversing the maze without task learning.**
**a** Schematic of the experiment with naïve mice without training on the delayed match-to-sample task. Naïve mice navigated through the virtual reality T-maze identical to the one used in the delayed match-to-sample task. When the mice reached the T-intersection, the trial ended with the delivery of a reward. **b** Sample cue and test cue information for naïve mice (solid curves), plotted similarly to Fig. 3a. Data from the trained mice in Fig. 3a are replotted for comparison (dashed curves). Gray regions indicate the period (first one second) analyzed for the test segment in panels (**e**–**m**). For naïve mice, V1: $n = 7015$ cells (99.9% of detected cells), RSC: $n = 5107$ cells (99.9%), MM: $n = 1582$ cells (99.9%) for panels (**b**–**i**). **c** Similar to panel (**b**), except for XOR information. **d** Zoomed view of XOR information from panel (**c**) for the first one second in the test segment. **e** Average sample cue information per cell for naïve mice in the sample, delay, and test segments (darker colors), plotted similarly to Fig. 3d. Data from the trained mice in Fig. 3d are replotted for comparison (lighter colors). The difference between naïve and trained mice was significant in the sample segment for V1, RSC, and MM ($p < 10^{-4}$), in the delay segment for V1 ($p = 0.0042$) and MM ($p < 10^{-4}$) but not for RSC ($p = 0.0226$), in the test segment for V1 and RSC ($p < 10^{-4}$) but not for MM ($p = 0.289$). The significance threshold was adjusted by Bonferroni correction with $\alpha = 0.05$ to account for 3 area-wise comparisons for panels (**e**–**g**). **f** Similar to panel (**e**) except for test cue information in the test segment. The difference between naïve and trained mice was significant for V1, RSC, and MM ($p < 10^{-4}$). **g** Similar to panel (**e**) except for XOR information in the test segment. The difference between naïve and trained mice was significant for V1, RSC, and MM ($p < 10^{-4}$). **h** For each cell in naïve mice (green circles), the sample cue information in the test segment and the test cue information in the test segment, plotted similarly to Fig. 3g. Data from the trained mice

in Fig. 3g are replotted for comparison (black circles). **i** Distribution of cells from panel (h) in discrete angle bins (green circles), plotted similarly to Fig. 3i. Data from the trained mice in Fig. 3i are replotted for comparison (black circles). Cells with noise-level information (magnitude $r < 0.01$) were not assigned angles but included in the total number of cells to calculate the fractions. The fraction of cells was significantly different between naïve and trained mice in the following bins ($p < 0.0002$); V1: $\theta < 0°$, $60° \leq \theta < 90°$, $\theta = 90°$, $90° < \theta$; RSC: $\theta = 0°$, $0° \leq \theta < 30°$, $30° \leq \theta < 60°$, $\theta = 90°$; MM: $\theta = 90°$. The significance threshold was adjusted by Bonferroni correction with $\alpha = 0.05$ to account for 7 bin-wise comparisons. Error bars indicate s.e.m. and are smaller than the data marker for some bins. **j** Normalized mean activity of mixed selectivity cells ($15° < \theta < 75°$ in panel (**h**)) in naïve mice for the four trial types, plotted similarly to Fig. 3j. The sum of mean activity across the four trial types in each cell was normalized to one. Nonlinearity Index for the naïve mice (NI = $0.27 \pm 0.01$, $n = 346$ cells; mean $\pm$ s.e.m.) was significantly smaller than that for the trained mice (NI = $0.54 \pm 0.01$, $n = 640$ cells; $p < 10^{-4}$). **k** Similar to panel (**j**), except for data from mixed selectivity cells in the trained mice (Fig. 3j) replotted for comparison. **l** Nonlinearity index for cells across angles (running average, window of 50 cells) in naïve mice (green trace) and trained mice (black trace). Cells with noise-level information (magnitude $r < 0.01$) were excluded. Blue trace shows chance nonlinearity index value computed with shuffled trial identities. Shading indicates mean $\pm$ s.e.m. $n = 4203$ cells. **m** XOR information for cells across angles for the naïve mice (green trace; running average, window of 50 cells), plotted similarly to Fig. 3n. Cells with noise-level information (magnitude $r < 0.01$) were excluded. Data from the trained mice in Fig. 3n are replotted for comparison (black trace). Shading indicates mean $\pm$ s.e.m. $n = 4203$ cells. Source data are provided as a Source Data file.

approach only inhibited relatively small cortical volumes. We therefore expanded our inhibition in RSC to three bilateral pairs of inhibition sites (orange circles in Fig. 1m). This expanded inhibition decreased the mouse's performance to near chance levels when RSC was inhibited throughout the trial ($55.8 \pm 4.0\%$ correct; mean $\pm$ s.e.m.) and resulted in a more substantial decrease in performance compared to the smaller inhibition sites (Fig. 1m, n). While the expanded RSC inhibition had the largest effect in the test segment, performance was also substantially decreased by inhibition in the sample segment, implying a possible role of RSC in encoding the sample cue identity on each trial. These pronounced inhibition effects with the expanded RSC inhibition suggest that RSC is essential for accurate performance in the flexible navigation decision task.

## Mixed selectivity cells emerged through learning of the flexible navigation decision task

If the mixed selectivity represents the combination of memory and visual information, we might expect that mixed selectivity emerges through learning of the flexible decision-making task as the mouse gets trained to maintain the memory and combine it with visual information. An alternative possibility is that the mixed selectivity reflects visual responses alone. For example, due to visual adaptation following the sample cue, the responses to the test cue could differ depending on the preceding sample cue. In this case, the mixed selectivity might exist even without learning and performing the task.

We imaged the same areas of posterior cortex in a separate cohort of "naïve" mice that were not trained on the delayed match-to-sample task but ran through the identical maze used in the task (Fig. 5a). Specifically, these mice ran through the sample segment, delay segment, and test segment and experienced the four combinations of sample cue and test cue. They received a reward at the end of the T-stem and were not trained to make left-right turns based on the cue identities. Thus, these mice experienced the identical cues but did not perform a decision-making task. Behaviorally, these mice ran through the maze similarly to the mice trained on the delayed match-to-sample task. They traversed the delay segment with comparable timing ($1.57 \pm 0.19$ s, mean $\pm$ s.d., n = 4 mice).

In these naïve mice, neural activity in all imaged areas contained markedly less XOR information per cell in the test segment

compared to the mice trained on the delayed match-to-sample task (Fig. 5c, d, g), despite high information about the visual cues (the sample cue in the sample segment and the test cue in the test segment; Fig. 5b, e, f). In the test segment, the distribution of information across individual cells in the naïve mice was biased towards test cue information and away from sample cue information in all the imaged areas, including RSC (Fig. 5h, i). Consistently, the fraction of mixed selectivity cells was much smaller in naïve mice than in the trained mice (Fig. 5i). For the rare mixed selectivity cells in the naïve mice, the type of mixing was different from those in the trained mice. Mixed selectivity cells in the naïve mice had activity less specific to single trial types and tended to have linear mixing of the sample cue and test cue (Fig. 5j–l; NI = $0.27 \pm 0.02$, mean $\pm$ s.e.m.). Consequently, these rare mixed selectivity cells in the naïve mice contained less XOR information per cell than the mixed selectivity cells in the trained mice (Fig. 5m).

These results rule out the idea that the mixed selectivity results predominantly from visual responses alone due to, for example, visual adaptation following the sample cue. Instead, the results suggest that mixed selectivity cells emerge through learning of the delayed match-to-sample task and acquire a nonlinear mixing of memory and visual signals.

## Efficient reward direction encoding in populations of mixed selectivity cells

Given the importance of mixed selectivity at the level of single cells, we further investigated if the mixed selectivity cells play a privileged role at the level of populations of neurons. In many cases, it is assumed that the downstream readout operates as a linear decoder. Given that the reward direction is determined by the nonlinear XOR combination of the sample cue and test cue, a linear decoder cannot read out the reward direction from a population of pure selectivity cells[50], which in our study consists of cells with only sample cue information and cells with only test cue information. In contrast, a linear decoder can read out the reward direction from cells that contain XOR information, which in our case are the nonlinear mixed selectivity cells. Thus, the mixed selectivity cells appear particularly important under common assumptions about the linearity of downstream decoding mechanisms in the brain.

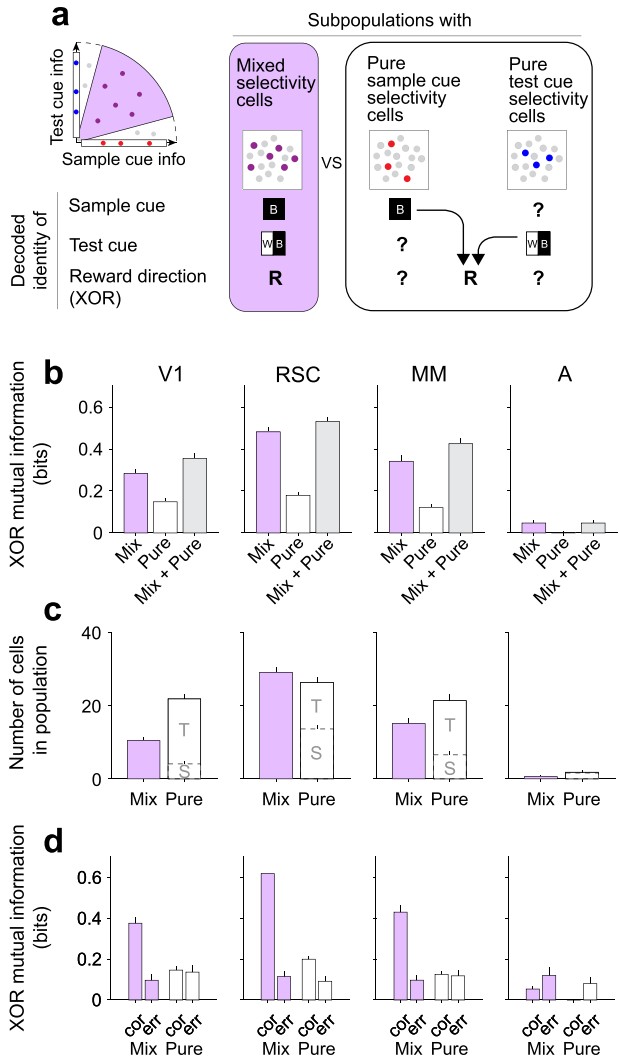

**Fig. 6 | Population code for XOR in populations of mixed selectivity or pure selectivity cells. a** Left: Partitions for mixed selectivity cells and pure selectivity cells based on their information. Right: Mixed selectivity cells encode the identity of the sample cue, test cue, and reward direction (XOR) (purple rectangle). Neither pure sample cue selectivity cells nor pure test cue selectivity cells encode XOR by themselves, but XOR can be nonlinearly decoded by combining the two types of pure selectivity cells (white rectangle). **b** Mutual information between the true and decoded XOR in populations of mixed selectivity cells (purple; $15° \leq \theta < 75°$ in Fig. 4h, correct trials), pure selectivity cells (white; $\theta = 0°$ and $\theta = 90°$ from Fig. 4i, correct trials), and both types of cells combined (gray). Error bars indicate mean ± s.e.m. V1: $n = 11$ sessions (1783 trials), RSC: $n = 12$ sessions (1685 trials), MM: $n = 7$ sessions (1230 trials), A: $n = 5$ sessions (822 trials). Mutual information in mixed selectivity cells was significantly greater than that in pure selectivity cells in V1 ($p = 0.0002$), RSC ($p < 10^{-4}$), MM ($p < 10^{-4}$), but not in A ($p = 0.049$). All $p$ values were calculated by bootstrap of cells in each session for panels (**b**–**d**) and the significance threshold was adjusted by Bonferroni correction with $\alpha = 0.05$ to account for 4 area-wise comparisons for panels (**b**–**d**), and 6 between-area comparisons for panel (**d**). **c** Number of cells classified as mixed selective or pure selective (T = test cue selective; S = sample cue selective) per session. Error bars indicate mean ± s.e.m. V1: $n = 11$ sessions, RSC: $n = 12$ sessions, MM: $n = 7$ sessions, A: $n = 5$ sessions. The number of cells was significantly larger for pure selectivity cells than for mixed selectivity cells in V1 ($p < 10^{-4}$) and MM ($p = 0.004$), but not in RSC ($p = 0.22$) and A ($p = 0.051$). **d** Mutual information between the true and decoded XOR, computed separately on correct and error trials. Error bars indicate mean ± s.e.m. V1: $n = 11$ sessions (1476 correct/307 error trials), RSC: $n = 12$ sessions (1420 correct/265 error trials), MM: $n = 7$ sessions (1009 correct/221 error trials), A: $n = 5$ sessions (753 correct/69 error trials). For mixed selectivity cells, the difference between correct and error was significantly different from zero in V1 ($p < 10^{-4}$), RSC ($p < 10^{-4}$), MM ($p < 10^{-4}$), but not in A ($p = 0.027$). For pure selectivity cells, the difference between correct and error was not significantly different from zero in V1 ($p = 0.95$), RSC ($p = 0.13$), MM ($p = 0.63$), and A ($p = 0.015$). The difference between correct and error in mixed selectivity cells was significantly larger than that for pure selectivity cells in V1, RSC, and MM ($p < 10^{-4}$), but not in A ($p = 0.90$). The difference between correct and error in mixed selectivity cells was significantly larger in RSC than in V1 ($p = 0.0001$), MM ($p = 0.0058$), or A ($p < 10^{-4}$). Source data are provided as a Source Data file.

However, the brain possesses mechanisms that could enable nonlinear decoding[83–85]. With a nonlinear decoder, a downstream network could read out the XOR identity by combining cells with pure selectivity, that is by nonlinearly combining cells with pure sample cue selectivity and cells with pure test cue selectivity (Fig. 6a). With the assumption of a nonlinear readout mechanism, we assessed whether populations of mixed selectivity cells were still more informative about the reward direction (XOR identity) than populations of pure selectivity cells, by comparing simultaneously imaged populations from a given imaging field-of-view (~0.5 mm² area). To evaluate the accuracy of the population code, we nonlinearly decoded XOR identity from the population and quantified the mutual information between the true and decoded XOR identity in units of bits[86]. Notably, this analysis evaluated information in a population as a whole, instead of averaging information across cells as was done in the previous analyses.

In RSC, MM, and V1, the mixed selectivity population represented XOR identity more accurately (higher mutual information) than the population of neighboring pure selectivity cells. The decoding from the mixed selectivity population was almost as accurate as the combined population of mixed and pure selectivity cells (Fig. 6b). Area A contained only little XOR information due to the lack of mixed selectivity and pure selectivity cells (Fig. 6b, c).

We wanted to understand what contributes to the difference in the accuracy of the population code between the mixed selectivity and pure selectivity populations. One possibility is that there is a significant difference in the number of cells that fall into these groups. However,

there were fewer, or at most equal, mixed selectivity cells relative to pure selectivity cells (Fig. 6c). Thus, the mixed selectivity populations efficiently encoded XOR because they contained more XOR information in a given number of cells. Second, it is possible that correlations in the activity between neurons in the population differed, such as noise correlations that create redundancy between neurons. However, in both the mixed selectivity and pure selectivity populations, the magnitude of noise correlations was similar (Supplementary Fig. 9a–c). Also, the difference in XOR information between these populations largely remained even after analytically disrupting noise correlations by shuffling trial labels independently for each neuron within a given trial type (Supplementary Fig. 9d). Together, these results indicate that the higher accuracy in the mixed selectivity population was not due to major differences in the population size or activity correlations. Instead, the mixed selectivity itself may be the key feature for an efficient code, which allows for a more accurate representation of the reward direction with a smaller number of neurons.

We tested if this efficiency was a general property of mixed selectivity populations. Many properties of a neural population can potentially contribute to a population code, including the number of cells, noise correlations, and signal-to-noise ratio of activity. Because it is difficult to control for and vary these properties in real data, we simulated neural activity and compared the decoding accuracy from either a mixed or pure selectivity population, while equalizing the properties between the two except for their selectivity (Fig. 7a). Interestingly, across all conditions, a mixed selectivity population showed an efficient XOR representation, in the sense that a mixed selectivity population encoded XOR more accurately than the same size of a pure selectivity population, although it was less informative about either the sample cue or the test cue (Fig. 7b–e). Furthermore, a

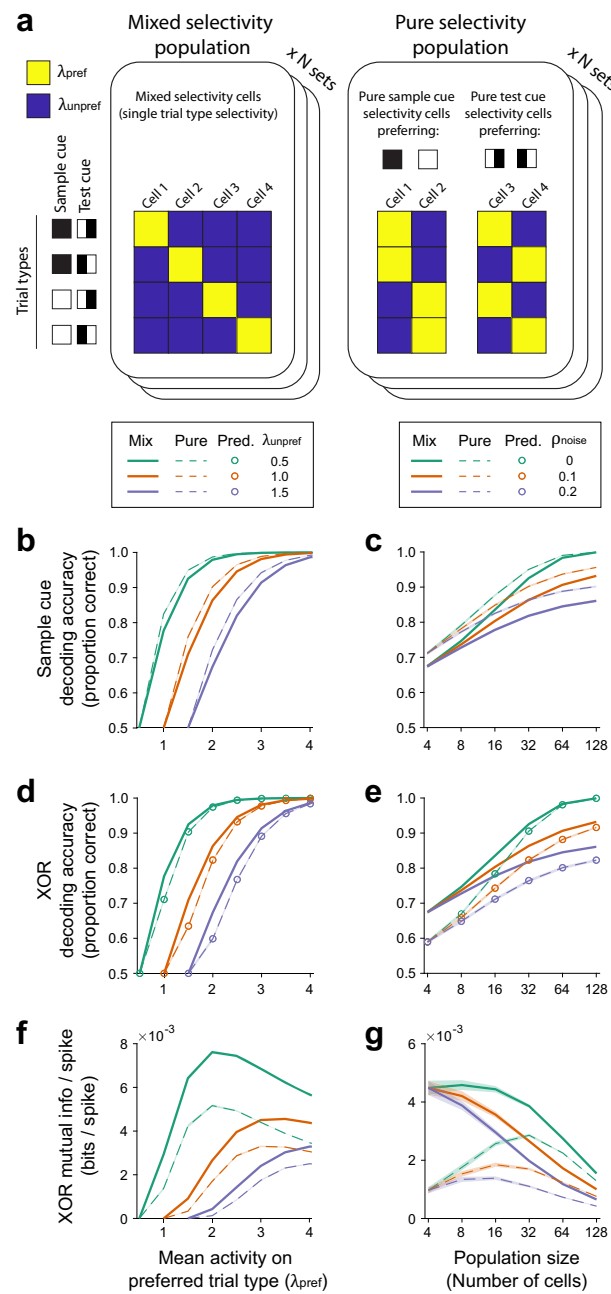

**Fig. 7 | Decoding accuracy from simulated mixed and pure selectivity populations. a** Schematic of simulated population. The number of spikes generated by each cell follows a Poisson distribution with the mean λ. Each cell responds to a preferred cue (or a preferred combination of cues) with the mean of $\lambda_{pref}$ and to an unpreferred cue with the mean of $\lambda_{unpref}$. The minimum unit population size is a set of four cells (rounded rectangle). Each cell preferentially responds to a specific trial type in the mixed selectivity population, or to a specific sample cue or test cue identity in the pure selectivity population. The population size increases by including N sets of the unit population with the same selectivity pattern. **b** Decoding accuracy for the sample cue (or test cue) in simulated population activity under a various combination of the mean activity ($\lambda_{pref}$ and $\lambda_{unpref}$). The SNR in the population increases with higher $\lambda_{pref}$ and lower $\lambda_{unpref}$ under Poisson noise. The population activity was simulated on 10,000 trials and repeated 10,000 times, separately in mixed selectivity population (solid lines) and pure selectivity population (dashed lines). Shading indicates mean ± s.e.m. and is equal or smaller than the line widths. The population activity was simulated with 8 cells and noise correlation ($\rho_{noise} = 0.1$) in panels (**b, d, f**). **c** Similar to panel (**b**), but under various combinations of the population size and noise correlation level. The SNR in the population increases with a larger population size and lower noise correlation. The population activity was simulated with $\lambda_{pref} = 2.0$ and $\lambda_{unpref} = 1.0$ in panels (**c, e, g**). **d** Similar to panel (**b**), but with the decoding accuracy for the reward direction (XOR). For a pure selectivity population, the decoding accuracy of XOR ($p_{XOR}$, dashed line in panels **d, e**) can be predicted by decoding accuracy for the sample cue (or test cue) ($p$, dashed line in panels **b, c**). Open circles show $\hat{p}_{XOR}$ predicted by $p^2 + (1-p)^2$: the sum of probabilities that both sample and test cues decoded correctly ($p^2$) or incorrectly ($(1-p)^2$). See Supplementary Fig. 10 for reasoning. **e** Similar to panel (**c**), but with the decoding accuracy for the reward direction (XOR). **f** Similar to panel (**d**), but with XOR mutual information divided by the total number of expected spikes in a population. **g** Similar to panel (**e**), but with XOR mutual information divided by the total number of expected spikes in a population. Source data are provided as a Source Data file.

make accurate choices (Fig. 6d). In contrast, the XOR information decoded from the populations of pure selectivity cells showed smaller differences between correct and error trials. Thus, the mixed selectivity populations may be used to guide the choice toward the reward direction, whereas the pure selectivity populations may be less critical. Notably, all the properties of mixed selectivity populations were present in RSC, V1, and MM (Fig. 6d), indicating that a distributed network of mixed selectivity cells could be important for flexible decisions. However, whereas V1 and MM had a lower proportion of mixed selectivity cells than pure selectivity cells, RSC had similar proportions of each type (Fig. 3i, Fig. 6c), resulting in the largest number of mixed selectivity cells in RSC (Fig. 6c). This difference in proportions of cells, together with the loss of XOR information in mixed selectivity cells on error trials (Fig. 4j, k), explains why RSC has the largest change in XOR information between correct and error trials when averaged across individual cells (Fig. 4f, g) or when quantified in its mixed selectivity population (Fig. 6d). For these analyses, similar results were present for a fixed neural population size (Supplementary Fig. 11). Together, these results suggest that a distributed network of mixed selectivity cells could be critical for flexible navigation decisions. These cells were surprisingly sparse in anterior PPC (area A) and densest in RSC, endowing RSC with the highest capacity to represent XOR information that could be read out to guide choices.

## Discussion
Our work demonstrates a visual-parietal-retrosplenial network plays a central role in the flexibility of decision-making during navigation. We first performed an optogenetics screen to find cortical areas necessary for the delayed match-to-sample task and then used calcium imaging in these areas to screen for the activity patterns related to accurate decision-making. This relatively unbiased approach allowed us to discover three key findings while providing evidence against several alternative outcomes. First, activity in the visual-parietal-retrosplenial network was necessary for accurate performance on the task, in

simulated mixed selectivity population was also energetically efficient for encoding XOR in our task because it conveyed more XOR information per spike compared to a pure selectivity population (Fig. 7f, g). Together, mixed selectivity appears to account for the higher efficiency of the population code.

This efficiency can be intuitively explained by the number of decisions required to decode XOR (Supplementary Fig. 10a). In a pure selectivity population, both the sample cue and test cue need to be encoded correctly for XOR to be decoded correctly from the population (Supplementary Fig. 10c). In contrast, XOR can be directly decoded from a mixed selectivity population, requiring only a single variable to be encoded correctly (Supplementary Fig. 10b). Thus, a mixed selectivity population enables more accurate decoding of XOR for a given condition even with a nonlinear decoding mechanism.

Finally, in imaged populations, XOR information was higher on correct trials than on error trials in mixed selectivity populations, supporting the notion that when mixing is reduced, the mouse did not

particular during the test segment when the memory information must be combined with the visual information to make a left-right decision. Neurons representing the XOR information that indicated the reward location were distributed across posterior cortex in a graded manner. These results argue against alternate possible representations, including the mixed selectivity representation being confined to one specific area or distributed uniformly across the entire cortex. Second, within this network, we identified neurons that nonlinearly mixed short-term memory with visual information in the form of single-trial-type selectivity. This nonlinear mixed selectivity appeared useful for guiding navigation choices because it formed an efficient code that accurately represented the reward direction (XOR) in a relatively small number of cells and could be read out easily with a linear decoder. Surprisingly, neurons with pure choice (or XOR) selectivity were rare in the cortical areas we imaged. The choice signal therefore did not appear to be constructed directly from cells with pure memory and visual signals but rather from the mixed selectivity cells. The accurate XOR encoding of mixed selectivity cells compared to pure selectivity cells makes it unlikely that only pure selectivity cells were used to guide the choice direction. Furthermore, the mixed selectivity emerged with learning of the flexible navigation decision task, ruling out that it merely reflected intrinsic visual responses or adaptation to the sequential cues. Third, these mixed selectivity cells lost their selectivity and had lower XOR information when the mouse made incorrect choices. We therefore speculate that they may be important for guiding the mouse's choices and play a critical role in the flexibility of navigation decisions. We note, however, that the correlation between mixed selectivity and the accuracy of a mouse's choices is only suggestive of this relationship and that direct manipulation of mixed selectivity cells will be necessary to establish their causal relationship to flexible decision-making.

To evaluate the general utility of mixed selectivity neurons for flexible decision-making, we need to consider broader conditions outside of our task. The cells with mixed selectivity for memory and visual signals had activity mostly on a single trial type. These cells therefore represented a specific combination of a sample cue and a test cue. An alternative possibility would have been cells that represent the XOR combination of the two cues and become active mostly on two of the four trial types. This alternative type of cells would signify the abstract notion of "match" and provide a generic solution to a match-to-sample-type task because they can be re-used for all new combinations of cues, potentially allowing for generalization across different cues. Instead, the single-trial-type selectivity we found functions as a basis set or look-up table for the specific learned combinations of cues. While it is not clear if this type of neural solution to the task affords the ability to generalize over a large set of cues, or novel cues, it may be advantageous for conditions that are utilized frequently as in the trials of our task. Furthermore, this type of solution may be especially advantageous for spatial navigation, where the sequential combination of cues defines a different spatial environment. In this case, an abstract 'match'-type signal may be less sensible behaviorally than a representation that specifies each environment separately, given that individual environments are units of ethological importance. Also, this type of mixed selectivity, in addition to the general advantages of nonlinear mixing[49,50], could be robust against noise[87] and energetically efficient[88]. Indeed, we showed how this type of code is efficient for the learned combinations of cues, compared to utilizing populations of pure selectivity cells, but we note that it may be energetically costly for forming separate representations for many possible combinations.

While the mixed selectivity cells were distributed across many areas of posterior cortex, RSC showed an enrichment of these cells, the most equal mixing of memory and visual information, the highest XOR information, and the largest differences in encoding of reward direction between correct and error trials. The discovery of RSC's involvement in flexible navigation decisions was surprising to us because studies of decision-making have often highlighted the importance of other cortical areas, such as frontal and parietal cortices[3,37-39,64,78,89-94]. In contrast, RSC is widely viewed as important for encoding current navigational variables[23-25,27,29,30,95-97]. RSC has also been shown to represent a wide range of internal signals, including spatial memory[46-48], time[98,99], and value[100,101], and is thought to play a role in fixed sensorimotor associations in decision-making tasks[41,42,44]. However, our work extends further by showing how memory and sensory signals in RSC can be combined to mediate flexible navigation decisions. We propose that a general function of RSC is to combine internal signals with current sensory signals by forming a mixed representation that can flexibly guide navigation actions. Thus, the mixed selectivity observed in our study could be a specific case of a more general function of RSC, and our results highlight RSC as a candidate area to investigate flexibility of decision-making during navigation.

Despite the enrichment of mixed representations in RSC, further investigation will be required to understand mechanisms that generate these representations. It is possible that RSC receives inputs that separately contain visual and memory information, as is consistent with the dense axonal projections it receives from visual cortices and the hippocampal formation[102-105]. In this case, RSC neurons might play an active role in mixing these streams of information. Alternatively, the mixed representations might arise through computations in a distributed network spanning multiple areas, including V1, which is consistent with the presence of mixed selectivity neurons throughout posterior cortex in our experiments. Future studies aimed at characterizing and manipulating the inputs to RSC will help to clarify whether RSC plays a causal role in mixing memory and visual information.

Mixed selectivity cells existed throughout posterior cortex in a graded manner, consistent with a representational principle that each area encodes all or most of the essential task variables but with key distinctions in the magnitude of encoding of those variables in each area[69,73]. An appreciable fraction of V1 neurons also showed similar mixed selectivity even though in the population of V1 neurons, the current visual cue was more strongly encoded than the memory. The mixed selectivity cells in V1 had similar properties to those in RSC. This finding adds to a growing list of functions for V1 during navigation, including representing running velocity[106], head angular velocity[107], spatial position[31], and reward-related activity[108]. Surprisingly, PPC had the lowest density of mixed selectivity cells in the population. Area MM had an activity profile intermediate between RSC and V1. The activity in area A was mostly related to the locomotion of the mouse. Together, the neural representation gradually shifted from mixed selectivity in RSC to locomotor selectivity in the anterior part of PPC (area A). This spatial gradient appears to overlap with the spatial gradient going from allocentric representations in RSC to egocentric representations in PPC[20]. This overlap might reflect a relationship between decision-making and navigation. Allocentric representations often rely on memory (e.g., past landmarks) when the current sensory cues do not fully identify the animal's location. If we regard each trial type as a different maze environment, the single-trial-type selectivity can be interpreted as an allocentric representation to identify the mouse's location (maze environment) by using the current sensory cue (test cue) and the memory of the past landmark (sample cue). By contrast, the locomotor-related activity can be interpreted as an egocentric representation that encodes running with respect to the mouse's heading direction. Therefore, along the spatial gradient from RSC to PPC, the conversion of the reference frames in navigation might share the same mechanism that converts the mixture of memory and visual information into locomotor selectivity during flexible navigation decisions.

Our optogenetics experiments did not find evidence in support of the involvement of frontal areas in our task. However, these experiments do not rule out the involvement of frontal areas in our task and related tasks, and we did not focus on identifying a role for these areas. Future work will be needed to understand how frontal cortices contribute to flexible navigation decisions. In contrast to our findings, flexible decision-making with different sensory-motor modalities identified the premotor area ALM as critical for an olfactory task with tongue-licking as choice reports[3] and prelimbic cortex and superior colliculus as important for a visual task with head-orienting to report choices[15]. It is sensible that posterior cortical areas would be important for a visual and navigation task, whereas frontal or tectal regions would be involved in tasks requiring licking or head-orienting. In line with this idea, when mice oriented their heading direction as they accumulated evidence in a decision-making task during virtual navigation, frontal cortex was more involved in addition to posterior cortex[43]. Here, we prevented the mouse from changing its heading direction in real and virtual space, which might contribute to why we observed less involvement of frontal areas. Taking all these studies together, for flexible decision-making, we speculate that similar computations can be performed in distinct cortical areas that are critical for the specific sensory processing and motor execution involved in the task, but this idea needs to be tested in future studies.

We demonstrated the importance of nonlinear mixing for accurate behavior in a real, large-scale, simultaneously recorded population, instead of inferring their importance from artificially created populations of non-simultaneously recorded neurons. The major differences between correct and error trials in populations of mixed selectivity cells, especially relative to pure selectivity cells, provide the suggestion that mixed selectivity has a causal role in flexible decisions. Now that we have identified these putative decision-related signals, as experimental approaches become readily available to manipulate the activity of specific cells based on their encoding properties[109–114], it will be of great interest to test directly the causal role of mixed selectivity cells for flexible navigation decisions.

## Methods

### Animals
All experimental procedures were approved by the Harvard Medical School Institutional Animal Care and Use Committee and were performed in compliance with the Guide for the Care and Use of Laboratory Animals. Eight male C57BL/6J mice (Jackson Laboratory, Strain No. 000664) and two male C57BL/6J-Tg(Thy1-GCaMP6s) GP4.3Dkim/J mice (Jackson Laboratory, Strain No. 024275) were used for the calcium imaging experiments, and twelve male VGAT-ChR2-EYFP mice (eleven mice from Jackson Laboratory, Strain No. 014548; one mouse with VGAT-IRES-Cre/+; Ai32/+) were used for the optogenetics experiments. Most of the mice were 8-16 weeks old at the start of behavioral training. The mice used for the expanded inhibition of RSC (Fig. 1m, n) tended to be on the upper side of the age range, potentially contributing to their faster running speeds (Supplementary Fig. 2j–o). Mice were housed on a reverse 12 h dark/light cycle and in pairs of littermates. Mouse health was evaluated daily.

### Surgical procedures
**Calcium imaging surgery.** For calcium imaging experiments, prior to behavioral training, mice were implanted with a titanium implant that was designed for imaging procedures. The headplate had an opening in the middle that allowed subsequent placement of a cranial imaging window. The location of the headplate opening was centered at 2.0 mm lateral and 2.5 mm posterior relative to bregma to allow imaging in V1, RSC, and PPC and was centered at 1.2 mm lateral and 1.2 mm anterior relative to bregma for imaging in M1 and M2. The headplate was affixed to the skull by dental cement (Radiopaque Metabond, Parkell) mixed with India ink (5% vol/vol) to increase light shielding for

imaging. To allow for an imaging plane parallel to the surface of cortex, the headplate was tilted by 15° in roll relative to the mouse's body axis. In the initial surgery prior to behavioral training, the planned center coordinates for the cranial window were marked by a drill and ink on the skull and covered by transparent dental acrylic (Ortho-Jet, Lang Dental), so that the coordinates could be recovered in a subsequent cranial window surgery.

A cranial window surgery was performed when the mouse had achieved high performance on the delayed match-to-sample task (greater than 80% correct). The cranial window was made centered at the coordinates marked during the initial implantation of the headplate (2.0 mm lateral and 2.5 mm posterior to bregma for posterior imaging; 1.2 mm lateral and 1.2 mm anterior to bregma for anterior imaging). A window for posterior imaging was constructed by bonding two 3.5 or 4.0 mm diameter coverslips to each other and to an outer 4.0 or 5.0-mm diameter coverslip (#1 thickness, Warner Instruments). A window for anterior imaging was constructed by bonding two laser-cut 2.0 × 2.5 mm oval coverslips to each other and to an outer 3.0 mm diameter coverslip (#1 thickness, Warner Instruments). Coverslips were bonded to each other by UV-curable optical adhesive (Norland Optics NOA 65 or 68).

AAV2/1-synapsin-GCaMP6s-WPRE-SV40 virus (University of Pennsylvania Vector Core Facility, Catalog No. AV-1-PV2824 or Addgene, Catalog No. 100843-AAV1) was diluted to ~0.5–1.0 × 10^13 gc/mL in phosphate-buffered saline. Injections were made using a glass pipette and custom air-pressure injection system. At each site, ~50–70 nL was injected 250–300 μm and 450–500 μm below the dura over 1–2 min. After each injection, the pipette was left in place for an additional 1–2 min. For the posterior cranial window, a glass pipette was inserted at a 30° angle relative to the brain surface, and virus was injected at three sites in V1 (2.2 mm lateral, 3.5 mm posterior; 2.8 mm lateral, 3.5 mm posterior; 2.8 mm lateral, 3.0 mm posterior to bregma), two sites in RSC (0.5 mm lateral, 2.15 mm posterior; 0.5 mm lateral, 2.85 mm posterior to bregma), and three sites for PPC (1.2 mm lateral, 1.75 mm posterior; 1.8 mm lateral, 2.0 mm posterior; 1.2 or 2.4 mm lateral, 2.25 mm posterior to bregma). For the anterior cranial window, a glass pipette was inserted perpendicular to the brain surface, and virus was injected at ten sites across M1/M2 areas (nine sites in a 3 × 3 grid pattern centered at 1.2 mm lateral, 1.2 mm anterior to bregma with 0.5 mm spacing, and one site near ALM at 1.2 mm lateral, 2.2 mm anterior to bregma). When the planned coordinates overlapped with blood vessels, injections were shifted by 50-100 μm to avoid rupturing the vessels.

The window was sealed using dental cement (Radiopaque Metabond, Parkell) mixed with India ink (5% vol/vol). To prevent light contamination and create a water bath for imaging, an aluminum ring was affixed to the top of the headplate with dental cement mixed with India ink. Experiments started after ~2 weeks from viral injection and typically continued for 6–8 weeks. Experiments were terminated when GCaMP6s expression appeared high, with some neurons exhibiting GCaMP6s in the nucleus.

**Optogenetics surgery.** For optogenetics experiments, mice were implanted with a clear skull cap, following procedures described previously[64,69]. The cranial surface was exposed, and the headplate was affixed to the interparietal bone (behind the lambda suture) by transparent dental cement (Clear Metabond, Parkell). The remaining cranial surface was covered by a thin layer of cyanoacrylate (Insta-Cure, Bob Smith Industries) and reinforced by ~1 mm layer of transparent dental acrylic (Ortho-Jet, Lang Dental) on top. After a mouse was trained, the acrylic surface was polished by a polishing drill (Model 6100, Vogue Professional) with denture polishing bits (HP0412, AZDENT) and coated by a thin layer of clear nail polish (Electron Microscopy Sciences, 72180, or OPI). For light shielding, dental cement mixed with India ink or carbon (Sigma-Aldrich) was used to affix an aluminum ring

to the top of the headplate and to fill the gaps between the aluminum ring and the circumference of the acrylic implant. Fiducial marks were made on the aluminum ring to aid laser alignment.

## Behavior

**Virtual reality system.** The virtual reality system was built based on previous designs[37,38]. Virtual reality environments were constructed and operated using MATLAB-based ViRMEn software (Virtual Reality Mouse Engine)[56]. A projector (PicoPro or PicoBit, Celluon Inc., Laser Beam Pro, KDC) was used to display the virtual environment onto the back side of a 24-inch diameter half-cylindrical screen. The virtual environment was updated in response to the mouse's locomotion on an open cell Styrofoam spherical treadmill (8-inch diameter, ~135 g). Two optical sensors (ADNS-9800, Avago Technologies) positioned 45° below the equator of the ball and separated by 90° in azimuth were connected to a Teensy-3.2 microcontroller (PJRC.COM) to measure the three-dimensional rotation velocity of the ball (roll, pitch, and yaw).

**Delayed match-to-sample task.** Mice performed the navigation-based delayed match-to-sample task in a virtual reality T-maze (Fig. 1a, Supplementary Fig. 12d). The mouse traversed three segments: sample, delay, and test in the T-maze. In each segment, the mouse spent a variable duration from trial to trial depending on the maze traversal speed. In the sample segment, either black (B) or white (W) patterns on the wall provided the sample cue. In the delay segment, the walls had a gray pattern in all trials. The test segment provided the test cue, either white patterns on the left side and black patterns on the right side (WB) or vice versa (BW). The test cue was not revealed (and not visible) until the mouse reached a defined spatial position (240 cm away from the maze start). At the T-intersection, the mouse was required to make a right or left choice by turning to the T-arm whose color matched the sample cue. When the mouse ran 20 cm into the T-arm, the trial ended. The correct choice was rewarded by 4-5 μL of 10x diluted sweetened condensed milk (Eagle Brand) or 30 mM acesulfame solution (Prescribed For Life). Incorrect choices were not rewarded, and consecutive error trials were penalized by prolonged inter-trial intervals. After the intertrial interval (at least 4 s), the next trial started with the mouse location reset to the beginning of the maze. The sample and test cues were randomly selected on each trial and were thus independent of one another.

The length of the T-stem was 400 cm, of which the last 160 cm was allocated to the test segment. The delay segment ranged from 20 to 100 cm in length and varied based on the behavioral performance (see below). The remaining maze length was allocated to the sample segment (140–220 cm).

In order to control the visual scene, movement in the virtual environment was constrained. The mouse's heading angle was fixed straight throughout a trial (Fig. 1d, top). In the T-stem, only forward/backward movements were allowed in the T-stem, and lateral movements were not allowed; the pitch velocity of the spherical treadmill was translated into forward/backward movement, and the roll and yaw velocities did not affect movement in the virtual environment (Fig. 1c, d). Backward movement was not observed in any mice. In the T-arms, only lateral movements were allowed; the roll velocity was translated into lateral movements, and the pitch and yaw velocities did not affect the movement (Fig. 1c, d).

The black and white walls for the sample cue and test cue had dot patterns (white dots on a black wall and black dots on a white wall). The gray wall for the delay segment had a striated pattern to create optic flow (Fig. 1a).

**Behavioral training.** Starting five days before the first training session, mice were put on a water restriction schedule that limited their total consumption to 1 mL per day. The weight of each mouse was monitored daily, and additional water was given if the mouse's weight fell below 80% of the pre-training weight. Mice were trained daily for 45–60 min at approximately the same time each day. Mice learned the task through four phases of training, as described below (Supplementary Fig. 12). Approximately 25-50% of mice learned to perform the delayed match-to-sample task with high accuracy (greater than 80% correct). To increase the statistical power for the analyses of error trials, we included more sessions with moderate accuracy (~80%) to analyze the imaging experiments, compared to the optogenetic experiments.

**Phase 1: Linear track.** In the first phase, mice were trained to run straight on the spherical treadmill in a linear track. Mice received a reward when they reached a gray tower presented at the end of the linear track. Pitch and roll velocity were translated to the forward speed and rotation of heading angle, respectively. The heading angle was not fixed at this stage, so that mice were required to correct their running trajectory to a straight path; this appeared to help produce good control of the spherical treadmill. For initial trials, the track length was short to increase the frequency of reinforcement. The track length was gradually extended if mice reached a tower within 30 s. Typically, mice learned to run along a ~300 cm track within 3–5 days of training.

**Phase 2: Single tower T-maze.** In the second phase, mice were trained to make visually guided turns to the right or left in a T-maze. The length of the T-stem was set to 200 cm, and a single gray tower was presented either behind the right or left T-arm throughout the trial. Mice received a reward when they reached the gray tower by turning towards the tower at the T-intersection. To facilitate training for the following phases, the T-stem had either a black or white wall pattern (later used as the sample cue), whereas T-arms had the black pattern on one arm and white pattern on the other (later used as the test cue). The gray tower was located behind the arm that matched the color to the initial cue. While mice could make turns based only on the location of the gray tower, the wall patterns acclimated mice for the next phase of training.

**Phase 3: Two tower T-maze.** In the third phase, mice learned associations between a combination of wall patterns and turning directions (choices). The maze structure was the same as in the second phase, except now we interleaved trials with a single tower (identical to Phase 2 trials) with trials with two towers, in which a tower was located behind both T-arms. For two tower trials, mice were no longer able to rely on the tower location to decide the turning direction and instead were required to use the wall patterns to make choices following the match-to-sample rule. The fraction of two-tower trials was dynamically adjusted based on the performance criteria described below; the fraction was increased or decreased with a step of 10%. When the fraction of two-tower trials reached 100% and the behavioral performance was consistently above 90% accuracy, the mice were advanced to Phase 4, the final phase of training.

**Phase 4: Delayed match-to-sample task with variable delay.** In this final phase, mice learned to make a choice in a longer maze (400 cm) based on the combination of the sample and test cues, except now with a delay segment added. With a delay segment, the sample cue disappeared at the onset of the delay, and the test cue appeared instantaneously at the offset of the delay. The test cue was not visible by looking down the T-stem; rather, we made the test cue appear when the mouse reached an exact spatial location (240 cm away from the maze start). As the mouse's performance improved, we gradually lengthened the delay segment (Supplementary Fig. 12d). When the delay segment was lengthened, the sample segment was shortened to keep the overall maze length constant. For our analyses, we included trials with a delay length of at least 20 cm.

Because this task was challenging for the mouse, we found it was helpful to include "crutch trials" that were easy for the mouse. In randomly selected 10% of trials, we omitted the delay segment (Supplementary Fig. 12e). These trials had a maze structure similar to the two tower T-maze in Phase 3. Crutch trials were not analyzed in the imaging and optogenetics experiments.

**Adjustment of task difficulty.** The task difficulty was adjusted by the length of the delay segment (Phase 4) or the fraction of two-tower trials (Phase 3). We adjusted the difficulty level based on the behavioral performance, statistically evaluated by a sequential probability ratio test[115,116]. When this statistical test decided that the mouse's performance was greater than 80% correct or at the chance level (50%), the task difficulty was increased or decreased, respectively. We set the upper and lower bounds of difficulty, so the level was adjusted within a limited range. Performance was evaluated using all trials from the last statistical decision up to a current trial in a session. For well-trained mice, the task difficulty typically remained stationary throughout a session after initial increases at the start of the session.

**Experiments with naïve mice.** For the imaging experiments with naïve mice (Fig. 5), mice were trained on the linear track (Phase 1) but skipped the subsequent training in Phases 2–4. During imaging sessions, these naïve mice navigated through an identical maze to the one used for the delayed match-to-sample task but trials ended when the mice reached the T-intersection without turning into a T-arm. If the mouse completed a trial within 30 s, the trial was labeled as a correct trial and the mouse received a reward at the end of trials. In a small fraction of sessions (8 out of 55 sessions), the forward speed in the virtual reality was decoupled from the mouse's running velocity (i.e., open-loop condition). In these sessions, a computer randomly chose a constant forward velocity that is comparable to the mouse's running speed. The results were qualitatively similar with or without these sessions.

## Optogenetics experiments

The photoinhibition system was built based on a previous design[69]. A 470 nm laser (LRD-0470-PFR-00200, Laserglow Technologies) was passed through galvanometric scan mirrors (6210H, Cambridge Technology) and focused using an achromatic doublet lens ($f$ = 300 or 400 mm, AC508-300-A-ML, AC508-400-A-ML, Thorlabs). The laser (analog power modulation, off to 95% power rise time, 50 ms) and mirrors (<5 ms step time for steps up to 20 mm) allowed simultaneous inhibition of bilateral sites by rapidly alternating the inhibition sites by moving the mirrors while the laser was turned off. The focused laser had a top-hat profile with a diameter of approximately 200 μm.

Optogenetics experiments were performed on VGAT-ChR2 mice, which express ChR2 in all GABAergic neuron types[65,117]. In each hemisphere, 28 inhibition target sites were arranged in a grid with 1 mm spacing, covering from 0.5 to 3.5 mm in the medial-lateral direction and from +3.0 mm to −4.0 mm in the anterior-posterior direction with respect to bregma (Fig. 1g). For three mice, inhibition was always applied throughout a whole trial, and for four mice, inhibition was applied throughout a whole trial or during a specific trial segment (sample, delay, or test segment). In a separate four mice, including one VGAT-IRES-Cre/+; Ai32/+ mouse, inhibition was expanded to three bilateral pairs of sites in RSC (centered at 0.5 mm lateral, 1.0, 2.0, 3.0 mm posterior to bregma) in all segments or in a specific trial segment (sample, delay, or test segment).

The inhibition was applied in randomly selected and interleaved 10–40% of trials. The fraction of inhibition trials was adjusted to maintain behavioral performance above ~80% correct across all trials. In each inhibition trial, the combination of bilateral inhibition sites and the inhibition segment were selected randomly without replacement. When all combinations were selected, the process was reinitialized and repeated.

The laser power at each inhibition site had a near-sinusoidal temporal profile at 40 Hz and time-averaged power of 2–5 mW. For the expanded inhibition in RSC, the total laser power across the three sites on each hemisphere was ~5–10 mW for three mice and ~5 mW for one VGAT-IRES-Cre/+; Ai32/+ mouse. To ensure inhibition prior to the mouse seeing the sample cue, the laser was turned on 1 s before the trial onset for the inhibition throughout the whole trial and for inhibition during the sample segment. For a similar reason for the test cue, the laser was turned on 20 cm before the test segment for inhibition during the test segment. The laser was turned off at the trial end (i.e., onset of the inter-trial interval).

**Electrophysiology.** To measure the spatial extent of optogenetic inhibition across cortical areas, extracellular recordings were conducted in one VGAT-ChR2-YFP mouse (Supplementary Fig. 2). A small craniotomy (diameter of ~1 mm) was created above the region of interest in the left hemisphere one day before the recording sessions. A 32-channel silicon probe (A1x32-Poly2-5mm-50s-177, NeuroNexus) was angled at 55-60° from the cortical surface at the entry point to avoid blocking the laser oriented perpendicularly to the cranial surface. The tip of the probe was advanced down to 1 mm below the dura for PPC and RSC recordings, and to 1.7 mm below the dura for M2/ACC recordings, based on manipulator readings. The signals were recorded at 20 kHz by the Intan RHD2000 Evaluation System and highpass-filtered at 250 Hz offline. Multi-unit activity was measured offline by counting the number of spikes that crossed a threshold, which was adjusted to include most of the local spikes in the absence of optogenetic inhibition.

The number of spikes was counted across recording sites during the first 5 s of a laser-on period, and then normalized to the spike counts during 5 s of a laser-off period immediately before the laser onset. Trials with less than 5 spike counts in the laser-off period were excluded. Recordings between 600 and 700 μm below the cortical surface were used for the evaluation of effects as a function of the horizontal distance from the laser center (Supplementary Fig. 2d–f). Recordings within 500 μm horizontal distance from the laser center were used for the evaluation across the cortical depth (Supplementary Fig. 2g–i).

## Widefield retinotopic mapping

To locate the imaged areas with respect to visual areas, a retinotopic map was obtained from the mice that were used for imaging of posterior cortical areas (V1, RSC, MM, and A): two of four mice trained on the delayed match-to-sample task and four naïve mice without learning the task. The procedures were similar to those described previously[45].

Widefield images were collected by a custom-built epifluorescence macroscope[118]. Excitation light was provided by a blue LED (455 nm LED, Thorlabs; bandpass filtered at 469 nm with 35 nm bandwidth, Thorlabs), focused on and reflected by a small mirror (8 mm Diameter 45° Rod Mirror, Edmund Optics). The reflected diverging light passed through a camera lens (NIKKOR AI-S FX 50mmf/1.2, Nikon) and illuminated the cortical surface. Emission light was collimated through the same camera lens, bandpass filtered (525 nm with 39 nm bandwidth, Thorlabs), and imaged by a second lens (SY85MAE-N 85 mm F1.4, Samyang) onto a CMOS camera (ace acA1920-155um, Basler).

Mice with expression of GCaMP6s were lightly anesthetized by isoflurane on a platform with their head angled at 30˚ with respect to a computer display (MG279Q, Asus), so that their right eye was positioned in front of the display. The visual stimulus was designed based on a previous study[119]. A spherically corrected black and white checkered moving bar (5 or 7 deg in width) swept vertically and horizontally in four directions across the display with the speed of 7 deg/s. Each sweep direction was repeated six times during a block, and four or six blocks were presented in a session. Images were collected at 60 Hz,

synchronized with the presentation of the visual stimulus on the display. The retinotopy was inferred from the spatial phase (location) of the stimulus that excited cells at the sweeping frequency[120]. Images were averaged across repeats for a given stimulus location (discretized in 2 deg bins) and smoothed by a Gaussian filter (s.d. = 25 μm). For each pixel., the spatial phase of the fluorescence signal was computed by Fourier transform, separately for the vertical and horizontal sweeps. The gradient of the spatial phase was used to compute the field sign map[121]. The retinotopic maps were overlaid on the maps of area boundaries based on the Allen Mouse Brain Common Coordinate Framework[71], by aligning the border between V1, PM, and AM (Fig. 2a, bottom)[122]. The cranial window contained the anterior medial part of V1 in the left hemisphere, where a large fraction of neurons had receptive fields in the lower right monocular visual field. This part of visual field largely overlapped with the right half of the test cue in the maze used for the delayed match-to-sample task.

## Two-photon calcium imaging

Imaging data were collected by a custom-built two-photon microscope. The microscope consisted of a resonant scanning mirror and a galvanometric mirror, separated by a scan lens relay telescope. Excitation light was delivered from Ti:sapphire laser (Coherent). The average laser power at the sample was ~50 mW. The emission light was separated into green and red channels by a dichroic mirror (580 nm long-pass, Semrock) and bandpass filters (525/50 and 641/75 nm, Semrock), and collected by GaAsP photomultiplier tubes (Hamamatsu). Signals were recorded only from the green channel for the analyses. Fluorescence light collection optics were housed in an aluminum enclosure to avoid the interference of visible light.

The behavioral setup (the mouse, spherical treadmill, head fixation system, and reward delivery system) was mounted on a 12″ x 12″ breadboard (Thorlabs) on a XYZ translation stage (Dover Motion).

A Nikon ×16, 0.8 NA objective lens was used to collect images from a ~700 μm × ~700 μm plane. Scanimage (Vidrio Technologies) was used to acquire 512 × 512 pixel images at 30 Hz. The imaging location and depth were varied daily to sample data from a wide range of areas and depths. The image plane was chosen between 100 μm and 400 μm below the dura, with the majority of planes between 100 μm and 200 μm (50/56 planes). 100,000 frames were recorded in each imaging session over ~1 h. To align timestamps between the image acquisition and the virtual reality software, clock pulses were output from ScanImage at each frame and from ViRMEn at each refresh of the virtual environment, and saved together in a separate computer.

## Image and signal processing

Acquired images were motion corrected in hierarchical steps[45]. First, "line-shift correction" was applied to align line-by-line alternating offsets in images caused by bidirectional scanning. Then, "sample movement correction" was applied to eliminate between-frame movement artifacts (rigid transformation) by FFT-based 2d cross-correlation[123], and to mitigate within-frame movement artifacts (non-rigid transformation) by Lucas-Kanade method[124]. Finally, "sample deformation correction" was applied to compensate the image deformation (non-rigid transformation) induced by expansion or contraction of the brain tissue over long timescales (over minutes to an hour). This was achieved by adjusting the mean image of each imaging block (1000 frames) to the global reference image (mean image of the middle block of a session), by sequential rigid, affine, and non-rigid transformations, in this order. All of these transformations were combined into one nonlinear displacement field using cubic interpolation and applied to the imaging data in a single step to minimize interpolation artifacts.

After the motion correction, potential fluorescence from sources and neuropils was extracted by Suite2P[125]. These sources were manually curated to remove sources not corresponding to cells and those with artifacts (e.g., drifting fluorescence baseline) or low signal-to-noise ratio. For the calculation of dF/$F_0$, $F_0$ was estimated as the running median of the raw fluorescence signal from each source. dF was estimated by subtracting neuropil fluorescence from the source fluorescence to extract the fluorescence from a cell, $F_{cell}$, and then subtracting the baseline of $F_{cell}$ (estimated by its running median).

dF/F was deconvolved by OASIS AR1[126], which models each fluorescence transient as a spike of an instantaneous increase followed by an exponential decay, whose decay constant was fitted to each cell. The deconvolved signal was sparse in time and varied in magnitude in units of dF/F. For each imaging frame (35 ms bin), the deconvolved signal was binarized into a spike if the signal magnitude was greater than zero, or no spike otherwise.

## Researcher degrees of freedom

Before data collection, we pre-specified locations of optogenetics inhibition and the groupings of locations into areas for pooled analysis based on the Allen Mouse Brain Common Coordinate Framework[71]. Because our study was exploratory, we did not pre-specify hypotheses to be tested, location of imaging, data analysis methods, statistical tests, indexes, or metrics. We did not pre-specify the number of mice or cells to be collected, but the collected data size was comparable to or larger than previous studies from our lab. For the optogenetics experiments, data collection typically continued as long as the mice maintained accurate performance on control trials (greater than ~90%). For the two-photon calcium imaging experiments, data collection typically continued while GCaMP6s was expressed at an optimal level (~2–10 weeks after viral injection) and terminated with its overexpression, which was detected by GCaMP6s expression in the nucleus of several cells in fields-of-view. We excluded mice from the experiments if they did not progress to the final stage of the training to perform the delayed match-to-sample task with reasonable accuracy (greater than ~80% correct). We excluded imaging sessions from the analyses if the imaged area was outside of the areas of interest or if the mouse performed with low accuracy (less than ~70% correct). We excluded cells from the GLM-based analyses if the model fit did not converge (typically due to temporally sparse activity). 1–16% of cells were excluded in V1, RSC, MM, and A. The number and fraction of cells included for each analysis are reported in the figure legends.

## Statistical analysis

Bootstrap methods were used to compute standard errors and to perform statistical tests unless noted otherwise. The bootstrap methods were performed by resampling the pooled data with replacement $10^4$ times. For statistical tests on behavioral performance in the optogenetics experiments (Fig. 1h–n), $n_{a,i}$ trials were pooled across sessions for each photostimulated area ($P_a$) (or control) and for each mouse ($i$) and bootstrapped to compute the weighted average performance $P_a$. For statistical tests on the activity of individual cells (Fig. 3, Fig. 4), cells were pooled across sessions and mice for the same imaged area and bootstrapped, unless noted otherwise. For statistical tests on the activity of simultaneously imaged populations (Fig. 6, Supplementary Fig. 11), cells were pooled on each session and bootstrapped.

We computed p-values for the difference across conditions as in two-tailed tests because the variable of interest could be larger or smaller across conditions (e.g., areas). Specifically, we computed the fraction of resampling that generated the statistic above ($p_+$) or below ($1-p_+$) the null hypothesis (zero difference). Then the smaller of the two fractions was multiplied by two to yield the p-value. The threshold for statistical significance was set at $\theta = 0.05$. The significance threshold was adjusted for multiple comparisons by Bonferroni correction, unless noted otherwise. The sample size (n), p-values, and statistical significance were reported in the Figure legends.

## Choice biases

Choice biases were measured for color (i.e., turns toward a black or white T-arm) and direction (i.e., turns toward a right or left T-arm) (Supplementary Fig. 1h). For each session, let $P_{X/Y}$ denote the task performance (% correct) on a trial type with X sample cue and Y test cue. Then, biases were measured by the following indices:

Color bias index:

$$\left| \frac{(P_{B/BW} + P_{B/WB}) - (P_{W/BW} + P_{W/WB})}{(P_{B/BW} + P_{B/WB}) + (P_{W/BW} + P_{W/WB})} \right| \quad (1)$$

Direction bias index:

$$\left| \frac{(P_{B/WB} + P_{W/BW}) - (P_{B/BW} + P_{W/WB})}{(P_{B/WB} + P_{W/BW}) + (P_{B/BW} + P_{W/WB})} \right| \quad (2)$$

Each index can range from 0 to 1, where 0 indicates no bias and 1 indicates that all choices were made for a particular color or direction. Because the mice in some sessions showed running direction biases based on the identity of the sample cue (Fig. 1e, Supplementary Fig. 1k) prior to the test cue onset, the mice could show different direction biases for different sample cues (e.g., turning more to the right on black sample cue trials and more to the left on white-sample cue trials). To measure such a bias, we calculated the direction bias index conditioned on the sample cue (Supplementary Fig. 1h), which can also range from 0 to 1, as follows:

Direction bias index conditioned on the same sample cue:

$$\frac{1}{2} \left( \left| \frac{P_{B/WB} - P_{B/BW}}{P_{B/WB} + P_{B/BW}} \right| + \left| \frac{P_{W/BW} - P_{W/WB}}{P_{W/BW} + P_{W/WB}} \right| \right) \quad (3)$$

## Behavioral effects of optogenetics inhibition

Task performance (% correct) in Fig. 1h–n was computed by pooling behavioral results from all mice. For given inhibition areas $a$ (either a bilateral pair or area), we computed the task performance for each mouse, $P_{a,i}$, where $i$ is an index for mice. Because the number of inhibition trials for a given area, $n_{a,i}$, was different across mice, the average performance across $m$ mice was weighted based on the number of trials as follows: $P_a = \sum_i^m w_{a,i} \cdot P_{a,i}$, where the weight for the $i$-th mouse was given by $w_{a,i} = n_{a,i} / \sum_j^m n_{a,j}$. Similarly, the change in the task performance associated with the inhibition was computed by $\Delta P_a = \sum_i^m w_{a,i} \cdot (P_{a,i} - P_{control,i})$, where $P_{a,i}$ and $P_{control,i}$ are the task performance during inhibition and control trials for the $i$-th mouse, respectively.

## Decoding task variables from running patterns

Task variables were decoded from running patterns of mice to assess whether and when they varied with task variables. The running velocity was measured around the three rotation axes and z-scored. For decoding of the sample cue in the delay segment, the sample cue identity (black or white) was decoded from the mean z-scored velocity during the last 0.35 s in the delay segment (Fig. 1e, Supplementary Fig. 1k). For decoding of choice in the test segment (Supplementary Figs. 1i, j, 5c), because the running patterns could depend on the sample cue identity prior to the test segment (Fig. 1e, Supplementary Fig. 1k), the choice (right or left) was decoded separately from trials with each identity of the sample cue. The choice for a given time point in the test segment was decoded from the mean z-scored velocity in the previous 0.35 s (Supplementary Figs. 1i, j, 5c).

Decoding was performed by first fitting a model using trials in a training set (80% of trials), and then predicting the task variable from the running patterns in trials in a test set (20% of trials), cycling through 5 splits of training/test sets. Because both the sample cue and choice were binary variables (A or B), the logit, $\log[P(A)/P(B)]$, was fitted by a weighted sum of the mean z-scored velocity by a logistic regression. Regularization and cross validation were used for fitting, following a similar procedure for fitting neural activity (see below). The identity of task variables was predicted for trials in the test set such that A (or B) was predicted if $\log[P(A)/P(B)] > 0$ (or <0). If $\log[P(A)/P(B)] = 0$, the decoded identity was randomly assigned. The decoding accuracy was the fraction of trials in which the predicted identity matched the observed identity in the experiment.

## Predicting choice from task variables and running patterns

The explanatory power of the task variable and running patterns on the mouse' choices was evaluated by a logistic regression. For Fig. 1e, the probability of turning to choose the white side of the test cue, $P_{white}$, was fitted by the presented sample cue's identity, $S$, and decoded sample cue's identity, $\hat{S}$, based on the mouse's running patterns during the last 0.35 s in the delay segment as follows:

$$\log \frac{P_{white}}{1 - P_{white}} = \beta_0 + \beta_1 S + \beta_2 \hat{S} \quad (4)$$

where $S$ and $\hat{S}$ took the value of +1 for the white and −1 for the black sample cue.

We sampled an equal number of white-chosen and black-chosen trials and used L2-regularization with 10-fold cross validation. If the mouse showed similar running patterns between trials with the black- and white-sample cues, $\beta_1$ would be larger than $\beta_2$. In contrast, if the mouse showed variable but distinct running patterns that predicted the choices, as in the case of relying on the running mnemonic, $\beta_2$ would be larger than $\beta_1$.

## Calculation of the mean and normalized activity across cells

Deconvolved neural activity was first smoothed by a moving average of 10 frames (350 ms) and then aligned to the start and end of the sample segment, start and end of the delay segment, start of the test segment, and T-intersection. In each segment, neural activity was plotted for the first and last parts of the segment (sample segment: first 1.2 s and last 1.2 s, delay segment: first 0.3 s and last 0.3 s, test segment: first 2 s and last 1.5 s prior to T-intersection, T-arm: 1.5 s after T-intersection). For each cell, neural activity was averaged across all trials, and then the mean activities for individual cells imaged in a specific area (across sessions and mice) were averaged to compute the mean activity for the area (Fig. 2c). For a plot of sequential activity across cells (Fig. 2d), on each cell, activity was averaged across trials and the average activity at each time point was divided by the peak average activity across analysis time windows. Cells were then sorted in the order of peak time. The sequence of activity was cross-validated by plotting activity on even-numbered trials sorted by peak time on odd-numbered trials. For the average activity of example cells (Fig. 2e–j, Supplementary Fig. 3), the activity was smoothed by running window of 350 ms (10 imaging frames), aligned to the start and end of each segment, and averaged for each trial type separately for correct and error trials.

## Generalized linear model

We used a generalized linear model (GLM) to model the time-dependent effects of experimentally measured variables, including those related to the task and the mouse's behavior, on the single trial activity of each neuron[39,74,127,128]. Figure 2k illustrates our GLM in a schematic diagram. We used a Poisson GLM because a Poisson error distribution is restricted to non-negative values, as for neural activity, and reasonably approximates a distribution of spike counts.

The objective of our GLM is to fit the model output about the distribution of spike counts to the observed distribution of a single-cell activity. The mean spike count of the Poisson GLM, μ, was related to the weighted sum of predictors ($\mathbf{X\beta}$) by an exponential link function

$|\mu| = \exp(\mathbf{X}\boldsymbol{\beta})$. We will describe below (i) the design matrix of predictors (**X**), (ii) regularized fitting procedure to determine the weights ($\beta$ coefficients), and (iii) evaluation of model performance based on the explanatory power.

**Predictors.** The predictors were divided into the task variable set (124 or 248 predictors) and the movement variable set (39 predictors).

The task variable set consisted of 8 subsets: offset (O), sample cue (S), test cue (T), choice (c), and the interactions between S, T, and C (S*T, T*C, S*C, S*T*C). S, T, and C were binary variables such that +1 (or −1) was assigned to white (or black) for the sample cue, BW (or WB) for the test cue, and right turn (or left turn) for the choice (Fig. 2k). Note that the interaction S*T represents XOR on the sample cue identity and test cue identity and is equivalent to choice identity (C) on correct trials. Only the first 4 subsets (O, S, T, C) were used to fit correct trials in 4 trial types (4 conditions) (Fig. 3, Fig. 5). All 8 subsets were used to fit both correct and error trials in 4 trial types ($2 \times 4 = 8$ conditions) (Fig. 4). Because each cell tended to have activity in a specific part of the maze (Fig. 2d), each subset comprised 31 spatially tuned predictors that collectively covered the entire maze length, yielding $31 \times 4 = 124$ or $31 \times 8 = 248$ predictors for the task variable set. Similarly to a previous study[39], the spatial tuning of each cell was expanded using raised cosine basis functions with a width of 40 cm,

$$f(x) = \begin{cases} V \cdot \frac{1}{2}\left[1 + \cos\left(2\pi \cdot \frac{(x-x_c)}{40}\right)\right], & \text{if } x_c - 20 < x < x_c + 20 \\ 0, & \text{otherwise} \end{cases} \quad (5)$$

and its center peak was either positive (+1) or negative (−1) based on the identity of a binary task variable ($V$), which denotes either the sample cue, test cue, choice, or their interactions. The center peaks $x_c$ were spaced with a 20 cm interval to tile the maze with a half-width overlap.

The movement variable set consisted of 3 subsets corresponding to the roll, pitch, and yaw velocity. For each component of velocity, the velocity was z-scored and bounded from −3 to 3. We assumed that the movement selectivity was invariant across maze position, so that movement predictors are functions of the z-scored velocity ($z$), but not on the maze position. The shape of velocity tuning followed one cycle of a raised cosine function with a width of 1 in z-scored velocity,

$$g(z) = \begin{cases} \frac{1}{2}\left[1 + \cos(2\pi \cdot (z - z_c))\right], & \text{if } z_c - 0.5 < z < z_c + 0.5 \\ 0, & \text{otherwise} \end{cases} \quad (6)$$

We included 13 predictors for each velocity component whose centers $z_c$ spanned from −3 to 3 with a spacing of 0.5 along z-scored velocity, yielding $13 \times 3 = 39$ predictors for the movement variable set.

**Regularized fitting procedure.** For each imaging session, the GLM was fitted to the neural activity with 80% of the trials (training set), whereas the data in the rest of the 20% of the trials were used to evaluate the fit quality and encoded information in individual cells (see below). Trials with the same condition (defined by the combination of the sample cue, test cue, and choice; $2 \times 2 \times 2 = 8$ conditions) were divided evenly into five subsets. Four subsets were used as the training set and the remaining one subset was used as the test set. We cycled through 5 splits of training/test set to evaluate all appropriate trials in the session.

Given the model output of the expected spike counts, $\mu = \exp(\mathbf{X}\boldsymbol{\beta})$, the likelihood of observing spike counts, $r$, follows a Poisson distribution: $P(r|\mu) = \mu^r \exp(-\mu)/r!$. The weights $\boldsymbol{\beta} = [\beta_0 .. \beta_M]^T$ were fitted to the data by minimizing the following objective function

with glmnet package in R (v2.0-16) with elastic-net regularization[129]:

$$-\frac{1}{N}\sum_{i=1}^{N}[r_i(\beta_0 + \mathbf{x}_i \cdot \boldsymbol{\beta}) - \exp(\beta_0 + \mathbf{x}_i \cdot \boldsymbol{\beta})] + \lambda\left[\alpha\sum_{j=1}^{M}|\beta_j| + (1-\alpha)\sum_{j=1}^{M}\frac{\beta_j^2}{2}\right] \quad (7)$$

Here $i$ is an index for the imaging frame, $N$ is the total number of frames in the training data, $j$ is an index for beta coefficients, and $M$ is the total number of beta coefficients without $\beta_0$ ($M = 124$ or 248 for full model). $r_i$ is the neural activity in the $i$-th frame calculated by the sum of estimated spike counts in neighboring 10 frames ($[i-5 .. i+4]$ th frames spanning ~350 ms). $\mathbf{x}_i$ is a set of values (i.e., a vector) of $M$ predictors for the $i$th frame, $\beta$ is a vector of $M$ beta coefficients for the cell. We used L1-like regularization with $\alpha = 0.95$, which tends to explain responses with a smaller number of large beta coefficients (as opposed to a large number of small beta coefficients for L2 regularization with $\alpha \approx 0$).

The regularization parameter, $\lambda$, sets the tradeoff between the fitting error (the first term in Eq. (7)) and the model complexity (the second term in Eq. (7)). The value of $\lambda$ was chosen from a set of $\lambda$ values by 10-fold cross validation, where each fold had an equal number of trials per condition (as in the training/test splits described above). For each value of $\lambda$, we computed 10 instances of the fitting error (deviance) from the cross validation and computed their mean, $\mu_\lambda$, and standard deviation, $\sigma_\lambda$ (i.e., standard error of the deviance). We found the $\lambda_{min}$ that produced the minimum deviance $\mu_{\lambda min}$, and $\lambda_{1se}$ that produced the most regularized model whose mean deviance was within one standard error from the minimum deviance, $\arg max_{\lambda \in [\lambda min, \infty]}[\mu_\lambda < (\mu_{\lambda min} + \sigma_{\lambda min})]$. We used $\lambda_{1se}$ to fit the model to all trials in the training set, and obtained fitted beta coefficients, $\boldsymbol{\beta}_{1se}$, which were used to predict responses for trials in the test set and to evaluate information for task variables (Fig. 3, Fig. 4, Fig. 5, see below).

Because the number of correct trials was greater than error trials, the imbalance in the number of trials could bias the fitting procedure to depend more on correct trials than on error trials. To compensate for this imbalance, we adjusted the weight of each trial during the fit. Specifically, on each trial with a specific trial type and trial outcome (correct or error), the fitting error of that trial was multiplied by the inverse of the total number of trials with the same trial type and outcome during the session. Results were qualitatively consistent with and without this balancing for differences in trial numbers. Thus, we showed results without the weight adjustment (i.e., equal weights for all trials).

**Evaluation of model performance.** The model performance was evaluated by the fraction of the deviance explained (FDE) on the test data set (Fig. 2l, m). FDE quantifies the performance of the fitted GLM on the test data with respect to the *null* and *saturated* models. For each time point in the test data, the null model predicts the same expected activity, which is given by the activity of a cell averaged across all time points in the training data. In contrast, the saturated model predicts the expected activity that exactly matches the observed activity at each time point in the test data. Let $L_{GLM}$, $L_{null}$, and $L_{sat}$ be the log likelihood of observing the test data given the GLM, null, and saturated model, respectively. Then, the deviance ($D$) and the FDE are defined as follows.

$$\begin{cases} D_{null} = 2(L_{sat} - L_{null}) \\ D_{GLM} = 2(L_{sat} - L_{GLM}) \end{cases} \quad (8)$$

$$\begin{aligned} FDE &= \frac{D_{null} - D_{GLM}}{D_{null}} \\ &= \frac{L_{GLM} - L_{null}}{L_{sat} - L_{null}} \end{aligned} \quad (9)$$

Thus, FDE = 1 indicates that the model prediction was as good as the saturated model, whereas FDE = 0 indicates that the model prediction was as poor as the null model. Note that FDE is a goodness-of-fit statistic and related to the fraction of variance explained (FVE). FVE would be appropriate under the assumption that each data point has the same Gaussian error distribution around the mean. However, because we used Poisson GLM with the assumption of Poisson error distributions, we chose FDE to take into account non-Gaussian error distributions that varied across time points.

## Single-cell information measured as log-likelihood ratio

We used the GLM fitted to the training data to compute the single-trial information for each cell about each task variable in each trial in the test data. We first describe the derivation of sample cue information below, and this derivation extends easily to test cue information and XOR information. Given that the activity of a cell at time $t$ in a given trial had a value of $r(t)$, we used the Poisson GLM to compute the likelihood that the observed activity was generated either in the presence of a

Note that P(T,C|S = +1) was approximated by $Pc/2$ or $Pe/2$, where the $Pc$ and $Pe$ denote the proportion of correct trials and that of error trials, respectively.

In a similar fashion, test cue information was computed as:

$$\log \frac{P(r(t)|T=+1,V(t))}{P(r(t)|T=-1,V(t))}$$
$$= \log \frac{\sum_S \sum_C P(r(t)|S,T=+1,C,V(t)) \cdot P(S,C|T=+1,V(t))}{\sum_S \sum_C P(r(t)|S,T=-1,C,V(t)) \cdot P(S,C|T=-1,V(t))} \quad (13)$$

For logLR of XOR, the likelihood was marginalized over the combinations of the sample cue and test cue that were associated with the same direction of reward assignment (e.g., XOR = +1 consists of [$S$ = +1, $T$ = +1] and [$S$ = −1, $T$ = −1]). Thus, its logLR was calculated as follows.

$$\log \frac{P(r(t)|XOR=+1,V(t))}{P(r(t)|XOR=-1,V(t))} = \log \frac{\sum_{[S,T]\in\{[+1,+1][-1,-1]\}} \sum_C P(r(t)|S,T,C,V(t)) \cdot P(C|S,T,V(t))}{\sum_{[S,T]\in\{[+1,-1][-1,+1]\}} \sum_C P(r(t)|S,T,C,V(t)) \cdot P(C|S,T,V(t))}$$
$$\approx \log \frac{\sum_{[S,T]\in\{[+1,+1][-1,-1]\}} \sum_C P(r(t)|S,T,C,V(t)) \cdot P(C|S,T)}{\sum_{[S,T]\in\{[+1,-1][-1,+1]\}} \sum_C P(r(t)|S,T,C,V(t)) \cdot P(C|S,T)}$$
$$= \log \frac{P(r(t)|S=+1,T=+1,C=+1,V(t)) \cdot Pc/2 \quad + P(r(t)|S=+1,T=+1,C=-1,V(t)) \cdot Pe/2 \quad + P(r(t)|S=-1,T=-1,C=+1,V(t)) \cdot Pc/2 \quad + P(r(t)|S=-1,T=-1,C=-1,V(t)) \cdot Pe/2}{P(r(t)|S=+1,T=-1,C=-1,V(t)) \cdot Pc/2 \quad + P(r(t)|S=+1,T=-1,C=+1,V(t)) \cdot Pe/2 \quad + P(r(t)|S=-1,T=+1,C=-1,V(t)) \cdot Pc/2 \quad + P(r(t)|S=-1,T=+1,C=+1,V(t)) \cdot Pe/2} \quad (14)$$

white-sample cue (S = +1) or a black sample cue (S = −1). Then, the single-trial information was computed as the log-likelihood ratio (logLR)[77,78,130]:

$$\log \frac{P(r(t)|S=+1)}{P(r(t)|S=-1)} \quad (10)$$

The logLR monotonically varies with the likelihood ratio, its sign indicates whether white (positive) or black (negative) was more likely, and its magnitude (absolute value) quantifies its informativeness. Because the running patterns of the mice can covary with the task variable identities (Fig. 1e, Supplementary Fig. 1k) and influence the neural activity[18,75,131], we controlled for running variability to derive "genuine" sample cue information by computing the log-likelihood also conditioned on the running velocity as follows.

$$\log \frac{P(r(t)|S=+1,V(t))}{P(r(t)|S=-1,V(t))} \quad (11)$$

To compute the likelihood of the given sample cue, we marginalized over all possible combinations of the remaining task variables (i.e., test cue and choice in this case).

$$P(r(t)|S=+1,V(t)) = \sum_{T\in-1,+1}\sum_{C\in-1,+1} P(r(t)|S=+1,T,C,V(t)) \cdot P(T,C|S=+1,V(t))$$
$$\approx \sum_{T\in-1,+1}\sum_{C\in-1,+1} P(r(t)|S=+1,T,C,V(t)) \cdot P(T,C|S=+1)$$
$$= P(r|S=+1,T=-1,C=-1) \cdot P_c/2 + P(r|S=+1,T=-1,C=+1)$$
$$\cdot P_e/2 + P(r|S=+1,T=+1,C=-1)$$
$$\cdot P_e/2 + P(r|S=+1,T=+1,C=+1) \cdot P_c/2$$
$$\quad (12)$$

The logLR is a signed value, where its sign indicates whether the activity supports one identity of the task variables or the other. For example, cells with the sample cue selectivity tend to show positive logLR for trials with S = +1 (white sample cue) and negative logLR for trials with S = −1 (black sample cue). As a result, if we simply average the logLR across all trials, they would cancel out and fail to detect the sample cue information. Therefore, we adjusted the sign of logLR, so that a positive logLR indicated that the neural activity correctly encoded the task variable identity observed in the trial. In contrast, a negative logLR indicated incorrect encoding of the task variable identity. Specifically, when the observed task variable identity was −1 in Eqs. (11–14), we multiplied the logLR by −1. In the results, we used the sign-adjusted logLR (Figs. 3–5, Supplementary Fig. 5, Supplementary Fig. 6, Supplementary Fig. 8).

## Population decoding and population mutual information

For the analyses of population information, instead of the single-cell activity $r(t)$, we used a vector of population activity, $\mathbf{r}(t)$, which was imaged simultaneously on each session within a given field-of-view (~0.5 mm² area). Under the assumption of conditionally independent noise across cells, the likelihood of observing a population activity $\mathbf{r}(t)$ was calculated from the GLM as follows:

$$P(\mathbf{r}(t)|S,T,C,V(t)) = \prod_{i=1}^n P(r_i(t)|S,T,C,V(t)) \quad (15)$$

where $i$ is an index for a cell in a population of $n$ cells. Combining Eqs. (14) and (15) yields population XOR information as follows:

$$
\log \frac{P(\mathbf{r}(t)|XOR = +1, V(t))}{P(\mathbf{r}(t)|XOR = -1, V(t))}
$$

$$
= \log \frac{\sum\limits_{[S,T]\in\{[+1,+1][-1,-1]\}} \sum\limits_{C} P(\mathbf{r}(t)|S,T,C,V(t)) \cdot P(C|S,T,V(t))}{\sum\limits_{[S,T]\in\{[+1,-1][-1,+1]\}} \sum\limits_{C} P(\mathbf{r}(t)|S,T,C,V(t)) \cdot P(C|S,T,V(t))}
$$

$$
\approx \log \frac{\sum\limits_{[S,T]\in\{[+1,+1][-1,-1]\}} \sum\limits_{C} P(\mathbf{r}(t)|S,T,C,V(t)) \cdot P(C|S,T)}{\sum\limits_{[S,T]\in\{[+1,-1][-1,+1]\}} \sum\limits_{C} P(\mathbf{r}(t)|S,T,C,V(t)) \cdot P(C|S,T)}
$$

$$
= \log \frac{P(\mathbf{r}(t)|S=+1,T=+1,C=+1,V(t))\cdot Pc/2 \; + P(\mathbf{r}(t)|S=+1,T=+1,C=-1,V(t))\cdot Pe/2 \; + P(\mathbf{r}(t)|S=-1,T=-1,C=+1,V(t))\cdot Pc/2 \; + P(\mathbf{r}(t)|S=-1,T=-1,C=-1,V(t))\cdot Pe/2}{P(\mathbf{r}(t)|S=+1,T=-1,C=-1,V(t))\cdot Pc/2 \; + P(\mathbf{r}(t)|S=+1,T=-1,C=+1,V(t))\cdot Pe/2 \; + P(\mathbf{r}(t)|S=-1,T=+1,C=-1,V(t))\cdot Pc/2 \; + P(\mathbf{r}(t)|S=-1,T=+1,C=+1,V(t))\cdot Pe/2}
$$

$$
= \log \frac{\prod\limits_{i=1}^{n} P(r_i(t)|S=+1,T=+1,C=+1,V(t))\cdot Pc/2 \; + \prod\limits_{i=1}^{n} P(r_i(t)|S=+1,T=+1,C=-1,V(t))\cdot Pe/2 \; + \prod\limits_{i=1}^{n} P(r_i(t)|S=-1,T=-1,C=+1,V(t))\cdot Pc/2 \; + \prod\limits_{i=1}^{n} P(r_i(t)|S=-1,T=-1,C=-1,V(t))\cdot Pe/2}{\prod\limits_{i=1}^{n} P(r_i(t)|S=+1,T=-1,C=-1,V(t))\cdot Pc/2 \; + \prod\limits_{i=1}^{n} P(r_i(t)|S=+1,T=-1,C=+1,V(t))\cdot Pe/2 \; + \prod\limits_{i=1}^{n} P(r_i(t)|S=-1,T=+1,C=-1,V(t))\cdot Pc/2 \; + \prod\limits_{i=1}^{n} P(r_i(t)|S=-1,T=+1,C=+1,V(t))\cdot Pe/2}
$$

(16)

The population information was derived using all cells in each imaging sessions (Fig. 6), or a subpopulation with the same number of cells (n = 100 cells, randomly sampled from the population, Supplementary Fig. 11).

The identity of the task variable was decoded from population activity based on the sign of population information (logLR). When logLR was zero, the decoded identity was randomly assigned. On each session, the accuracy of the population coding was evaluated as the mutual information in the decoding confusion matrix, that is the mutual information between the true and decoded identity of the task variable[86]. The weighted average of mutual information was reported by averaging across sessions with weights proportional to the number of trials on the sessions (Fig. 6b, d, Supplementary Figs. 9d, 11a, c–e).

### Other population decoding methods

Our population decoding method described above modeled the likelihood of activities in individual neurons with the GLM and assumed conditional independence of neurons. Whereas the GLM is useful to discount the effect of the mouse's movement on the neural activity, the assumption of conditional independence may fail to capture some aspects of the information in population activity, including the information carried by the modulation of noise correlations for different cues or choices, termed stimulus-dependent correlations[132]. To understand if our conclusions were robust to changes of assumptions in decoders, we also decoded neural population activity with another linear decoding method (logistic regression) and with a nonlinear decoding method (support vector machine with the redial basis function kernel) (Supplementary Fig. 7). As a population activity, we used mean dF/F during the first 1 s (30 imaging frames) in the test segment with deconvolution (Supplementary Fig. 7a–d) or without deconvolution (Supplementary Fig. 7e–h). To equalize the population size across sessions, 100 cells were randomly subsampled from a simultaneously imaged population. Subsampling was repeated 100 times to estimate the mean and s.e.m. of the population decoding accuracy.

The logistic regression was used as an alternative linear decoding method. Because each task variable had binary identity (A or B), the logit, log[P(A)/P(B)], was fitted by a weighted sum of the simultaneously imaged neural activity (dF/F) with deconvolution (Supplementary Fig. 7a–c) or without deconvolution (Supplementary Fig. 7e–g). Decoding was performed by first fitting a model using trials in a training set (80% of trials), and then predicting the task variable from the neural activity in trials in a test set (20% of trials), cycling through 5 splits of training/test sets. Regularization and cross-validation were used for fitting, following a similar procedure for fitting neural activity (see *regularized fitting procedure*), except with L2 regularization. The identity of task variables was predicted for trials in the test set based on the sign of the logit. The decoding accuracy was the fraction of trials in which the predicted identity matched the observed identity in the experiment.

The support vector machine with the redial basis function kernel was used as a nonlinear decoding method[133,134]

(Supplementary Fig. 7d, h). This method captures nonlinear relationships between the activity of different neurons and can thus capture the information potentially carried by the cue or choice modulation of noise correlations. We used 5 splits of training/test sets, and 10-fold cross validation to optimize two hyper parameters: $C$ (>0) used as a cost parameter for classification errors and $\gamma$ (>0) used to determine the width of the radial basis function $K(x_i, x_j) = \exp(-\gamma|x_i - x_j|^2)$. For each combination of $C$ and $\gamma$, the model was fitted by a MATLAB package of *libsvm* (https://github.com/cjlin1/libsvm)[135].

### Signal and noise correlation

For each pair of cells recorded simultaneously, the signal correlation was quantified by Pearson's correlation on their mean neural activity across four trial types. To compute the noise correlation for each pair, the *residual* neural activity was derived for each cell on each trial type, by subtracting the mean activity on each trial type from the observed activity on that trial type. The noise correlation was quantified by Pearson's correlation on the residual activity of the pair of neurons combined across all trials in all trial types. To evaluate the effect of noise correlation on the accuracy of the population code, the noise correlation was disrupted by shuffling the trial identity within each condition (defined by the combination of the sample cue, test cue, and choice; 2 ×2 x 2 = 8 conditions) for each cell. Trial shuffling was repeated $10^4$ times to estimate the standard error of decoding accuracy.

### Simulation of population activity

To compare the efficiency and accuracy of encoded information between a population of mixed selectivity cells and a population of pure selectivity cells, we generated population activity by Monte Carlo simulation and analyzed the simulated data similarly to the real imaging data. In the simulation, the number of spikes generated by each cell followed a Poisson distribution with the mean $\lambda$. Each cell responded to a preferred cue (or a preferred combination of cues) with the mean of $\lambda_{pref}$ and to an unpreferred cue with the mean of $\lambda_{unpref}$, where $\lambda_{pref}$ was greater than $\lambda_{unpref}$ by design. The minimum unit population size was a set of four cells, each of which preferentially responded to a specific trial type in the mixed selectivity population, or to a specific sample cue or test cue identity in the pure selectivity population (Fig. 7a). In the simulation, we varied $\lambda_{pref}$, $\lambda_{unpref}$, the population size (by including N sets of the unit population), and the level of noise correlations across cells. Because the noise correlations were mostly observed for cells with a higher signal correlation (Supplementary Fig. 9a–c), the simulation included positive noise correlation only between cells that shared the same selectivity. To introduce noise correlations across n cells, our simulation first generated a set of n Gaussian-distributed random variables with an n-by-n correlation matrix using *copularnd.m* in MATLAB (MathWorks), and then correlated Poisson-distributed random variables were generated by the inverse CDF sampling technique[136] using *poissinv.m* in MATLAB (MathWorks). Population decoding was performed by following the

decoding method for the real imaging data as described above. A total of $10^4$ trials were simulated $10^4$ times for each set of free parameters to estimate the mean and s.e.m. of the decoding accuracy.

### Analyses conditioned on the cue information

To evaluate the ratio of information for the sample cue vs. test cue across individual cells, the two types of information represented in the cartesian coordinates (Figs. 3g, 4h, 5h) were converted to polar coordinates (Figs. 3h–n, 4i–k, 5i–m). For the analyses that are conditioned on the polar angle (Figs. 3h–n, 4i–k, 5i–m, 6b–d, Supplementary Figs. 5e–k, 5o, 6d, 8f–h, 9, 11), cells with zero or noise-level information ($|\log LR| < \epsilon$) were not assigned an angle because their angle could not be reliably estimated. The noise level $\epsilon$ was set to 0.01, which approximates the chance-level information ($\log LR = 0.012$ for V1, 0.004 for RSC, and 0.003 for MM) inferred for each cortical area by one standard deviation of the test cue information averaged across the last 1 s (30 frames) in the sample segment (i.e., prior to the appearance of the test cue). Because the test cue has not been presented in the sample cue segment, any test cue information is expected to be noise and thus is a sensible estimate of our noise level. We verified that the results did not qualitatively change even when we varied $\epsilon$ in the range of $\log LR = 0.005$–0.02 or used the area-specific chance value as $\epsilon$ for each area. The skewness of the angle distribution of cells was calculated without cells with extreme angles (the highest and lowest 1% of cells) because the skewness is sensitive to outliers (Fig. 3h). To show the angle distribution of cells, histograms were created by discretizing the angles into 7 bins (Figs. 3i, 4i, 5i, Supplementary Figs. 5f, 5o, 6d, 8f). Cells with noise-level information were not categorized into these bins, but counted towards the total number of cells to calculate the fractions across the bins. Bins for cells with only sample cue information or test cue information ($\theta = 0°$ and $\theta = 90°$) included cells within a chance-level deviation from the two axes. For the analyses of normalized activity (Fig. 3j, Supplementary Fig. 5g, i) and population information (Fig. 6, Supplementary Figs. 9d, 11), cells in the bins for $\theta = 0°$ and $\theta = 90°$ with less than the chance-level magnitude of XOR information (Figs. 3i, 4i correct trials, 5i) were defined to have pure selectivity to the sample cue and test cue, respectively, whereas cells with $15° \leq \theta < 75°$ (Figs. 3g, 4h correct trials, 5h) were defined to have mixed selectivity. Similarly, pure XOR selective cells were defined to have XOR information greater than logLR of 0.01 and less than the chance-level magnitude of the sample and test cue information (Supplementary Fig. 5l, m).

### Comparison of XOR information for correct vs. error trials conditioned on the sample cue information

We tested whether XOR information between correct and error trials was different conditioned on the same amount of preexisting sample cue information (Fig. 4l, Supplementary Fig. 8i). Trials were binned according to their sample cue information during the last 0.35 s in the delay segment (bin size = 0.01 logLR). The mean XOR information for correct and error trials was calculated by averaging over trials within a bin, $\bar{I}_{\{XOR,cor\}} = \frac{1}{Nc} \sum_i^{Nc} I_{i,\{XOR,cor\}}$ and $\bar{I}_{\{XOR,err\}} = \frac{1}{Ne} \sum_j^{Ne} I_{j,\{XOR,err\}}$, where $Nc$ and $Ne$ are the number of correct and error trials within the bin (combined across sessions), respectively; $I_{i,\{XOR,cor\}}$ and $I_{j,\{XOR,err\}}$ are the mean XOR information across all cells in a session for the $i$-th correct and $j$-th error trials within the bin, respectively. $\bar{I}_{\{XOR,cor\}}$ and $\bar{I}_{\{XOR,err\}}$ were computed $10^4$ times by bootstrapping on trials, and the statistical significance of their difference was tested against a null hypothesis $\bar{I}_{\{XOR,cor\}} = \bar{I}_{\{XOR,err\}}$. For comparison of two areas (A and B), we tested whether area A showed a larger difference in XOR information between correct vs. error trials than area B. $\Delta\bar{I}_{\{XOR\}} = \bar{I}_{\{XOR,err\}} - \bar{I}_{\{XOR,cor\}}$ was computed for the two areas A and B, denoted as $\Delta\bar{I}_{A,\{XOR\}}$

and $\Delta\bar{I}_{B,\{XOR\}}$. The statistical significance of their difference was tested against a null hypothesis $\Delta\bar{I}_{A,\{XOR\}} = \Delta\bar{I}_{B,\{XOR\}}$ by bootstrapping on trials.

### Normalization of activity across trial types

We evaluated the distribution of activity across the four trial types in individual cells (Figs. 3j, 4j, 5j, k, Supplementary Figs. 5g–i, 8g). For each cell, the mean activity on correct trials was calculated for each of the four trial types: $\mathbf{r} = [r_{B/BW}, r_{B/WB}, r_{W/BW}, r_{W/WB}]$, where $r_{X/Y}$ denotes the mean deconvolved dF/F for the trial type with X sample cue and Y test cue. We normalized the sum of the activity across the trial types to one by multiplying $\mathbf{r}$ by the normalization factor $k = 1/(r_{B/BW} + r_{B/WB} + r_{W/BW} + r_{W/WB})$. The trial type associated with the highest activity was defined to have a preferred sample cue and preferred test cue (Fig. 3l). With respect to this trial type, the other three trial types had preferred or unpreferred cues (Fig. 3l). The normalized activity for the four trial types were sorted according to these cue preferences and averaged across cells with each type of selectivity. To cross-validate the cue preferences, for each trial type, the activity in a half of trials was used to determine the cue preference of the cell and the activity in the other half of trials were sorted by the cue preference and averaged across cells. To compare the activity between correct and error trials, the mean activity on error trials was sorted by the preferred sample and test cues determined on correct trials and scaled by the normalization factor $k$ calculated on correct trials (Fig. 4j, Supplementary Fig. 8g).

### Nonlinearity index

For evaluation of nonlinear mixing of information (or nonlinear mixed selectivity), the nonlinearity index was calculated for individual cells (Figs. 3j–m, 4j, 5j–l, Supplementary Fig. 5j). If a cell has linear mixed selectivity to sample cue and test cue, the activity differences associated with the different test cues are the same regardless of the sample cue ($r_{B/BW} - r_{B/WB} = r_{W/BW} - r_{W/WB}$). In contrast, if a cell has nonlinear mixed selectivity, these differences are not equal. To quantify the degree of nonlinearity, the nonlinearity index (NI) was defined as:

$$\text{NI} = |(kr_{B/BW} - kr_{B/WB}) - (kr_{W/BW} - kr_{W/WB})| \quad (17)$$

where $k = 1/(r_{B/BW} + r_{B/WB} + r_{W/BW} + r_{W/WB})$ was set for each cell to normalize the sum of activity across the four trial types to one. Theoretically, cells that showed no trial type selectivity, pure selectivity, or linear mixed selectivity should have an index value of 0. However, the chance level index was -0.2, as determined by shuffling trial identities (Fig. 3m). The reason for the non-zero chance level is because the index takes the absolute value, and thus any noise will show up as a positive-valued index. The maximum index of 1 was assigned to cells that showed no activity for left-choice trials ($r_{B/BW} = 0$ and $r_{W/WB} = 0$) or no activity for right-choice trials ($r_{B/WB} = 0$ and $r_{W/BW} = 0$), including cells that showed activity only on a single trial type.

### Reporting summary

Further information on research design is available in the Nature Portfolio Reporting Summary linked to this article.

## Data availability

Data are available upon request to the corresponding author. Source data are provided with this paper.

## Code availability

Customized code is available upon request to the corresponding author.

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

## Acknowledgements

The authors thank members of the Harvey lab and Giuseppe Pica for helpful discussions and Leah Thomas, David Allen, Vishal Sharoff, Theodore Lutkus, Joseph Deluisi, and Alec Nelson for assistance with behavioral training of mice and experiments. This work was supported by grants from the NIH (R01 MH107620, R01 NS089521, R01 NS108410, DP1 MH125776). S.K. was supported by a NARSAD Young Investigator Grant, a JSPS Overseas Research Fellowship, an Uehara Foundation Research Fellowship, a Leonard and Isabelle Goldenson Postdoctoral Fellowship, and an Alice and Joseph Brooks Fund Fellowship. S.P. received support from the Fondation Bertarelli.

## Author contributions

S.K. and C.D.H. conceived of the study. A.S.M. and S.K. developed the behavioral task. S.K. performed all experiments (optogenetics inhibition, two-photon calcium imaging, and electrophysiology). S.K. analyzed the data with input from C.D.H., S.P., and H.S. S.P. contributed to the design of data analysis approaches. S.K., S.P., and C.D.H. interpreted the results. C.D.H. supervised all aspects of the research. S.K., S.P., and C.D.H. wrote the paper with input from H.S.

## Competing interests

The authors declare no competing interests.
