## [Peer Review File · Nature Communications]

A distributed and efficient population code of mixed selectivity neurons for flexible navigation decisionsREVIEWER COMMENTS

Reviewer #1 (Remarks to the Author):

Kira et al examined cortical areas and neural activity supporting flexible decision-making in a delayed match-to-sample task in which mice running in corridor compared a sample visual cue with a test cue to reach a navigation decision. An optogenetic screen identified posterior regions of cortex, including PPC, V1, and RSC, to be important for task performance. Population calcium imaging revealed neurons coding specific conjunctions of sample and testing cues. This nonlinear mixed selectivity is most prevalent in RSC and it is weakened during incorrect decisions. The authors proposed that this nonlinear mixed selectivity provides an efficient population code for flexible decisions.

Overall, this is a carefully done study that combines several cutting-edge approaches in a difficult decision-making task for mice that advances new ideas on how flexible decisions could be computed. The notion that neurons coding specific conjunctions of sample and test cues (XOR selectivity) provide an efficient population code is intriguing. And the observation that such selectivity is diminished before erroneous decision is interesting. However, a few conceptual issues around how mixed selectivity could be interpreted and how it produces flexible decision should be better clarified. I additionally have a few technical comments that could be addressed through additional analysis.

Major

1) The involvement of posterior cortical regions in flexible decision should be more carefully interpreted or at least clarified. There seems to be several incongruencies.

a. In several places in the text, the behavioral deficits caused by inhibiting posterior cortical regions are interpreted as those regions being involved in combining test cue with memory of sample cue for flexible decision (e.g. line 162-163 and 181). However, the behavioral deficits seem to be more (or at least equally) consistent with a deficit in visual perception which affects the detection of test cue. For example, large effects are also seen when these cortical regions are inhibited during the sample segment, which does not require combining memory and visual signals.

b. The authors propose that mixed selectivity for specific conjunctions of sample and test cues is important for flexible decision. However, in all the candidate cortical regions, the mixed signal is very sparse overall. Rather, the largest modulation appears to be visual modulation by the sample and test cues (Fig 3a-f), which seems to be more consistent with a role in visual processing.

c. Behavior deficits by site-specific optogenetic inhibitions was stronger in V1 and PPC rather than RSC (Fig. 1g-k). However, mixed selectivity is concentrated in RSC and less in V1 and PPC (Figs. 3-5). How to reconcile this mismatch between manipulation effects and neuronal selectivity across the brain regions?

d. Expanded inhibition in RSC yielded larger behavioral effects (Fig 1l). However, V1 is also a large area, which exhibits large effects even with restricted inhibition. Expanded V1 inhibition should also be provided to fairly compare to the more dramatic effect of expanded RSC inhibition.

2) The finding of nonlinear mix selectivity and its correlation with performance is interesting (i.e. higher selectivity in correct trials), which leads to the interesting proposal that the mixed selectivity has a causal link to the decision. However, a few conceptual issues regarding the function of mixed selectivity could be better clarified.

a. The text suggests that the mixed selectivity reflects “mixing” of visual and memory signals for the purpose of generating the decision. However, it is not clear if the responses reflex a mixing operation, or simply reflects a visual response modulated by the context of the preceding stimulus. For example, would mixed selectivity be encountered in these regions if mice are not engaged in decision-making behavior?

b. Conceptually, what is the relationship of nonlinear mixed selectivity to flexible decision? For a population of neurons with nonlinear mixed selectivity, decision in every instance (i.e. specific combinations of sample and test cues) would depend on different neuronal populations (coding specific conjunctions of sample and test cues). This is very different from the notion of flexible decision in the form of a “match” vs. “nonmatch” signal irrespective of the details of the stimulus. Can the authors clarify how mixed selectivity can give rise to flexible decisions if the selectivity is so specific to the exact combination of task elements? For example, if a different instance of stimulus is encountered or if a different motor response is required, the coding of decision would change.

c. The nonlinear mixed selectivity does not actually represent a code for the decision itself, as far as I can tell. It, however, could serve as a basis set from which decision could potentially be constructed.

3) A few technical comments on the quantification of selectivity.

a. Most quantifications of mixed selectivity are based on 1 second of the test segment. Based on the running pattern from the test segment, mice have completed decision 500ms after the test cue onset

(Fig S1). During the second 500ms of the test segment, different running patterns could also affect the responses of neurons, which impacts the calculation of selectivity. For these reasons, it is important to confirm that key results on mixed selectivity (Fig 3 & 4) stand if analysis is kept to the first 500 ms of the test segment.

b. The text suggests XOR selective neurons exhibit “single-trial-type selectivity”, i.e. only responding to a single combination of stimulus and test cues. This claim is based on a sparsity index of 3. It is somewhat counterintuitive what this metric reflects. Will the authors consider also including a simpler analysis to show this selectivity, for example in the form of average activities rank ordered by trial-type preference?

c. Related to point b, the authors find that neurons coding both stimulus and test cues also preferentially encode XOR (Fig 3j). It would be beneficial to also show simpler analysis to help readers understand this relationship. Perhaps by showing more single-neuron examples (as in Fig 2e-j) or average responses of different types of neuronal populations.

4) Some parts of the Introduction and Abstract should be better clarified.

a. Several places in the text refers to “behavioral goal in short-term memory”. It seems the short-term memory in the delayed match-to-sample task is about the identity of the stimulus cue. What is meant by behavioral goal here?

b. A few previous studies are not accurately cited (line 38-49):

i. Wu et al and Condylis et al both compared decision signals in multiple brain areas during similar delayed match-to-sample tasks (but not navigation decisions).

ii. Duan et al also studies trial-to-trial switch of task rules, not slow changes in sensorimotor association.

c. Several places in the text focus on “rapidly switch[ing] sensorimotor associations”. This is counterintuitive because the coding scheme of mixed selectivity relies on storing specific combination of sample and test cues. There is no mechanism supporting “switching” associations.

Minor:

1) Can the authors discuss potential differences between this study and the Pinto et al study? Pinto et al found strong involvement of frontal cortex in addition to posterior cortex. Here the frontal cortical regions are surprisingly dispensable.

2) It seems mice undergoing expanded RSC inhibition exhibit faster running overall, even in control trials. Why is this the case?

3) Statistical tests in Fig 5 require better description. Are these based on bootstrap? For the statement: "The difference between correct and error in mixed selective cells was significantly larger than that for pure selective cells in V1, RSC, and MM", this should ideally be tested for significant interaction between 2 factors, correct vs. error trials and mixed vs. pure selective cells.

4) Are the imaged V1 cells from monocular V1, binocular V1, or both? In general, the medial and lateral parts of V1 are monocular and binocular, respectively. The two types of V1 cells could be modulated differently by the test cue where monocular V1 is responding to white vs. black stimuli in the contralateral visual field, leading to higher test cue selectivity, whereas binocular V1 and RSC neurons might be modulated by ipsilateral visual field. If imaging covered both regions of V1, the authors might consider separately analyzing monocular and binocular V1 data and compare with RSC, specifically for Fig 3g-i.

5) Even though GLM poorly predicted neural activity in M1 and M2 (Fig. 2l), it will be informative to show the M1/M2 data in a similar way to RSC, V1, and PPC as in Figs 3-5, perhaps in the supplemental.

6) A few typos:

a. Fig. 1h legend, 2nd line: ... computed from the data in panel '(f)' -> '(g)'

b. Fig. 1h legend, 2nd line: ..., as marked in panel '(k)' -> '(l)'

c. Fig. 1h legend, 5th line: ... comparisons for panels '(g-j) -> (h-k)'

d. Fig. 1i-k legend, 1st line: ... panels '(f-g)' -> '(g-h)'

e. Fig. 1i legend, 2nd line: '(h-j)' -> '(i-k)'

f. Fig. 1m legend, 1st line: ... panels '(g-j)' -> '(h-k)'

g. Fig. 1m legend, 1st line: ... circles in panel '(k)' -> '(l)'

h. Line 1007, "Figure 1d", presumably this is referring to Figure 1e

Reviewer #2 (Remarks to the Author):

This manuscript by Kira and colleagues describes a combination of experiment and analysis of neural dynamics related to cortical processing during flexible decision-making. The results are intriguing, the experimental approach is sound, and the analysis appears rigorous, but clarity in the writing and in data presentation limits the impact. Furthermore, while the authors claim that their results support a particular model of cortical information processing (namely that missed selective neurons in RSC support feasible decision-making), it is not clear what models their experiments rule out. This is a key question. Demonstrating more clearly that their experiments rule out existing or alternative models would improve the manuscript. More detailed comments are provided below.

1) Intellectually the aspect of the work I found most interesting was the combination of theory/simulation (supplementary figures 7) and analysis of experimental data (figure 5, supplementary figure 5 and 6) demonstrating the efficiency of mixed representations in RSC for encoding reward location. This observation is particularly satisfying when combined with their inactivation experiments. This aspect of the manuscript could be strengthened by the addition of a clear mechanistic model for how mixed selectivity neurons in posterior cortex drive decisions. Such a model would allow quantitative predictions that could be evaluated with future experiments.

2) Several statements and claims throughout the text are either unclear or not supported by the data. They interfere with the reading of the paper and should be addressed. Some of these sentences are not critical to the main point of the paper, so removing or altering them would have no significant effect on the results. Examples include, but are not limited to the following:

A. Line 32 - "Together, these characteristics imply that specific neural mechanisms exist for rapid flexibility over times as short as seconds, which are too brief for substantial neural circuit plasticity."

This statement is vague, do the authors mean to suggest that flexible decision-making does not involve moment-to-moment changes in synaptic plasticity? What is the evidence for this? This is an example of a claim which if removed would not impact the manuscript.

B. Line 38 - "The flexibility of decision-making has been investigated in experimental paradigms that do not involve spatial navigation."

This statement is a bit unclear. Do the authors mean that some studies of flexible decision-making have been performed in a non-spatial context or that ALL studies of flexible decision-making have been performed in a non-spatial context. Given the authors' definition of flexible decision-making it seems that the latter is not accurate. There are many papers using, for example, delayed non-match to sample T-mazes and alternation T-maze - based tasks to study the influence of working memory on decision-making. The authors should not ignore this literature.

C. Line 81 - "We discovered neurons that mix short-term memory and visual information, and these neurons were present in most parts of posterior cortex, even in V1. Surprisingly, RSC had the highest density of these neurons, whereas these cells were sparsest in PPC, with a near absence in anterior PPC. These neurons formed an efficient population code, which appeared to be critical for accurate decisions because their activity was more informative when the mouse made correct decisions compared to errors."

Do the authors mean to suggest that because these neurons represented task variables with higher fidelity on correct trials, they support decision-making? I'm not sure I follow this logic. Does this argument require the assumption that correlation implies causation? Throughout the paper claims about the role of cortical neurons in flexible decision making are made based on the response properties of RSC neurons during behavior. The authors should be more careful and precise about the language used when inferring function based on recorded dynamics.

D. Line 244 - "The GLM poorly predicted neural activity in M1 and M2 (Fig. 2I) although the mice showed similar performance during imaging sessions for each area (Supplementary Fig. 4a-b), suggesting these areas may be less informative for solving the task compared to posterior cortical areas."

I have a logical problem with the idea that just because the model does not predict the activity in cells, that those cells are somehow less likely to be involved in the behavior. The optogenetic experiments provide information about this causal relationship. However, if the authors' model does not predict the activity of a cell, perhaps this tells you more about the parameterization of the model rather than the role of these cells in driving behavior. For example, is it possible that different non-linearities exist between task variables and activity of cells in M1 and M2 and that the diminished performance reflects a poor choice of non-linearity?

E. Line 293 - "The distribution of information in RSC cells in the test segment was approximately uniform between sample cue selective, mixed selective, and test cue selective cells, which was markedly different from the distribution in V1...Together, the results show that RSC has approximately equal mixing of information, the largest fraction of cells with mixed information, and the highest XOR information, indicating that RSC could be a central area for mixing memory information and visual information."

Do the neural dynamics suggest it is RSC plays a central role or the inactivation experiments? What models do these data rule out?

F. Line 347 - "For trials with identical sensory cues, if cells have higher information about the reward direction (XOR) on correct trials than on error trials, then such an observation would support the notion that the information in those cells was used to guide accurate choices."

This is another example of the issue highlighted in C: Does this argument require the assumption that correlation implies causation?

G. Line 372 - "Together, these observations support the idea that aberrant activity of mixed selectivity cells across a distributed set of areas, and most strongly in RSC, contributes to incorrect choices."

These observations are also consistent with a variety of other models. Does this data rule out any models?

3. Figures and figure legends could be improved for clarity. Revising figure titles to state the main finding or discovery made rather than the experiment performed would help. For example, in figure 1: "Optogenetics screen for cortical areas involved in a flexible navigation task", the authors might consider a title that indicates the main finding of the experiment such as "Optogenetic inactivation screen reveals the key role of RSC in flexible navigation". Many figure titles could be altered in this way and such changes will help the reader absorb the main idea information presented in the figure more readily.

In addition to the titles both the figures and the figure captions themselves should be revised for clarity. For example, in Figure 1 – the organization and order of panels g-m is confusing. In particular, Panel l seems to be the bridge between g and h but comes well after h and is unlabeled. Perhaps it could be labeled, relocated or placed in supplemental. Another example is the caption for Figure 5a. The caption reads “Mixed selectivity cells encode the identity of the sample cue, test cue, and reward direction (XOR) (purple rectangle) ...”. But what is shown is a schematic of the approach for separating recorded neurons into two subpopulations (mixed and pure” based on their response properties). Stating this clearly would help the reader understand the purpose of this panel rather than forcing them to deduce it.

Reviewer #3 (Remarks to the Author):

Kira et al. investigated the way mice integrate information about visual stimuli and remembered cues during a delayed match-to-sample task. They study this task in an interesting spatial navigation context in which the animals navigate in virtual reality, with the first "sample" cue and second "test" cue separated in both time and (virtual) space. Once the test cue is presented, the animals are required to turn towards the side that matches the sample cue.

To investigate the cortical regions involved in this task, they iteratively inhibited different areas using optogenetics, and determined which such perturbations most impaired task performance: retrosplenial cortex (RSC) and V1 stood out as quite important, while frontal areas were surprisingly less important. While I love the idea of causally testing the importance of each brain area in this way, I have some practical concerns about these experiments that I discuss further below.

Thereafter, the authors probed the structure of the neural code in each relevant brain area, using neural activities recorded by Ca²⁺ imaging. They found that neurons in RSC in particular often had mixed selectivity -- responding nonlinearly to both the sample cue and the test cue, in a fashion that provided information needed to perform the delayed match-to-sample task. This is referred to as XOR information since the reward matrix for the task matches the logical XOR operation.

Finally, the authors speculated that this mixed selectivity was critical in generating an efficient neural code to support the mouse's ability to perform the task.

This is an interesting study with some exciting findings. I have some fairly substantial concerns that I outline below. I am hopeful that the authors could address my concerns in a suitable revision.

My specific comments are as follows:

1) While lots of researchers have studied neural computations during delayed match-to-sample tasks, a key point of novelty for this study is that this is done in the context of navigation. While I agree with the authors that navigation is an important naturalistic behavior in which to probe context-dependent neural computations (like delayed-match-to-sample), I am not convinced that this task really has much to do with navigation, because the experimenters decoupled the animal's motion through the virtual space from their locomotion on the ball, up until the time of their decision. This means that the task that the mouse needs to perform can be reduced to: "after the delay, turn the ball in the direction that matches the remembered sample cue". Put in this way, it is unclear to me how the navigation aspect really affects the task. Overall, I would find the novelty of this aspect of the study to be more persuasive if the authors could explain better why the navigation aspect is critical to the mouse's ability to do the task. Similarly, I would be very curious to know the roles of hippocampus and/or superior colliculus in this task, given their known roles in navigation/memory, and in orienting movements.

2) I really like the idea behind using optogenetic inhibition experiments to causally test the necessity of each brain area! The practical implementation, however, comes with some pretty major limitations, only some of which are discussed by the authors. While they do convincingly show that V1 and RSC are needed for the task (at least during the sample and test period), I had some major questions about possible false negatives, among other things:

a) In their optogenetic experiments, do the authors silence neurons all the way down to layer 5 (or 6), or is the silencing restricted to more superficial cortical laminae? This information is somewhat available in Fig. S2, but without knowing the cortical thickness of each brain area, and the reference against which they are measuring depth (which would define the location of "zero depth"), it's hard for me to map their cortical depth values to cortical laminae. Those data (the depth to which neurons are inhibited by the laser) are also absent for the frontal areas for which I think they might be most important in ruling out false negatives from the silencing experiment. I explain this last point below.

Given the different projection targets of pyramidal cells in the different laminae, different results may be obtained if excitatory neurons in only some laminae were silenced vs if they were all silenced, down to and including L6. For this reason, it is hard to say from the data presented whether frontal areas are really not necessary for the task (as implied in Fig. 1ijk, for example), or just whether the superficial areas of those areas are not necessary for the task, while the deeper neurons (potentially less affected by the optogenetic manipulation in these experiments) might still be necessary.

b) In what inhibitory neuron types is the Chr2 expressed in the VGAT-Chr2 mouse? Perhaps it is all of them, in which case the authors should emphasize this point in their methods. Given the different roles played by VIP, PV, SST, etc., interneurons, if only some of interneuron types express the Chr2, I would worry that the effects of the optogenetic perturbation might be more complicated than "just" silencing excitatory cells. This would make it harder to interpret the data in Fig. 1 that aim to show the necessity of each brain area.

Given what's in Fig. S2, it looks like spiking activity in RSC is completely shut off down to ~700um of depth (although I do not know what cortical layer that corresponds to), whereas in M2, for example, it is strongly attenuated but not quite shut off. This poses the question of what these distorted activity patterns look like, and that will depend on the types of interneurons driven by the laser.

This means that, in interpreting the M2 data from Fig. 1, rather than concluding that "M2 has only a modest role in this task", one could potentially conclude that "M2's contribution to this task is surprisingly robust against the manipulation performed here".

3) I would remove Fig. 1C since it is uninformative and confusing. It took me a while to realize you had forced the VR headings and positions to take on those specific values, and during that period I was puzzled by why there was so little behavioral variability. This is a minor point of course, but I think that removing this panel could help with the clarity.

4) I liked that you checked whether the wheel motion during the delay period could carry a mnemonic encoding of the sample cue. If you have video recordings of the mouse doing the task, I'd suggest also looking at those to see if anything obvious pops out at you (e.g., tail orientation). There are in principle a huge number of possible ways the mouse could develop a mnemonic for the sample cue, and I don't expect you to exhaustively probe those possibilities. But -- if you haven't done this already -- watching some video from each animal (since they may develop their own unique mnemonics) doing the task could help to spot any obvious mnemonic behaviors beyond the wheel motion.

5) Line 202-203, you mention deconvolving the calcium traces. This is notoriously tricky, with isolated spikes often being missed (e.g., see recent publications including Huang et al. eLife 2021). I think the authors should discuss this limitation, and help readers like me to understand whether or not it severely confounds their analyses. (Hopefully the answer is "no it does not", for some good reasons).

6) Line 234: I am curious about why the authors used the FDE measure instead of something more common like fraction explained variance (FEV). Unless there's a good reason to choose this non-standard measure, I suggest using a more common measure (like FEV). That would make it a bit easier

for readers to follow since they wouldn't have to figure out what this new measure is, and would help readers use their prior knowledge to more easily interpret the values you present.

7) Fig. 2 EFG, HIJ: could you please make it more clear on the figure (maybe next to the cell ID numbers) what brain areas each neuron comes from? I initially thought these were all from the same area, which of course they are not.

8) Given the discussion of correlations later in the paper, it is worth asking why the GLM has no coupling between cells. Including that coupling in the GLM would facilitate estimation of log-likelihood ratios for *populations* of cells. Instead, the present analysis estimates the logLR for each cell and then presents it as though it describes the population. I am sure it's unintentional, but I found this part a little bit misleading. E.g., in lines 287-288, the authors state that "RSC had significantly larger XOR information than V1. . .", but based on their analysis, I think a more accurate statement would be "On average, individual neurons in RSC had significantly larger XOR information than did individual neurons in V1". Making claims about the information in the population (rather than the single-neuron properties) requires consideration of the joint distributions, which could be achieved by using GLMs with coupling between neurons.

9) Fig. 3ab: could you add to your caption or text a bit more information to explain what the breaks are along the time axis. I.e., "we show the first and last second of activity during the sample period, but not the middle portions", and similar statements for what is shown during the delay and test periods. It took me a fair while to figure out why there were breaks in the logLR vs time plots.

10) Can the authors quantify the statistical significance of each neuron's logLR values? This is important for understanding the strength of the claims about single-neuron information representations from those logLR values. This especially comes up on line 328-329, where a cutoff of $\log LR > 0.01$ is used to define cells with "appreciable" XOR information. Without knowing the uncertainties in the logLR values, it is hard to know whether this is a reasonable cutoff. A better criterion might be to test the significance of each neuron's XOR logLR against zero, and use $p < 0.05$ (or $p < 0.01$) for the cutoff.

11) Relatedly, in Fig. 3HI and 4HI, the measurement of polar angles seems problematic unless at least one of the two logLR values is statistically significantly non-zero.

12) I really liked the analysis in Fig. 4K! I kept wondering earlier in the paper whether all of the effects reduced to the mouse just forgetting the sample cue on the incorrect trials, and I thought this analysis was a great way to show that there's more going on.

13) Lines 409-410: why show the mutual information between decoder and truth, and not just the accuracy of the decoder? The two are monotonically related of course, but the accuracy is probably much more easily understood by most of your readers.

14) In Fig. 5B, why not also show decoding accuracy for the combination of both types of cells, in addition to just the mixed selectivity and the pure selectivity ones? The brain has all of these cells, and I was curious about how the information from the two types of cells combines. I.e., if the full population including both types of cells has about the same information as just the sub-population of mixed selectivity cells, that tells me that the mixed ones really dominate the representation. But maybe the two cell types together have more information than the sum of the information presented by each cell type in isolation. This can happen depending on the correlations between cells of the two types. Adding this comparison would help support the assertion on line 464, that the mixed selectivity cells really are the key to the animal being able to perform the task.

We thank the reviewers for providing constructive feedback and helpful suggestions on how to improve the manuscript. We are pleased the reviewers found the work to be of interest. We have worked hard to incorporate the reviewers' suggestions for new experiments, data analyses, and text revisions. We feel our manuscript is greatly improved as a result.

Reviewer #1 (Remarks to the Author)

Kira et al examined cortical areas and neural activity supporting flexible decision-making in a delayed match-to-sample task in which mice running in corridor compared a sample visual cue with a test cue to reach a navigation decision. An optogenetic screen identified posterior regions of cortex, including PPC, V1, and RSC, to be important for task performance. Population calcium imaging revealed neurons coding specific conjunctions of sample and testing cues. This nonlinear mixed selectivity is most prevalent in RSC and it is weakened during incorrect decisions. The authors proposed that this nonlinear mixed selectivity provides an efficient population code for flexible decisions.

Overall, this is a carefully done study that combines several cutting-edge approaches in a difficult decision-making task for mice that advances new ideas on how flexible decisions could be computed. The notion that neurons coding specific conjunctions of sample and test cues (XOR selectivity) provide an efficient population code is intriguing. And the observation that such selectivity is diminished before erroneous decision is interesting. However, a few conceptual issues around how mixed selectivity could be interpreted and how it produces flexible decision should be better clarified. I additionally have a few technical comments that could be addressed through additional analysis.

We thank the reviewer for the positive and constructive feedback.

Major

1. The involvement of posterior cortical regions in flexible decision should be more carefully interpreted or at least clarified. There seems to be several incongruencies.

a. In several places in the text, the behavioral deficits caused by inhibiting posterior cortical regions are interpreted as those regions being involved in combining test cue with memory of sample cue for flexible decision (e.g. line 162-163 and 181). However, the behavioral deficits seem to be more (or at least equally) consistent with a deficit in visual perception which affects the detection of test cue. For example, large effects are also seen when these cortical regions are inhibited during the sample segment, which does not require combining memory and visual signals.

We agree that our optogenetics results are consistent with roles for posterior cortical areas in visual perception and processing of visual signals. We observed an appreciable decrease in behavioral performance due to inhibition during the sample segment. It thus seems possible that some of the decrease in behavioral performance in the test segment could relate to visual

perception. However, if these areas were mostly or only responsible for processing visual signals, then we might expect the same level of behavioral deficits with inhibition during the sample segment and test segment. Instead, we observe a larger decrease in behavioral performance with inhibition during the test segment (Fig. 1j, l). This result suggests that the inhibition during the test segment may be affecting functions beyond visual perception. The primary additional function during the test segment is combining the visual information of the test cue with the memory of the sample cue, which is why we suggest that the inhibition may relate to this function. However, there is the potential that visual processing requirements differ between the sample segment and test segment, which could potentially contribute to the differences in inhibition results between these segments. We have therefore clarified our interpretation about this point in the text (LL 194-201). Notably we clarified that the inhibition experiments were used to identify the cortical areas necessary for accurate performance of the task and not to identify the specific functional roles of these areas in the task (e.g., visual perception vs. decision-making).

b. The authors propose that mixed selectivity for specific conjunctions of sample and test cues is important for flexible decision. However, in all the candidate cortical regions, the mixed signal is very sparse overall. Rather, the largest modulation appears to be visual modulation by the sample and test cues (Fig 3a-f), which seems to be more consistent with a role in visual processing.

We agree with the reviewer that there are other signals present in these areas beyond the mixed selectivity. The goal for our study was to understand how cortex contributes to flexible decision-making. The mixed selectivity cells seem to be the best candidate based on the approaches we have taken in the areas we investigated. We do not intend to imply that this is the only function of these areas. However, we believe that the mixed cells are a good population to highlight when focusing on the topic of flexible decision-making. The reasons include because these cells were the ones that most strongly encoded the reward direction (XOR information), were robustly active on correct trials but not error trials, and were only present in trained mice and not naïve mice (see new experiments described in reply to point 2a of this Reviewer and reported in new Figure 5). We have added new text that clarifies this point. Specifically, we mention that there exist other signals in these areas that could be important for solving the task, including for visual processing. We also now explain that the reason we focused on the mixed selectivity cells, even if they are a sparse population, is because they are the best candidates for the computation of interest in this study. See LL 370-375 for the revised text. In future studies, it will be of interest to more thoroughly characterize the other signals present in these areas, but we feel a complete characterization of all signals is beyond the scope of this study.

c. Behavior deficits by site-specific optogenetic inhibitions was stronger in V1 and PPC rather than RSC (Fig. 1g-k). However, mixed selectivity is concentrated in RSC and less in V1 and PPC (Figs. 3-5). How to reconcile this mismatch between manipulation effects and neuronal selectivity across the brain regions?

This is a good point that was not discussed clearly in our first submission. It is true that the optogenetic inhibition effects are larger in V1 and PPC than RSC when considering the single spot

inhibition experiments. When we inactivated a larger extent of RSC, using three spots, we found similar amplitude, or even larger, behavioral effects when targeting RSC compared to V1 and PPC. It therefore appears that the smaller effects are due, at least in part, to not inhibiting all or most of RSC with a single spot. Aside from this point, however, there is another issue. The inhibition experiments can affect any number of neural processes that are important for solving the task, including perceiving the visual cues, remembering the cues, combining the memory and visual signals, and executing a turn. From our inhibition experiments, we cannot determine which of these processes are affected in each of the areas because all of them will result in a decrease in behavioral performance. It is possible that large inhibition effects in V1 are due more to V1's role in processing visual stimuli than in the mixing operation. Similarly, it is possible that PPC's role is more in guiding a behavioral output than in the mixing operation. In this case, we might expect similar or larger behavioral deficits when V1 and PPC are inhibited than when RSC is inhibited, but for reasons not entirely related to the mixed selectivity cells. Stated more concisely, our inhibition experiments cannot determine which cells are the critical ones for the task and likely affect multiple processes including the mixing operation and others. We have therefore added clarifying text to that section of the results. We now state more clearly that the optogenetics experiments are aimed solely at identifying cortical areas that are necessary for performance of the task. We also state that these experiments cannot, and are not intended to, identify which computations (mixing or visual perception or others) are critical in each of the inhibited areas. See LL 194-201 in the revised text.

d. Expanded inhibition in RSC yielded larger behavioral effects (Fig 1l). However, V1 is also a large area, which exhibits large effects even with restricted inhibition. Expanded V1 inhibition should also be provided to fairly compare to the more dramatic effect of expanded RSC inhibition.

We agree that the comparison of three spots in RSC and one spot in V1 is not an apples-to-apples comparison. We reasoned that inhibiting all of V1 would prevent visual processing and lead to a full behavioral deficit. Indeed, in other experiments in our lab for different behavioral tasks, when we have optogenetically inhibited most of V1, we find that the mouse runs in a manner consistent with it not being able to see the virtual maze on the screen. The mouse spins and runs randomly without being able to progress through the maze. Each of these optogenetics experiments involves a fair amount of work because we can only inhibit on a small fraction of trials (~10% of trials) to avoid causing the mouse to disengage from the task. Thus, we need a lot of sessions and mice to accumulate enough samples to reach sufficient statistical power. We therefore chose not to perform the proposed experiment as we felt it might not be the most informative one given prior expectations and these unpublished results in other tasks from our lab. We did not feel these same issues applied to RSC and thus tried the multiple spot inhibition of RSC. Indeed, mice could still run and navigate through the virtual maze with most of RSC inhibited, but their ability to make the correct choices decreased to near-chance levels. We have now explained this rationale more clearly. As mentioned above, we have also more clearly stated that our goal of the inhibition experiments is to identify cortical areas that are necessary for the task, and our goal is not to interpret the specific functions of those areas in the task. We use the calcium imaging experiments for the latter. In addition, we have moved the text related to the expanded

inhibition of RSC (LL433-444). In this revised text, we now state that we were surprised to find an enrichment of mixed selectivity cells in RSC given its single-spot inhibition effects were not as large as for V1 and PPC. We therefore reasoned that we wanted to test RSC's necessity by inhibiting all or most of it with multiple spots. This rationale makes it clearer why we are performing these experiments and why we did not feel the need to do expanded inhibition of V1.

2) The finding of nonlinear mix selectivity and its correlation with performance is interesting (i.e. higher selectivity in correct trials), which leads to the interesting proposal that the mixed selectivity has a causal link to the decision. However, a few conceptual issues regarding the function of mixed selectivity could be better clarified.

a. The text suggests that the mixed selectivity reflects “mixing” of visual and memory signals for the purpose of generating the decision. However, it is not clear if the responses reflex a mixing operation, or simply reflects a visual response modulated by the context of the preceding stimulus. For example, would mixed selectivity be encountered in these regions if mice are not engaged in decision-making behavior?

We thank the reviewer for raising this alternative possibility that we had not considered. We performed a new experiment to directly test whether the mixed selectivity reflects a visual response. We did calcium imaging in a cohort of “naïve” mice that were not trained to perform the delayed match-to-sample task. These mice ran through the identical virtual maze and saw the identical visual cues to the mice trained on the delayed match-to-sample task. However, they were not trained to make left-right choices based on the sample and test cues. Instead, they simply got a reward by reaching the end of the T-stem. Importantly, these mice traversed the virtual maze with similar running speeds and trial durations to the mice trained on the delayed match-to-sample task. If the mixed selectivity reflects a visual signal, such as due to visual adaptation, then we would expect to see similar levels of mixed selectivity in the naïve and trained mice. Alternatively, if the mixed selectivity truly reflects a mixing of visual and memory signals, then we would expect to see higher mixed selectivity in the trained mice than in the naïve mice. We analyzed the neural activity from the naïve mice in the same way as we did for the trained mice. The results from these new experiments revealed much weaker mixed selectivity in the naïve mice than in the trained mice. Therefore, this new experiment rules out the idea that the mixed selectivity is due mostly to visual responses, such as visual adaptation. Instead, it indicates that the mixed selectivity indeed represents the mixing of visual and memory signals and emerges through the course of learning the delayed match-to-sample task. These results are now reported in a new main figure (Fig. 5).

b. Conceptually, what is the relationship of nonlinear mixed selectivity to flexible decision? For a population of neurons with nonlinear mixed selectivity, decision in every instance (i.e. specific combinations of sample and test cues) would depend on different neuronal populations (coding specific conjunctions of sample and test cues). This is very different from the notion of flexible decision in the form of a “match” vs. “nonmatch” signal irrespective of the details of the stimulus.

Can the authors clarify how mixed selectivity can give rise to flexible decisions if the selectivity is so specific to the exact combination of task elements? For example, if a different instance of stimulus is encountered or if a different motor response is required, the coding of decision would change.

c. The nonlinear mixed selectivity does not actually represent a code for the decision itself, as far as I can tell. It, however, could serve as a basis set from which decision could potentially be constructed.

We thank the reviewer for raising this interesting point. Our initial prediction was that we would find a good number of cells that signaled something like “match” or “nonmatch”, which could mediate the flexibility of the task. In the terminology used in our paper, these cells would have been cells with pure XOR selectivity. Cells with pure XOR selectivity do not have sample cue selectivity or test cue selectivity, by task design. We were surprised that we observed very few cells with this type of selectivity. This caused us to re-evaluate how the task was being solved. The most striking thing that stood out was the prevalence of single-trial-type selective cells. We were not initially expecting or looking for these types of cells, but their prevalence relative to pure XOR selective cells indicated a way in which the task was solved. We view these cells as forming a basis set for all the different task conditions (four in our case). The mouse can then use this basis set to inform the correct choice on a given trial. This seems like an interesting and perfectly suitable solution for the flexibility required for our task, as we define it. We refer to flexibility as the ability to make a different action in response to a sensory cue depending on information stored in memory. The mice can do this by using this basis set of single-trial-type selective cells. To our knowledge, our work is one of the first descriptions of this type of solution to a flexible decision-making task.

It is an interesting question of how this solution would generalize to new cues and associations. In the case of “match” cells or pure XOR selective cells, new cues would utilize these cells to perform the task in these new cases. In the basis set solution that we found in our data, new cues would have to be learned and associated to form a new set of single-trial-type selective cells. In future work, we hope to understand how this type of solution may be used for generalization to new conditions. It is possible that if the cues are varied from day to day with new cues being introduced frequently then the brain would come to a different solution using “match” cells. Here the mice rehearsed the same sets of cues for extended periods, in which case the solution to create the basis set for these trial types might be a better and more efficient solution. Therefore, there could be a tradeoff between a solution that allows generalization and one that is best for frequently experienced situations. In addition, it’s possible that the basis set solution is valuable for a navigation case. We mention in the discussion that the single-trial-type selectivity could relate to navigation signals about locations, where each trial type might be considered to be a different maze/location by the mouse. In this case, where the task has a concrete/physical manifestation instead of being a purely abstract task (like looking at shapes on a screen), it might be more natural for the mouse to adopt the basis set solution to represent each environment as unique instead of forming an abstract concept.

We have revised the text to discuss these possibilities in more depth (LL 591-609). We note the expectation of “match” type cells in the introduction as an alternative that we might expect beforehand. In the discussion, we have now added a paragraph that talks about the relationship between the “match” type solution and the basis set type solution. We also discuss the limitation of generalization for the basis set solution that we report and how future work is needed to test how generalization is performed in these types of navigation tasks.

3) A few technical comments on the quantification of selectivity.

a. Most quantifications of mixed selectivity are based on 1 second of the test segment. Based on the running pattern from the test segment, mice have completed decision 500ms after the test cue onset (Fig S1). During the second 500ms of the test segment, different running patterns could also affect the responses of neurons, which impacts the calculation of selectivity. For these reasons, it is important to confirm that key results on mixed selectivity (Fig 3 & 4) stand if analysis is kept to the first 500 ms of the test segment.

Thank you for raising this point. We have now repeated the key analyses limited to the first 500 ms of the test segment. We report these results in a new supplementary figure (Supplementary Fig. 8). The results are largely consistent with what we observed when analyzing the first 1 s of the test segment. Our major conclusions would be the same using the analysis from the first 0.5 or 1 s of the test segment.

An important point to consider is that while the mouse’s running pattern started to indicate the choice around 500 ms after the onset of the test cue onset, the running evolved over the next 200-300 ms until its decoding accuracy reached ~80% (Supplementary Fig. 5c). Note that this is different than ballistic movements, such as licks or eye saccades, that do not have a long temporal evolution. The time evolution of the running pattern varied across sessions, and on many trials it took longer than 500 ms for the mouse to report its choice (Supplementary Figure 1i). Therefore, while mice completed reporting their decision by 500 ms on some trials, they spent a longer time to report their decision on a large fraction of trials. We therefore prefer to focus on the first 1 s for the analyses to take into account some of these variabilities.

b. The text suggests XOR selective neurons exhibit “single-trial-type selectivity”, i.e. only responding to a single combination of stimulus and test cues. This claim is based on a sparsity index of 3. It is somewhat counterintuitive what this metric reflects. Will the authors consider also including a simpler analysis to show this selectivity, for example in the form of average activities rank ordered by trial-type preference?

We thank the reviewer for the suggestion to provide a simpler and more intuitive metric. We have replaced the sparsity index with a new metric and added new graphics that illustrate the single-trial-type selectivity (Figs. 3j, 4j, 5j, Supplementary Figs. 5g-i, 8g). In the new method, we compute the mean activity for each of the four trial types and normalize these values by the sum of these means across all four trial types. Therefore, if neural activity is similar on each trial type,

each trial type would have a value of 0.25. Instead, if the cell is active on only one trial type, it would have a value of 1. We now plot these values for each of the four trial types. In the new plots, we refer to the trial type with the highest value as the trial type with the preferred sample cue and preferred test cue. We then show the values for the other three trial types with the preferred or unpreferred sample and test cues.

c. Related to point b, the authors find that neurons coding both stimulus and test cues also preferentially encode XOR (Fig 3j). It would be beneficial to also show simpler analysis to help readers understand this relationship. Perhaps by showing more single-neuron examples (as in Fig 2e-j) or average responses of different types of neuronal populations.

We thank the reviewer for this suggestion. We have added a new supplementary figure (Supplementary Fig. 3) that shows the selectivities of the example cells shown in Figure 2. We made separate plots to show each cell's average activity for the different sample cues, the different test cues, and the different XOR values. These plots show that single-trial-type selective cells have different levels of activity for the trials with different sample cues, different test cues, and different XOR values. In contrast, the cells with pure selectivity do not show differences across all these sets of variables. We think these new plots are helpful in illustrating the types of information carried by the single-trial-type selective cells.

4) Some parts of the Introduction and Abstract should be better clarified.

a. Several places in the text refers to “behavioral goal in short-term memory”. It seems the short-term memory in the delayed match-to-sample task is about the identity of the stimulus cue. What is meant by behavioral goal here?

We apologize for the lack of clarity in our terminology. We were using this terminology to refer to the sample cue stored in short-term memory. We have removed the use of “behavioral goal” and now refer to this directly as the memory of the sample cue. We hope the revised wording in the introduction is more straightforward and clearer.

b. A few previous studies are not accurately cited (line 38-49):

b-i. Wu et al and Condylis et al both compared decision signals in multiple brain areas during similar delayed match-to-sample tasks (but not navigation decisions).

We have updated how we refer to these citations to make it clear that our work is not the first to compare multiple brain areas during a delayed match-to-sample task. We feel the difference between our work and these studies is that we performed a relatively unbiased screen across cortical space using the optogenetics experiments. We have clarified this difference between studies in the main text.

b-ii. Duan et al also studies trial-to-trial switch of task rules, not slow changes in sensorimotor association.

We have corrected the citation for Duan et al.

c. Several places in the text focus on “rapidly switch[ing] sensorimotor associations”. This is counterintuitive because the coding scheme of mixed selectivity relies on storing specific combination of sample and test cues. There is no mechanism supporting “switching” associations.

We apologize for the confusing wording here. We were referring to “switching sensorimotor associations” from a behavioral viewpoint. In terms of behavior, the mouse makes a different action in response to the same visual cue depending on information stored in memory. Behaviorally, this could be considered as switching associations. We agree, however, that this is confusing when thinking about the neural mechanism we have reported. We have therefore updated the wording in the introduction to make this point clearer and to remove the usage of the term “switching sensorimotor associations”.

Minor:

1) Can the authors discuss potential differences between this study and the Pinto et al study? Pinto et al found strong involvement of frontal cortex in addition to posterior cortex. Here the frontal cortical regions are surprisingly dispensable.

We are not sure of the reason for the difference between our results and Pinto et al., but we have speculative ideas about what could contribute. In Pinto et al, mice were allowed to change and adjust their heading direction in the virtual maze as they ran down the maze. This corresponds to virtual changes in heading orientation. In contrast, in our task, we fixed the virtual (and real) heading angles as the mouse ran down the virtual T-stem. We did this to make sure the visual scene was identical from trial-to-trial and not altered by the behavior of the mouse. One possibility is that frontal areas of the cortex are important for these adjustments of heading direction. We have added these ideas, and a longer discussion about our results in frontal cortex, to the discussion section (LL 659-674).

2) It seems mice undergoing expanded RSC inhibition exhibit faster running overall, even in control trials. Why is this the case?

We are unsure of the reason for this difference. We notice some cohort-to-cohort variability in running speed. It is possible that this difference is just normal variability. The mice in this cohort

were also slightly larger in physical size (higher body weight) than mice in other cohorts because they were slightly older. It is possible this size difference led to faster running. We now mention this in the Methods (LL 720-722).

3) Statistical tests in Fig 5 require better description. Are these based on bootstrap? For the statement: “The difference between correct and error in mixed selective cells was significantly larger than that for pure selective cells in V1, RSC, and MM”, this should ideally be tested for significant interaction between 2 factors, correct vs. error trials and mixed vs. pure selective cells.

We apologize that the statistical method was not clearly described in the caption. Yes, the statistics are based on bootstrap. We bootstrapped cells in each session and analyzed whether the difference in XOR mutual information between correct and error trials was different between mixed selectivity cells and pure selectivity cells, using a two-tailed test. Although this was briefly described in Methods (LL 1061-1062), we have now added more description in the caption (LL 1998-1999). This bootstrap test is conceptually equivalent to a two-way ANOVA that could have been performed if the data had a normal distribution and was identical across conditions, but it is not general and non-parametric and it does not make the above assumptions. Thus, our analyses did test the interaction between the two factors, as suggested.

4) Are the imaged V1 cells from monocular V1, binocular V1, or both? In general, the medial and lateral parts of V1 are monocular and binocular, respectively. The two types of V1 cells could be modulated differently by the test cue where monocular V1 is responding to white vs. black stimuli in the contralateral visual field, leading to higher test cue selectivity, whereas binocular V1 and RSC neurons might be modulated by ipsilateral visual field. If imaging covered both regions of V1, the authors might consider separately analyzing monocular and binocular V1 data and compare with RSC, specifically for Fig 3g-i.

We apologize that these points were not described. We recorded from the anterior medial part of V1 in the left hemisphere, which corresponds to the lower right monocular visual field. This part of the visual field largely overlapped with the right half of the test cue. We now mention this point in the Results (L 204) and Methods (LL 983-986).

5) Even though GLM poorly predicted neural activity in M1 and M2 (Fig. 2I), it will be informative to show the M1/M2 data in a similar way to RSC, V1, and PPC as in Figs 3-5, perhaps in the supplemental.

We have added a new supplementary figure that reports the M1/M2 data in a manner similar to how we reported the imaging from RSC, V1, and PPC (Supplementary Fig. 6). This new supplementary figure shows that M1/M2 do not have much information about the sample cue, test cue, and XOR, but they do have information about the mouse’s choice late in the trial when the mouse is turning.

R1-m6) A few typos:

- a. Fig. 1h legend, 2nd line: ... computed from the data in panel '(f)' -> '(g)'
- b. Fig. 1h legend, 2nd line: ..., as marked in panel '(k)' -> '(l)'
- c. Fig. 1h legend, 5th line: ... comparisons for panels '(g-j) -> '(h-k)'
- d. Fig. 1i-k legend, 1st line: ... panels '(f-g)' -> '(g-h)'
- e. Fig. 1i legend, 2nd line: '(h-j)' -> '(i-k)'
- f. Fig. 1m legend, 1st line: ... panels '(g-j)' -> '(h-k)'
- g. Fig. 1m legend, 1st line: ... circles in panel '(k)' -> '(l)'
- h. Line 1007, "Figure 1d", presumably this is referring to Figure 1e

We have corrected all these typos. Thank you for catching them.

Reviewer #2 (Remarks to the Author)

This manuscript by Kira and colleagues describes a combination of experiment and analysis of neural dynamics related to cortical processing during flexible decision-making. The results are intriguing, the experimental approach is sound, and the analysis appears rigorous, but clarity in the writing and in data presentation limits the impact. Furthermore, while the authors claim that their results support a particular model of cortical information processing (namely that mixed selective neurons in RSC support feasible decision-making), it is not clear what models their experiments rule out. This is a key question. Demonstrating more clearly that their experiments rule out existing or alternative models would improve the manuscript. More detailed comments are provided below.

We thank the reviewer for the feedback and constructive suggestions.

We have now added text in the introduction and discussion that aims to highlight the alternative models that we are testing and the ones that are ruled out by our results. These types of models fall into two categories of topics. The first regards what brain areas are involved in flexible decision-making during spatial navigation. This topic has not been investigated in depth previously, to our knowledge. Regarding which areas are involved, there are several possibilities. We present evidence from the literature that suggests that flexibility could be mediated by frontal areas of cortex, posterior parietal cortex, or retrosplenial cortex. It is possible that one of these areas would stand out as the key area. Alternatively, it is possible that the underlying processes are more distributed and span a wider range of cortex. We found that the mixed selectivity cells and the inactivation effects are distributed across posterior parts of cortex, including V1, RSC, and PPC. These results therefore rule out the model in which flexible navigation decisions are mostly localized to one region. They also rule out the model in which frontal areas, which are commonly thought to mediate flexibility, are the key areas. Finally, they add the surprising finding that V1 is involved in this process. We now highlight these findings in more depth in the introduction and discussion.

The second topic regards how the flexibility arises in terms of neural mechanism. Our prediction beforehand was that this task could be solved by a collection of cells with pure selectivity. We thought it might be possible to see a collection of cells with pure sample cue selectivity, others with pure test cue selectivity, others with pure XOR selectivity or choice selectivity (signifying something similar to a “match” in the delayed match-to-sample task). In this case, the pure selective cells with sensory information and those with memory information could converge onto the choice cells to drive the behavior. This type of solution was noted by Reviewer 1 and is mentioned in previous studies of delayed match-to-sample tasks. Instead, we found, unexpectedly, a small number of pure selective cells for XOR (“match”) and a prevalence of single-trial-type selectivity cells. These single-trial-type selectivity cells can serve as a basis set (or look-up table) for each trial type to guide the choice of which direction to turn. We were surprised by this finding because it creates a specific, rather than general, solution to the delayed match-to-sample task, as noted by Reviewer 1. However, we think this solution is sensible for cases of highly important conditions (i.e., ones that are utilized frequently to get rewards) and may be

especially sensible for navigation in terms of representing maze locations. Therefore, our results have ruled out the idea that this task is directly solved by neurons with pure XOR (or choice) selectivity and instead utilizes a basis set of mixed selectivity cells. We now present these possibilities in the introduction (LL 84-91) and discuss their implications and interpretation in the discussion section (LL 591-609).

1) Intellectually the aspect of the work I found most interesting was the combination of theory/simulation (supplementary figures 7) and analysis of experimental data (figure 5, supplementary figure 5 and 6) demonstrating the efficiency of mixed representations in RSC for encoding reward location. This observation is particularly satisfying when combined with their inactivation experiments. This aspect of the manuscript could be strengthened by the addition of a clear mechanistic model for how mixed selectivity neurons in posterior cortex drive decisions. Such a model would allow quantitative predictions that could be evaluated with future experiments.

We thank the reviewer for the suggestion to highlight the computational aspect of mixed selectivity. To highlight the efficient coding of mixed selectivity cells, we moved the simulation results to a new main figure (Fig. 7). We feel this change should help highlight the results that the reviewer found to be of interest.

Overall, we feel our work is better suited to talking about the computational implications of the coding schemes we report rather than developing a mechanistic model. Therefore, we have chosen to emphasize the modeling work that explores the efficiency of codes with mixed versus pure selectivity. We considered the possibility of adding a mechanistic model, such as for how the mixed representations arise. We agree with the reviewer that this is an interesting topic and a rich area to pursue for modeling. When we considered the possibilities for a mechanistic model, however, we realized that our data might not be ideally suited for this type of model. The reason is that the data are mostly observations of activity patterns, and these activity patterns could be consistent with multiple possible underlying mechanisms. For example, if we consider a mechanistic model for how the mixing arises, it is possible that the mixing arises within posterior cortex, but we cannot exclude that it is inherited in the inputs from other regions. We feel follow up work involving a dedicated set of experiments and a full study would be needed to provide more insight into possible mechanisms that could be incorporated into a mechanistic model. We therefore felt a mechanistic model might be too speculative at this point. Instead, we included in the discussion some ideas about the mechanisms underlying the mixing and where the mixing might arise from (LL 626-634).

2) Several statements and claims throughout the text are either unclear or not supported by the data. They interfere with the reading of the paper and should be addressed. Some of these sentences are not critical to the main point of the paper, so removing or altering them would have no significant effect on the results. Examples include, but are not limited to the following:

Line 32 - “Together, these characteristics imply that specific neural mechanisms exist for rapid flexibility over times as short as seconds, which are too brief for substantial neural circuit plasticity.”

This statement is vague, do the authors mean to suggest that flexible decision-making does not involve moment-to-moment changes in synaptic plasticity? What is the evidence for this? This is an example of a claim which if removed would not impact the manuscript.

We agree that this sentence was poorly worded and unclear in its meaning. We did not intend to imply that synaptic plasticity is not involved in flexible decision-making. We actually cite several papers that propose how short-term synaptic plasticity could play a role (Masse et al., 2020; Masse et al., 2019; Mongillo et al., 2008). We have therefore removed that sentence from the paper.

Line 38 - “The flexibility of decision-making has been investigated in experimental paradigms that do not involve spatial navigation.”

This statement is a bit unclear. Do the authors mean that some studies of flexible decision-making have been performed in a non-spatial context or that ALL studies of flexible decision-making have been performed in a non-spatial context. Given the authors’ definition of flexible decision-making it seems that the latter is not accurate. There are many papers using, for example, delayed non-match to sample T-mazes and alternation T-maze - based tasks to study the influence of working memory on decision-making. The authors should not ignore this literature.

We thank the reviewer for pointing out these areas of related work. We do not intend to state or imply that our work is the first and only one to study these questions in the area of spatial navigation. We agree that works related to non-match to sample T-mazes and alternation T-mazes are relevant for how memory impacts decision-making. However, we note that our work is focused on the interaction between sensory and memory signals, which is not studied explicitly in some of this earlier work. We have now included citations in the introduction to earlier work in the spatial navigation area, as suggested, and have updated our wording on this topic (LL 66-67).

C. Line 81 – “We discovered neurons that mix short-term memory and visual information, and these neurons were present in most parts of posterior cortex, even in V1. Surprisingly, RSC had the highest density of these neurons, whereas these cells were sparsest in PPC, with a near absence in anterior PPC. These neurons formed an efficient population code, which appeared to be critical for accurate decisions because their activity was more informative when the mouse made correct decisions compared to errors.”

Do the authors mean to suggest that because these neurons represented task variables with higher fidelity on correct trials, they support decision-making? I’m not sure I follow this logic.

Does this argument require the assumption that correlation implies causation? Throughout the paper claims about the role of cortical neurons in flexible decision making are made based on the response properties of RSC neurons during behavior. The authors should be more careful and precise about the language used when inferring function based on recorded dynamics.

We did not intend to state that the analysis of correct and error trials indicates a causal relationship. We find the difference in the activity of the mixed selectivity cells on correct and error trials to be interesting and important to report. Specifically, we find that mixed selectivity cells have lower XOR information on error trials than on correct trials. Following logic similar to that of the long-studied analyses of choice probability in cortex (Britten et al., 1996), we feel these analyses of correct and error trials indicate that these cells are a candidate for having a causal role in decision-making. However, we agree entirely with the reviewer that these analyses are not suitable for making claims about causal relationships and experiments to manipulate these mixed selectivity cells would be necessary to establish such relationships. While these experiments are of high interest to us, they are at the moment not experimentally tractable. Because the mixed selectivity cells are intermingled with other cells and because we do not have a molecular label specific to the mixed selectivity cells, we would need to use patterned, two-photon optogenetics to silence only these cells. While this approach is emerging, we are not aware of any papers that have silenced a substantial fraction of cells that are functionally defined and broadly distributed across areas. We have therefore reduced the strength of our wording throughout to make it clear that the correct/error analyses are only suggestive of a role in decision-making. We have also added explicit statements in the discussion that say that the correct/error trials do not reveal a causal relationship and that future experiments will be needed to test for such a causal relationship (LL 587-589, 680-683).

D. Line 244 – “The GLM poorly predicted neural activity in M1 and M2 (Fig. 2I) although the mice showed similar performance during imaging sessions for each area (Supplementary Fig. 4a-b), suggesting these areas may be less informative for solving the task compared to posterior cortical areas.”

I have a logical problem with the idea that just because the model does not predict the activity in cells, that those cells are somehow less likely to be involved in the behavior. The optogenetic experiments provide information about this causal relationship. However, if the authors' model does not predict the activity of a cell, perhaps this tells you more about the parameterization of the model rather than the role of these cells in driving behavior. For example, is it possible that different non-linearities exist between task variables and activity of cells in M1 and M2 and that the diminished performance reflects a poor choice of non-linearity?

We agree that our GLM analyses are dependent on the design and parameterization we choose for the model. We note that our GLM does include the capability of identifying complex interactions and non-linear relationships between task variables, although it cannot approximate some aspects of activity (e.g., the saturation of activity not captured by our choice of exponential link function). We have now examined M1/M2 activity using decoders, which have

complementary advantages and limitations to the GLM approach. We performed decoding from the activity in neural populations using logistic regression as a linear decoder and a support vector machine with a radial basis function as a nonlinear decoder. With these decoders, we again found that the M1/M2 activity was less informative about the sample cue, test cue, and XOR in the early part of the test segment. We did notice choice information in these areas later in the trial when the mouse was executing left and right turns. Despite these new analyses, we do not aim to emphasize the point that M1/M2 are less important and have thus made sure our wording is not strong regarding these areas (LL 659-661). Furthermore, we have added new sentences to the discussion that explains why M1/M2 might not be required or involved in our task, whereas others have found it plays a role (LL 661-674). In particular, in our task, we fix the mouse's virtual (and real) heading orientation in the maze as it traverses the virtual maze. It is possible that M1/M2 are more involved in cases in which the heading orientation is controlled and adjusted by the mouse in a constant and iterative manner. We therefore expect M1/M2 could very well play an important role in different manifestations of our task.

E. Line 293 - "The distribution of information in RSC cells in the test segment was approximately uniform between sample cue selective, mixed selective, and test cue selective cells, which was markedly different from the distribution in V1...Together, the results show that RSC has approximately equal mixing of information, the largest fraction of cells with mixed information, and the highest XOR information, indicating that RSC could be a central area for mixing memory information and visual information."

Do the neural dynamics suggest it is RSC plays a central role or the inactivation experiments? What models do these data rule out?

We have used our inactivation experiments to identify brain areas that are important for the task. Because there are many aspects of the task, including visual perception, mixing of memory and sensory signals, and executing motor outputs, it is challenging to use the inactivation experiments to pinpoint the precise role of a given brain area in the task. Rather, we can say confidently that the brain area is necessary for accurate performance of the task and then follow up with experiments to see how that area is involved in the task. We have clarified this point in the results section when we present the inactivation experiments (LL 194-201). We therefore find the calcium imaging experiments more informative about RSC's role in mixing. Of all the areas we imaged, RSC had the most equal mixing, the largest XOR information, and the largest fraction of mixed selectivity cells. It is true that mixed selectivity cells were present in all parts of posterior cortex, but they were densest in RSC.

As mentioned above, one of the topics we aimed to address is what cortical areas are important for flexible decision-making during spatial navigation. One possibility is that the relevant cells and computations are restricted to a single cortical area, where frontal areas, PPC, and RSC are candidates based on prior literature. An alternative possibility is that the relevant cells and computations are widely distributed across cortex. Before performing these experiments, we were not sure which to expect, and we found evidence in the literature that suggested both

alternatives could be possible. Our results reveal that the mixed selectivity cells are distributed across posterior cortex but are less prominent in frontal areas. Furthermore, they show that RSC has the highest density of these cells and that PPC unexpectedly has a lower density. Also, surprisingly, V1 appears to be important for this representation of mixed sensory and memory signals. In the introduction and discussion, we have now more directly stated these alternatives and where our results fall regarding these alternatives (LL 76-82, 568-574).

F. Line 347 – “For trials with identical sensory cues, if cells have higher information about the reward direction (XOR) on correct trials than on error trials, then such an observation would support the notion that the information in those cells was used to guide accurate choices.”

This is another example of the issue highlighted in C: Does this argument require the assumption that correlation implies causation?

We do not intend to imply that this analysis reveals a causal relationship. Rather, we feel this logic and results indicates that these cells could be candidates for having a causal relationship. We have now tried to be more careful in our wording and have now lessened the strength of this wording. We also note in the discussion section that these correlations do not imply causation and that future experiments are needed to test for causal relationships (LL 587-589). Our thinking follows the long line of studies that have investigated choice probabilities in the cortex as a way to identify cells that are potentially causally involved in a task. For example, in the seminal study of perceptual decision-making from Britten et al. (1996), they use a similar analysis of the correlation between neural signals and the animal’s choices. Based on these analyses, they state: “These observations are consistent with the idea that the neuronal signals in MT are used by the monkey to determine the direction of stimulus motion”. We have tried to mirror their language and terminology and have tried to be careful about interpretation and presentation of correlative analyses.

G. Line 372 – “Together, these observations support the idea that aberrant activity of mixed selectivity cells across a distributed set of areas, and most strongly in RSC, contributes to incorrect choices.”

These observations are also consistent with a variety of other models. Does this data rule out any models?

We did not intend for this statement to conclude that some models are correct and others are ruled out. We think it is a valuable observation that the mixed selectivity cells have aberrant activity on error trials. Following the logic that has been used for decades in the decision-making field, going back to the work of Britten, Newsome and colleagues, this change in the mixed selectivity signal on error trials suggests that these cells could be candidates for playing an important role in the decision. That is, the mouse might be “listening” to these cells or using these cells to make its decision, and when these cells’ signals are disrupted, the mouse may be more

likely to make errors. We understand that this is a correlation and does not reveal causality. We agree with the reviewer's point about this above, and we have revised our wording in the text to make it clear that we are not implying causality (LL 587-589). However, we think this observation is interesting and worth reporting. It is true that signals that do not vary with correct and error trials could also be critical for the task, such as signals in the retina for the visual virtual maze, but the fluctuations between correct and error trials could indicate that these cells are a critical node in the decision circuit. Again, this is a correlation only, and future work will be needed to address this point. Nevertheless, we felt this observation was interesting enough to report and emphasize because it provides a key characterization of the activity patterns we are reporting.

3. Figures and figure legends could be improved for clarity. Revising figure titles to state the main finding or discovery made rather than the experiment performed would help. For example, in figure 1: "Optogenetics screen for cortical areas involved in a flexible navigation task", the authors might consider a title that indicates the main finding of the experiment such as "Optogenetic inactivation screen reveals the key role of RSC in flexible navigation". Many figure titles could be altered in this way and such changes will help the reader absorb the main idea information presented in the figure more readily.

We thank the reviewer for this suggestion. We prefer to keep the figure titles as factual about what was done and to avoid adding interpretation into the figure titles. We understand that this is a matter of style and are happy to change the titles as suggested if other reviewers agree this would be helpful. Instead, we have tried to clarify the text around what is learned by placing our results and experiments into a better motivating and interpretative context in the introduction and discussion.

4a: In addition to the titles both the figures and the figure captions themselves should be revised for clarity. For example, in Figure 1 – the organization and order of panels g-m is confusing. In particular, Panel l seems to be the bridge between g and h but comes well after h and is unlabeled. Perhaps it could be labeled, relocated or placed in supplemental.

We have rearranged the figure panels as suggested.

b: Another example is the caption for Figure 5a. The caption reads "Mixed selectivity cells encode the identity of the sample cue, test cue, and reward direction (XOR) (purple rectangle) ...". But what is shown is a schematic of the approach for separating recorded neurons into two subpopulations (mixed and pure" based on their response properties). Stating this clearly would help the reader understand the purpose of this panel rather than forcing them to deduce it.

We have updated the caption to explain separately the left panel, which shows the partition for mixed vs. pure selectivity cells, and the right panel, which shows our decoding approach.

Reviewer #3 (Remarks to the Author):

Kira et al. investigated the way mice integrate information about visual stimuli and remembered cues during a delayed match-to-sample task. They study this task in an interesting spatial navigation context in which the animals navigate in virtual reality, with the first "sample" cue and second "test" cue separated in both time and (virtual) space. Once the test cue is presented, the animals are required to turn towards the side that matches the sample cue.

To investigate the cortical regions involved in this task, they iteratively inhibited different areas using optogenetics, and determined which such perturbations most impaired task performance: retrosplenial cortex (RSC) and V1 stood out as quite important, while frontal areas were surprisingly less important. While I love the idea of causally testing the importance of each brain area in this way, I have some practical concerns about these experiments that I discuss further below.

Thereafter, the authors probed the structure of the neural code in each relevant brain area, using neural activities recorded by Ca²⁺ imaging. They found that neurons in RSC in particular often had mixed selectivity -- responding nonlinearly to both the sample cue and the test cue, in a fashion that provided information needed to perform the delayed match-to-sample task. This is referred to as XOR information since the reward matrix for the task matches the logical XOR operation.

Finally, the authors speculated that this mixed selectivity was critical in generating an efficient neural code to support the mouse's ability to perform the task.

This is an interesting study with some exciting findings. I have some fairly substantial concerns that I outline below. I am hopeful that the authors could address my concerns in a suitable revision.

We thank the reviewer for the feedback and constructive suggestions.

My specific comments are as follows:

1) While lots of researchers have studied neural computations during delayed match-to-sample tasks, a key point of novelty for this study is that this is done in the context of navigation. While I agree with the authors that navigation is an important naturalistic behavior in which to probe context-dependent neural computations (like delayed-match-to-sample), I am not convinced that this task really has much to do with navigation, because the experimenters decoupled the animal's motion through the virtual space from their locomotion on the ball, up until the time of their decision. This means that the task that the mouse needs to perform can be reduced to: "after the delay, turn the ball in the direction that matches the remembered sample cue". Put in this way, it is unclear to me how the navigation aspect really affects the task. Overall, I would find the novelty of this aspect of the study to be more persuasive if the authors could explain better

why the navigation aspect is critical to the mouse's ability to do the task. Similarly, I would be very curious to know the roles of hippocampus and/or superior colliculus in this task, given their known roles in navigation/memory, and in orienting movements.

The reviewer raises an interesting and important point. During the sample segment and delay segment, the mouse is not allowed to make lateral movements in the virtual maze. Thus, the rotations of the ball around the roll axis are not used to generate movements in the virtual maze during these segments. However, the mouse's movements are coupled to forward translation in the maze, with the mouse having full control of how rapidly and slowly it traverses the maze. We chose this implementation to make sure that there are no lateral or rotational movements that change the visual scene presented on the VR display, ensuring that the visual scene is identical on every trial with the same cues. We agree with the reviewer that this comes with the disadvantage of making the VR less interactive and not closed loop in all dimensions in the sample and delay segments. We decided that this was reasonable given the advantage of having a perfectly controlled visual scene.

We still feel this task involves navigation because the mouse still has to move through the maze to reach a reward site that is a location in a virtual environment. The task thus has a strong spatial component, is operated in a virtual maze, and requires the mouse to actively move in a closed-loop manner to reach the reward site. We think these factors are sufficient to recreate some aspects of navigation. In other experiments in the literature, similar setups have been used to study the key cells involved in navigation, including place cells and grid cells. For example, place cells and grid cells have been studied during traversal of a virtual linear track in which movement is only allowed forward and backwards (no turns or lateral movement)(Campbell et al., 2018; Domnisoru et al., 2013; Low et al., 2021; Pettit et al., 2022a; Pettit et al., 2022b; Plitt and Giocomo, 2021), which is similar to the movements in the sample and delay segments in our maze. These studies have found robust place cell and grid cell activity, including with many properties that are similar to those in freely moving mice. It is true that not all properties are identical, and some aspects of navigation and navigation-related neural activity are compromised with such a setup. This reasoning and previous results indicate to us that navigation is involved in this type of task despite the caveats mentioned by the reviewer.

We note that our task is very different from one in which the visual stimuli are presented in a manner paced by the experimenter. In many visual DMS tasks, a sample cue is displayed for a duration determined by the experimenter and then a test cue is presented after an experimenter-determined delay duration. In this case, the timing and progress of a trial are determined solely by the experimenter. The animal reacts much as the reviewer describes, that is by making action upon seeing the test cue. In our case, the trial progresses at the speed determined by the mouse, and the trial's progress will halt at any time the mouse stops running. While it may be possible to think of other task implementations that require active movements of the subject, in our case, these active movements generate movement through a virtual maze with sensible spatial relationships between locations. We feel these are major differences that likely affect the brain areas involved in solving a flexible decision task.

We have preliminary evidence that the task implementation in a virtual navigation context like we used does indeed make a difference compared to a non-navigation context. In a separate cohort of mice, we tried to train them to do a visual delayed match-to-sample with visual cues presented by the experimenter. The visual cues were similar to those used in this paper (black and white cues). In these mice, instead of running on a ball, they were stationary in a tube. Also, instead of actively moving through a maze in a self-paced manner and moving through spatial locations with consistent spatial relationships to one another, the cues were presented on a screen by the experimenter with experimenter-determined timing. This alternative of the task clearly did not involve navigation because there was no maze and because the trial progression was not controlled by the mouse. We found that mice learned the virtual navigation version of the delayed match-to-sample task much more quickly and easily than the non-navigation version. In fact, it was quite challenging to get mice to learn the non-navigation version, even though we have lots of experience training mice on complex tasks. This difference in ease of training was striking to us and suggested that the virtual-navigation version does indeed tap into naturalistic behaviors, such as the spatial relationships involved in navigation. We completely acknowledge that this is speculation and that there are likely equally good alternative interpretations, but we felt this was worth mentioning as it was informative to us.

We have now updated the description of the task to note why we consider this task to include navigation. Due to space restrictions, we have not expanded on the ideas as deeply as we do here. However, we hope the new sentences in the Results section provide the reader with a stronger appreciation of why we feel this task involves navigation (LL 127-131).

We agree that it would be very interesting to understand the role of hippocampus and superior colliculus (and other areas) in this type of task. However, given the intensive nature of these experiments, we feel that this is beyond the scope of the current paper, especially because different experimental methods would be necessary to reach deeper brain structures below cortex. We feel studying these areas would benefit from a dedicated study.

2) I really like the idea behind using optogenetic inhibition experiments to causally test the necessity of each brain area! The practical implementation, however, comes with some pretty major limitations, only some of which are discussed by the authors. While they do convincingly show that V1 and RSC are needed for the task (at least during the sample and test period), I had some major questions about possible false negatives, among other things:

In their optogenetic experiments, do the authors silence neurons all the way down to layer 5 (or 6), or is the silencing restricted to more superficial cortical laminae? This information is somewhat available in Fig. S2, but without knowing the cortical thickness of each brain area, and the reference against which they are measuring depth (which would define the location of "zero depth"), it's hard for me to map their cortical depth values to cortical laminae. Those data (the depth to which neurons are inhibited by the laser) are also absent for the frontal areas for which I think they might be most important in ruling out false negatives from the silencing experiment. I explain this last point below.

Given the different projection targets of pyramidal cells in the different laminae, different results may be obtained if excitatory neurons in only some laminae were silenced vs if they were all silenced, down to and including L6. For this reason, it is hard to say from the data presented whether frontal areas are really not necessary for the task (as implied in Fig. 1ijk, for example), or just whether the superficial areas of those areas are not necessary for the task, while the deeper neurons (potentially less affected by the optogenetic manipulation in these experiments) might still be necessary.

We thank the reviewer for raising this important point. We apologize that some methods details were missing. The “zero depth” was defined as the entry point of the electrode at the cortical surface. The electrode was inserted ~1 mm laterally from the laser spot with a 60-degree angle from the horizontal plane. The depth, however, was measured along the axis perpendicular to the horizontal plane. The cortical thickness is approximately 0.9 mm in PPC, 1.3 mm in RSC, and 1.4 mm in M2. Note that RSC is folded into the longitudinal fissure, so the depth does not correspond to cortical layers in its ventral granular region.

Given that the photoinhibition effects faded around 1 mm from the cortical surface, it is possible that some activity remained in the deep layers of cortex. This may be especially true in the thicker parts of cortex (M1/M2). Similarly, the ventral granular region of RSC and ventral region of ACC may have also had some remaining activity that was not photoinhibited. We note that even if some of the activity remained, many of the inputs to the non-inhibited cells had been removed. Deep cortical layers receive significant input from upper cortical layers, so it is possible that any remaining activity might be aberrant in nature. However, we were unable to verify this speculation. We now mention in the Results section that some activity may have remained in the deeper layers, which complicates interpretation of the results and could lead to false negatives (LL 173-175). Based on these considerations, we have lessened our emphasis on the idea that frontal areas of cortex are not involved in the task and instead have focused more on the areas that are involved in the task. We have also mentioned in the Discussion section that our experiments do not rule out the involvement of frontal areas in the task (LL 660-661).

2b) In what inhibitory neuron types is the Chr2 expressed in the VGAT-Chr2 mouse? Perhaps it is all of them, in which case the authors should emphasize this point in their methods. Given the different roles played by VIP, PV, SST, etc., interneurons, if only some of interneuron types express the Chr2, I would worry that the effects of the optogenetic perturbation might be more complicated than "just" silencing excitatory cells. This would make it harder to interpret the data in Fig. 1 that aim to show the necessity of each brain area.

Chr2 is expressed in all GABAergic neuron types in VGAT-Chr2 mice (Li et al., 2019; Zhao et al., 2011). We now mention this in the Methods section (LL 913-914).

Given what's in Fig. S2, it looks like spiking activity in RSC is completely shut off down to ~700um of depth (although I do not know what cortical layer that corresponds to), whereas in M2, for example, it is strongly attenuated but not quite shut off. This poses the question of what these distorted activity patterns look like, and that will depend on the types of interneurons driven by the laser.

This means that, in interpreting the M2 data from Fig. 1, rather than concluding that "M2 has only a modest role in this task", one could potentially conclude that "M2's contribution to this task is surprisingly robust against the manipulation performed here".

We thank the reviewer for proposing an alternative interpretation. While this is a possible interpretation, if the inhibition of most activity in one area results in only little behavioral deficits, we think a more parsimonious explanation would be that the inhibited area is less important for the behavior, rather than that the small residual activity is so important that it can compensate to produce the entire function and normal behavior. That said, it remains possible that the activity in L5/6 of M2 is necessary for the task performance as mentioned above. We have therefore made our interpretations more cautious and lessened our emphasis on the idea that frontal areas are not involved.

3) I would remove Fig. 1C since it is uninformative and confusing. It took me a while to realize you had forced the VR headings and positions to take on those specific values, and during that period I was puzzled by why there was so little behavioral variability. This is a minor point of course, but I think that removing this panel could help with the clarity.

Thank you for this suggestion. In past versions of the manuscript, some readers missed the point that we fixed the lateral position and view angle during the sample, delay, and test segments. We thought adding this panel would clarify this point, and we apologize that it caused confusion. We prefer to keep this figure panel because we feel this is an important point that was missed by other readers. We have aimed to clarify this by making the wording in the text clearer (LL 131-137) and by adding new components to the figure panel that clearly mark where the lateral position is constrained by task design (Fig. 1c-d). We hope these changes allow readers to understand this important point more easily.

4) I liked that you checked whether the wheel motion during the delay period could carry a mnemonic encoding of the sample cue. If you have video recordings of the mouse doing the task, I'd suggest also looking at those to see if anything obvious pops out at you (e.g., tail orientation). There are in principle a huge number of possible ways the mouse could develop a mnemonic for the sample cue, and I don't expect you to exhaustively probe those possibilities. But -- if you haven't done this already -- watching some video from each animal (since they may develop their own unique mnemonics) doing the task could help to spot any obvious mnemonic behaviors beyond the wheel motion.

We recorded videos of the behavior in a subset of sessions, but unfortunately we do not have enough data to do a systematic analysis of things like tail orientation. In these videos, nothing obvious popped out, but regardless it is likely that some other parts of behavior may also correlate with the sample cue identity. We have tried our best to measure and control for these types of potential mnemonics with our current data, and further investigations would require collecting a new dataset.

Interestingly, in our new experiments in “naïve” mice that are trained to run the maze but have not been trained on the delayed match-to-sample task, we noticed that some mice ran in different directions for different cues. This is surprising because these “naïve” mice were not performing a task and had not associated the cues with any specific meaning in terms of reward. There was no benefit for these distinct running patterns in the “naïve” mice. Based on this observation, it seems likely that the distinct running patterns for different cues can be, at least partially, idiosyncratic without a clear behavioral purpose.

R3-5) Line 202-203, you mention deconvolving the calcium traces. This is notoriously tricky, with isolated spikes often being missed (e.g., see recent publications including Huang et al. eLife 2021). I think the authors should discuss this limitation, and help readers like me to understand whether or not it severely confounds their analyses. (Hopefully the answer is "no it does not", for some good reasons).

We agree that deconvolving calcium traces is a difficult and imperfect process. We chose to use deconvolution because the calcium transients have long decays that extend past the events that triggered the transients. We wanted to shorten the signals to around the events so that we did not get bleed-through of activity from, for example, the sample segment into the delay segment or test segment. Now we mention this reasoning for deconvolution in LL 223-224.

We have added new analyses to build confidence that the deconvolution is not confounding our results. We now performed population decoding on the calcium transients without deconvolution. The results are qualitatively similar to those that we get with deconvolution (Supplementary Fig. 7). We note that some of our concerns mentioned above seemed to play out. Without deconvolution, the sample cue was decoded more accurately in the test segment, consistent with the long transients bleeding across segments. In contrast, the test cue and XOR were decoded less accurately in the test segment without deconvolution, suggesting that the deconvolution actually denoised the signal. We feel these new analyses help demonstrate that deconvolution does not create spurious results and help to justify the use of deconvolution.

6) Line 234: I am curious about why the authors used the FDE measure instead of something more common like fraction explained variance (FEV). Unless there's a good reason to choose this non-standard measure, I suggest using a more common measure (like FEV). That would make it a bit easier for readers to follow since they wouldn't have to figure out what this new measure is, and would help readers use their prior knowledge to more easily interpret the values you present.

We agree that fraction of deviance explained (FDE) is less commonly reported than fraction of variance explained (FVE). These two quantities are closely related, with the key difference between the choice of the error distribution around the predicted mean value for fitting. FVE assumes the same Gaussian error distribution around the mean at every time point, whereas FDE takes into account distinct error distributions around the mean (based on a Poisson distribution in our case). We reasoned that the choice of a Poisson error distribution is more appropriate than Gaussian error in our case. That is because a Gaussian error distribution assumes that the neural activity can take a negative value, whereas a Poisson error distribution is restricted to positive values. If the mean activity is always high (e.g., > 20-30 spikes/s), the error distribution could be approximated by a Gaussian distribution. However, the neural activity in L2/3 of the mouse cortex is typically lower (< 10 spikes/s), so it does not conform to a Gaussian error distribution. We therefore feel it is more appropriate to present our results with FDE than FVE. We now mention this in the Methods section (LL 1157-1159, LL 1251-1255).

7) Fig. 2 EFG,HIJ: could you please make it more clear on the figure (maybe next to the cell ID numbers) what brain areas each neuron comes from? I initially thought these were all from the same area, which of course they are not.

We thank the reviewer for this suggestion. For each example cell in each panel, we indicated the area they were recorded from (Fig. 2e-j, Supplementary Figs. 3-4).

8) Given the discussion of correlations later in the paper, it is worth asking why the GLM has no coupling between cells. Including that coupling in the GLM would facilitate estimation of log-likelihood ratios for *populations* of cells. Instead, the present analysis estimates the logLR for each cell and then presents it as though it describes the population. I am sure it's unintentional, but I found this part a little bit misleading. E.g., in lines 287-288, the authors state that "RSC had significantly larger XOR information than V1. . .", but based on their analysis, I think a more accurate statement would be "On average, individual neurons in RSC had significantly larger XOR information than did individual neurons in V1". Making claims about the information in the population (rather than the single-neuron properties) requires consideration of the joint distributions, which could be achieved by using GLMs with coupling between neurons.

We apologize for the imprecise wording and thank the reviewer for the suggestion on how to improve it. We have revised the sentence to read: "Importantly, RSC had on average significantly larger XOR information per cell than V1." (LL 310-311)

In our initial submission, we did not include coupling terms for three reasons. First, in many cases, our analyses focused on the average information in individual cells, as noted by the reviewer, so we did not need the coupling terms. Second, the models would have grown to hard-to-manage sizes due to the large number of simultaneously recorded neurons. Third, even with the assumption of conditional independence across neurons, our GLM decoder can capture some

effects of noise correlations (Panzeri et al., 2022). For example, an uncoupled GLM may be able to capture the information limiting effects due to the similarity between signal- and noise-correlations (Supplementary Fig. 9a-c). This effect seemed to manifest in our data as we observed an improvement in the decoding accuracy after reducing noise correlations (by shuffling trial identities within each trial type for each cell) (Supplementary Fig. 9d).

However, we agree with the reviewer that additional population-level analyses that take into account correlations are helpful. That is because our GLM decoder may fail to capture some aspects of the information in population activity. For example, it will fail to capture the information carried by the modulation of noise correlations for different cues or choices, termed stimulus-dependent correlations (Panzeri et al., 2022). To understand if our conclusions were robust to changes of assumptions in decoders, we performed new analyses with different decoding methods, including nonlinear ones that can directly take into account the effects of correlations (Supplementary Fig. 7). Specifically, we performed logistic regression (linear decoding) and support vector machine decoding with a radial basis function kernel (non-linear decoding). These methods have the limitation that they do not discount the effects of movement and other variables on neural activity, in contrast to the GLM decoder. That said, these decoding methods produced qualitatively similar results to those with the GLM decoder. Therefore, these new analyses provide further support for our population-level analyses.

9) Fig. 3ab: could you add to your caption or text a bit more information to explain what the breaks are along the time axis. I.e., "we show the first and last second of activity during the sample period, but not the middle portions", and similar statements for what is shown during the delay and test periods. It took me a fair while to figure out why there were breaks in the logLR vs time plots.

We apologize for not explaining this clearly. We have revised the caption for Figure 3a accordingly. (LL 1826-1827)

10) Can the authors quantify the statistical significance of each neuron's logLR values? This is important for understanding the strength of the claims about single-neuron information representations from those logLR values. This especially comes up on line 328-329, where a cutoff of $\log LR > 0.01$ is used to define cells with "appreciable" XOR information. Without knowing the uncertainties in the logLR values, it is hard to know whether this is a reasonable cutoff. A better criterion might be to test the significance of each neuron's XOR logLR against zero, and use $p < 0.05$ (or $p < 0.01$) for the cutoff.

11) Relatedly, in Fig. 3HI and 4HI, I the measurement of polar angles seems problematic unless at least one of the two logLR values is statistically significantly non-zero.

We apologize that our methodology was not clear enough. We agree with the reviewer that this is an important point. We did not quantify the statistical significance of each neuron's

information values because of limits on the computational resources we have available. Such a calculation would involve running many shuffles of each cell's model. Fitting each GLM for a large number of cells and shuffles leads to intractable computational times, at least with the resources we have available at the moment.

Instead, the chance level was estimated from the standard deviation of the test cue information in the sample segment. Because the test cue has not been presented yet in the sample segment, there should be zero test cue information. We could therefore use any test cue information values in the sample segment to estimate our noise level because any test cue information in the sample segment must be noise. We identified these chance values summarized across all cells for an area. We found chance LogLR values of 0.012 for V1, 0.004 for RSC, and 0.003 for MM. Based on these chance values, we set 0.01 as a cutoff value on the magnitude of logLR. This sets the criteria for “appreciable” information and excludes cells around the origin in Figures 3h-i and 4h-i. As the reviewer mentioned, it is critical to eliminate these cells because the polar angle cannot be measured accurately for these cells. We verified that the results did not qualitatively change even when we varied this cutoff value in the range of logLR = 0.005–0.02 or used the area-specific noise value as the cutoff for each area. We realize that a per-cell cutoff for significance may be better, but this estimate of the cutoff level serves a similar purpose and is amenable to the computational resources we have available. We apologize that this methodology was not clear enough. We now briefly mention it in the Results section (LL 363-364) and reference the Methods (LL 1394-1399) for more information.

12) I really liked the analysis in Fig. 4K! I kept wondering earlier in the paper whether all of the effects reduced to the mouse just forgetting the sample cue on the incorrect trials, and I thought this analysis was a great way to show that there's more going on.

Thank you for this positive comment.

13) Lines 409-410: why show the mutual information between decoder and truth, and not just the accuracy of the decoder? The two are monotonically related of course, but the accuracy is probably much more easily understood by most of your readers.

We thank the reviewer for the suggestion to present the decoding accuracy. While decoding accuracy is more intuitive in its units, we chose to report the mutual information for three reasons. First, the amount of information in the population scales linearly with the mutual information values but not with the decoding accuracy. For example, a given increase in population information will lead to a larger increase in decoding accuracy around 60% correct compared to around 95% correct. Said another way, to increase the decoding accuracy by 5%, a larger increase in information is necessary to go from 90 to 95% compared to going from 50 to 55%. Second, the mutual information better corresponds to the logLR used for the single cell analyses than the decoding accuracy because the chance value of information is zero, as in logLR, but the chance value of decoding performance is non-zero. Third, for a suggested analysis in the

reviewer's comment below, it would be more sensible to sum information in mutual information for comparisons rather than in decoding accuracy. If two populations have independent information from each other, the sum of the information value in each population equals to the information value jointly encoded by the two populations. If two populations have redundant (or synergistic) information, however, the sum is smaller (or greater) than the jointly encoded information value (Azeredo da Silveira and Rieke, 2021; Nigam et al., 2019; Pola et al., 2003; Schneidman et al., 2003). In contrast, the sum of two decoding performances cannot be interpreted in this way. We therefore prefer to stick with mutual information for reporting the population information values.

14) In Fig. 5B, why not also show decoding accuracy for the combination of both types of cells, in addition to just the mixed selectivity and the pure selectivity ones? The brain has all of these cells, and I was curious about how the information from the two types of cells combines. I.e., if the full population including both types of cells has about the same information as just the sub-population of mixed selectivity cells, that tells me that the mixed ones really dominate the representation. But maybe the two cell types together have more information than the sum of the information presented by each cell type in isolation. This can happen depending on the correlations between cells of the two types. Adding this comparison would help support the assertion on line 464, that the mixed selectivity cells really are the key to the animal being able to perform the task.

We thank the reviewer for this suggestion. In new Figure 6b, we now show the XOR mutual information in a population of pure selectivity cells, a population of mixed selectivity cells, and a population that combines mixed and pure selectivity cells. This new analysis shows that a population of mixed selectivity cells has nearly the full information present in the population with both mixed and pure selectivity cells. Thus, the mixed selectivity cells appear to dominate. This result also suggests that the information in the pure selectivity cells is largely redundant with the information in the mixed selectivity cells.

References

- Azaredo da Silveira, R., and Rieke, F. (2021). The Geometry of Information Coding in Correlated Neural Populations. *Annu Rev Neurosci* 44, 403-424.
- Britten, K.H., Newsome, W.T., Shadlen, M.N., Celebrini, S., and Movshon, J.A. (1996). A relationship between behavioral choice and the visual responses of neurons in macaque MT. *Vis Neurosci* 13, 87-100.
- Campbell, M.G., Ocko, S.A., Mallory, C.S., Low, I.I.C., Ganguli, S., and Giocomo, L.M. (2018). Principles governing the integration of landmark and self-motion cues in entorhinal cortical codes for navigation. *Nat Neurosci* 21, 1096-1106.
- Domnisoru, C., Kinkhabwala, A.A., and Tank, D.W. (2013). Membrane potential dynamics of grid cells. *Nature* 495, 199-204.
- Li, N., Chen, S., Guo, Z.V., Chen, H., Huo, Y., Inagaki, H.K., Chen, G., Davis, C., Hansel, D., Guo, C., and Svoboda, K. (2019). Spatiotemporal constraints on optogenetic inactivation in cortical circuits. *eLife* 8.
- Low, I.I.C., Williams, A.H., Campbell, M.G., Linderman, S.W., and Giocomo, L.M. (2021). Dynamic and reversible remapping of network representations in an unchanging environment. *Neuron* 109, 2967-2980 e2911.
- Masse, N.Y., Rosen, M.C., and Freedman, D.J. (2020). Reevaluating the Role of Persistent Neural Activity in Short-Term Memory. *Trends in cognitive sciences* 24, 242-258.
- Masse, N.Y., Yang, G.R., Song, H.F., Wang, X.J., and Freedman, D.J. (2019). Circuit mechanisms for the maintenance and manipulation of information in working memory. *Nat Neurosci* 22, 1159-1167.
- Mongillo, G., Barak, O., and Tsodyks, M. (2008). Synaptic theory of working memory. *Science* 319, 1543-1546.
- Nigam, S., Pojoga, S., and Dragoi, V. (2019). Synergistic Coding of Visual Information in Columnar Networks. *Neuron* 104, 402-411 e404.
- Panzeri, S., Moroni, M., Safaai, H., and Harvey, C.D. (2022). The structures and functions of correlations in neural population codes. *Nat Rev Neurosci* 23, 551-567.
- Pettit, N.L., Yap, E.L., Greenberg, M.E., and Harvey, C.D. (2022a). Fos ensembles encode and shape stable spatial maps in the hippocampus. *Nature* 609, 327-334.
- Pettit, N.L., Yuan, X.C., and Harvey, C.D. (2022b). Hippocampal place codes are gated by behavioral engagement. *Nat Neurosci* 25, 561-566.
- Plitt, M.H., and Giocomo, L.M. (2021). Experience-dependent contextual codes in the hippocampus. *Nat Neurosci* 24, 705-714.
- Pola, G., Thiele, A., Hoffmann, K.P., and Panzeri, S. (2003). An exact method to quantify the information transmitted by different mechanisms of correlational coding. *Network* 14, 35-60.
- Schneidman, E., Bialek, W., and Berry, M.J., 2nd (2003). Synergy, redundancy, and independence in population codes. *J Neurosci* 23, 11539-11553.
- Zhao, S., Ting, J.T., Atallah, H.E., Qiu, L., Tan, J., Gloss, B., Augustine, G.J., Deisseroth, K., Luo, M., Graybiel, A.M., and Feng, G. (2011). Cell type-specific channelrhodopsin-2 transgenic mice for optogenetic dissection of neural circuitry function. *Nat Methods* 8, 745-752.

REVIEWERS' COMMENTS

Reviewer #1 (Remarks to the Author):

The authors have done an excellent job revising the manuscript, adding new analyses and discussions that greatly clarify the role of mixed selectivity in supporting flexible decision. The new experiment comparing naive vs. trained animals, which rules out visual adaptation, is a substantial addition to support the notion that mixed selectivity arises to support the decision behavior. All my comments are satisfactorily addressed. The study represents a huge amount of work that is careful done, which should be commended for. The findings advance new ideas about how flexible decisions are computed in the brain. I congratulate the authors on this exciting study. These findings will have a substantial impact in the field.

Reviewer #2 (Remarks to the Author):

The authors have adequately addressed my concerns.

We thank the reviewers for their suggestions and feedback, which have helped to significantly improve the paper.

Reviewer #1 (Remarks to the Author)

The authors have done an excellent job revising the manuscript, adding new analyses and discussions that greatly clarify the role of mixed selectivity in supporting flexible decision. The new experiment comparing naive vs. trained animals, which rules out visual adaptation, is a substantial addition to support the notion that mixed selectivity arises to support the decision behavior. All my comments are satisfactorily addressed. The study represents a huge amount of work that is carefully done, which should be commended for. The findings advance new ideas about how flexible decisions are computed in the brain. I congratulate the authors on this exciting study. These findings will have a substantial impact in the field.

Thank you for these positive comments and for the constructive suggestions throughout the peer review process.

Reviewer #2 (Remarks to the Author)

The authors have adequately addressed my concerns.

We thank the reviewer for their time and valuable input.